# What Preferences Can—and Cannot—Predict in Multi-Agent Online Learning

**Omar Abbadi** [1 2]   **Rida Laraki** [1]   **Panayotis Mertikopoulos** [2]

## Abstract

We examine the interplay between ordinal, preference-based solution concepts in games and the outcomes of payoff-driven learning dynamics, asking to what extent the combinatorial data of a game—its *preference graph*—can predict the long-run behavior of no-regret dynamics such as *follow-the-regularized-leader* (FTRL). In one direction, we show that the skeleton of every *dynamically stable* set (i.e., the set of pure profiles it contains) must also be *preferentially stable*, that is, it must be closed under profitable deviations. We then ask the converse question: when are preferences sufficient to describe the long-run behavior of the players' learning dynamics? We begin by showing that preferences are indeed enough to fully characterize asymptotic stability in the case of *subgames*—i.e., subsets of pure profiles obtained by restricting players' action sets. Beyond this case however, the equivalence between dynamic and preferential stability breaks down: in particular, we construct a three-player game with a preferentially stable set whose span is dynamically *unstable*, showing that preferences are *not sufficient* to describe dynamically stable behavior in general. To restore stability, we introduce the notion of *leaklessness*, a measure of aggregate payoff drift away from a set of pure profiles, and we use it to identify a payoff-based condition guaranteeing that the span of a set of pure profiles is stable and attracting.

## 1. Introduction

*Can players learn to behave rationally by adapting to feedback from their environment?*

[1]Moroccan Center for Game Theory, UM6P, 11103 Rabat, Morocco [2]Univ. Grenoble Alpes, CNRS, Inria, Grenoble INP, LIG, 38000 Grenoble, France. Correspondence to: Omar Abbadi <omar.abbadi@um6p.ma>.

*Proceedings of the 43rd International Conference on Machine Learning*, Seoul, South Korea. PMLR 306, 2026. Copyright 2026 by the author(s).

This question has been at the forefront of non-cooperative game theory and multi-agent learning [15, 23, 53], where interacting agents are commonly modeled through their governing dynamics [33, 58]. In particular, in many modern applications—from recommender platforms and auctions to large-scale multi-agent reinforcement learning [4, 25, 27, 44, 55]—the classical premises of full rationality, knowledge of the game, equilibrium computation, and consistently optimal play are unrealistic, so it is natural to take the above question as a starting point and ask what it implies for the players' long-run behavior.

In this regard, we focus on the dynamics of *no-regret learning*, and in particular the dynamics of *follow-the-regularized-leader* (FTRL) [32, 59], which form a canonical class of no-regret procedures in online learning [15]. Under these dynamics, player react to *cumulative* scores rather than instantaneous payoffs [19, 48], so correlation between players can emerge endogenously as the dynamics unfolds, prompting various forms of conflict and cooperation [21, 30, 40, 49]. At the same time, this richness makes asymptotic behavior in *general* finite games hard to characterize [3]. Indeed, while strict Nash equilibria are precisely the locally stable and attracting points of regularized learning [22, 26], they need not govern global behavior, and in their absence trajectories may be recurrent or chaotic [16, 41, 42, 46, 52]. Moreover, impossibility results rule out uncoupled dynamics that converge to Nash equilibrium in all games [7, 31, 50]. These considerations reveal that the relevant asymptotic objects are often *set-valued*. Moreover by characterizing these stable and attracting sets for the continuous-time learning dynamics, one can transfer conclusions to discrete-time algorithmic implementations via the theory of stochastic approximation [6, 45]. Accordingly, rather than asking when learning converges to a particular equilibrium, we ask which *regions of the strategy space* can arise as stable long-run outcomes, and how structural features of the game—in particular the players' preferences—constrain them.

Along these lines, a common modeling stance in game theory is that the primitive object is players' preferences, while numerical payoffs are only a utility representation [11, 20, 43, 54]. Motivated by this viewpoint, Papadimitriou and Piliouras [56] conjectured—for the replicator dynamics, the continuous-time limit of exponential weights [34, 61, 63]—that attractors are determined by a graph en-

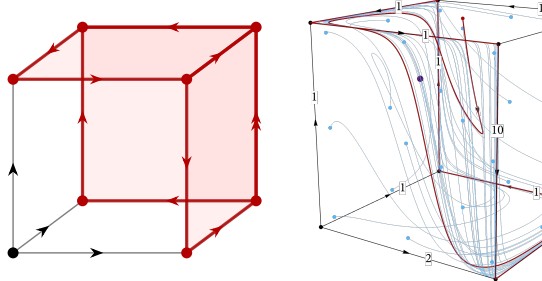

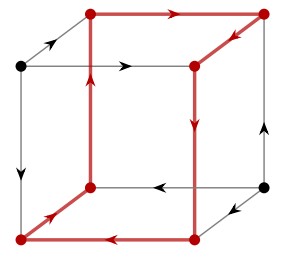

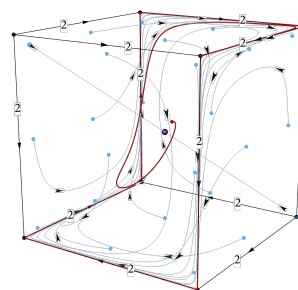

**(a)** When preferences and behavior do not align.

**(b)** When preferences and behavior do align.

**Figure 1.** In each of these $2 \times 2 \times 2$ games, vertices are pure profiles and arrows indicate the direction of profitable unilateral deviations. The part of the graph that is highlighted in red is closed under profitable deviations, so players have no *preferential* incentive to move away from these sets. Translating this intuition to the players' dynamic behavior fails on the left: trajectories starting arbitrarily close to the top face drift away toward the bottom face, contradicting the preferential viewpoint. On the right the preferential and dynamical viewpoints are aligned: trajectories starting anywhere near the preferentially closed 6-edge cycle (highlighted in red) converge to it.

coding (weakly) profitable unilateral deviations. This conjecture was made precise in subsequent work by Biggar and Shames [9], who, exploiting invariance properties of the replicator flow, showed that every attractor of the dynamics must contain "terminal components" of this graph. Nonetheless, although this clarifies that preferences and learning are intimately linked, it does *not* yield predictive power, as it does not specify which sets are stable outcomes beyond requiring that they contain those components. It thus remains far from clear whether preferences are *enough* to characterize stable outcomes and, if not, how to ensure stability in general finite games and under a broader class of dynamics.

In light of this, our main message is that this "preferences-only" intuition *fails* in general: for the replicator and regularized dynamics, ordinal information need not determine stability, and payoff *magnitudes* can be essential for predicting long-run behavior. This is particularly relevant in multi-agent machine learning, where a designer must instantiate *numerical* rewards to implement an intended *ordinal* specification—such as in preference-learning [17, 28] or reward specification [2] in the context of AI safety—and different cardinal realizations of the same ordering can induce different emergent behavior. Accordingly, we develop a theory for regularized dynamics that delineates when preferences do and do not pin down asymptotic outcomes, and how to recover stability and attraction in the latter case.

We do so through the game's *preference graph* [11], the directed graph on pure profiles whose arcs encode the directions of (weakly) profitable unilateral deviations, and which therefore retains only *ordinal* incentive information. Our guiding premise is that, notwithstanding this deliberate minimality, preferences can still impose strong constraints on stable learning outcomes, despite not fully characterizing them, and we therefore ask:

*To what extent do player preferences constrain their long-run behavior?*

Our results show that the preference graph places unavoidable constraints on the stable outcomes of regularized learning while also identifying regimes in which ordinal information alone cannot determine learning outcomes. Concretely:

1. For a broad class of regularizers, we show that any asymptotically stable set must contain every pure profile reachable from one of its pure states via unilateral improvements, that is, its *skeleton* must be closed under better replies. We next show that every attractor must contain the full mixed region spanned by each connected set of pure profiles it contains. In particular, if the preference graph is strongly connected, no proper attractor can exist.

2. For *subgames*, we show that preferences suffice: the span of a subgame is asymptotically stable if and only if it is closed under profitable deviations in the preference graph, complementing [9, 14, 57]. In particular, for *weakly acyclic* games this generically yields a sharp characterization of attractors, extending results previously known only for potential games [47].

3. Beyond subgames, we construct an example where a set of pure profiles is closed under better replies but its span is nevertheless unstable, even in the interior of the strategy space, thus refuting a recent conjecture of Biggar and Papadimitriou [8] and establishing that preferential and dynamical stability need not coincide.

4. We then introduce a sufficient condition for asymptotic stability—which we call *leaklessness*—that is invisible to the preference graph (ordinal information) and instead depends on the *magnitudes* of pure individual deviations (cardinal information). Importantly, leaklessness is, to our knowledge, the first condition of this kind that works for arbitrary finite games and arbitrary spans. Moreover, it admits a natural game-theoretic interpretation in terms of *robustness to aggregate unilateral deviations*, thereby yielding a novel setwise generalization of the notion of pure Nash equilibrium.

All in all, these results develop a preference-based theory for regularized payoff-driven dynamics—thus going beyond the replicator—and clarify how the preference graph *constrains*, but does not always *determine*, long-run learning behavior.

## 2. Preliminaries

**Game and strategies.** We consider a finite normal-form game $\mathcal{G}$, specified by a player set $\mathcal{N} = \{1, \ldots, n\}$, finite action sets $\mathcal{A}_i$ and payoff functions $u_i : \mathcal{A} \to \mathbb{R}$ for each $i \in \mathcal{N}$, where $\mathcal{A} := \prod_{i \in \mathcal{N}} \mathcal{A}_i$ is the set of pure profiles. A pure action profile is denoted by $\alpha = (\alpha_1, \ldots, \alpha_n) \in \mathcal{A}$, and for a given player $i$ we write $\alpha_{-i} = (\alpha_j)_{j \neq i}$ for the opponents' component of $\alpha$. In addition to pure actions, players may randomize over $\mathcal{A}_i$. Accordingly, player $i$'s mixed strategy space is the simplex $\mathcal{X}_i = \Delta(\mathcal{A}_i)$, the set of probability distributions over $\mathcal{A}_i$, and we write $\mathcal{X} := \prod_{i \in \mathcal{N}} \mathcal{X}_i$ for the set of mixed profiles $x = (x_1, \ldots, x_n)$, which we shall call the *strategy space*. For $\alpha_i \in \mathcal{A}_i$, the coordinate $x_{i\alpha}$ denotes the probability assigned to $\alpha_i$ by $x_i$,[1] and we set $x_{-i} = (x_j)_{j \neq i}$. In addition, we denote

$$x_\alpha := \prod_{i \in \mathcal{N}} x_{i\alpha}$$

for the probability of pure profile $\alpha$ being played. We also use the notation $\mathcal{X}_i^\circ$ for the relative interior of $\mathcal{X}_i$ (i.e., the set of mixed actions with full support) and similarly, $\mathcal{X}^\circ := \prod_i \mathcal{X}_i^\circ$ denote the relative interior of $\mathcal{X}$. By a slight abuse of notation, we identify each pure action $\alpha_i$ with the vertex $e_{\alpha_i}$ (i.e. the mixed strategy assigning probability 1 to $\alpha_i$), and likewise each pure profile $\alpha$ with the vertex $e_\alpha$. Finally, for $x_i \in \mathcal{X}_i$ and $x \in \mathcal{X}$, we define the *support*:

$$\operatorname{supp}(x_i) := \{\alpha_i \in \mathcal{A}_i : x_{i\alpha} \neq 0\}$$

and $\operatorname{supp}(x) := \prod_i \operatorname{supp}(x_i)$.

**Payoff field and equilibria.** The payoff functions extend multilinearly to mixed profiles via expectation under the product distribution induced by $x$:

$$u_i(x) = \sum_{\alpha \in \mathcal{A}} x_\alpha u_i(\alpha).$$

Denoting $\mathcal{Y}_i := \mathbb{R}^{\mathcal{A}_i}$ and $\mathcal{Y} := \prod_{i \in \mathcal{N}} \mathcal{Y}_i$ for the game's *payoff space*, we consider the *payoff field* $v : \mathcal{X} \to \mathcal{Y}$, defined componentwise by

$$v_{i\alpha}(x) = u_i(\alpha_i, x_{-i}), \quad \text{so that} \quad \langle v_i(x), x_i \rangle = u_i(x).$$

Equivalently, since $u_i(\,\cdot\,, x_{-i})$ is linear in $x_i$, the vector $v_i(x)$ coincides with the gradient of $u_i$ with respect to $x_i$:

$$v_i(x) = \nabla_{x_i} u_i(x).$$

---

[1]Stricto sensu, we should denote it $x_{i\alpha_i}$, but the $i$ subscript would be redundant, so we omit it from the action for readability.

A profile $x^* \in \mathcal{X}$ is a *Nash equilibrium* if no player can improve their payoff by a unilateral deviation from $x^*$, i.e., $u_i(x^*) \geq u_i(x_i, x^*_{-i})$ for all $x_i \in \mathcal{X}_i$ and all $i \in \mathcal{N}$. A *strict* Nash equilibrium is a pure profile $\alpha^* \in \mathcal{A}$ such that each player strictly prefers $\alpha_i^*$ against $\alpha^*_{-i}$ to any other pure action, namely $u_i(\alpha^*) > u_i(\alpha_i, \alpha^*_{-i})$ for all $\alpha_i \neq \alpha_i^*$.

## 3. Preferences, learning and stability

In this section, we shall define the preference-based solution concepts of interest and the regularized learning dynamics, along with their respective notions of stability.

### 3.1. Preference graphs and ordinal stability

**Preference graph.** We encode ordinal incentives via the game's *preference graph*. Its vertex set is $\mathcal{A}$, and it contains an arc $\alpha \to \alpha'$ whenever $\alpha$ and $\alpha'$ differ only in a single player's action (say player $i$, in which case $\alpha$ and $\alpha'$ are $i$-*comparable*) and the deviation is weakly profitable for $i$, i.e., $u_i(\alpha') \geq u_i(\alpha)$. Thus, for each pure profile, the outgoing edges indicate which comparable profiles are preferred by each player, hence inducing a (partial) order on the pure profiles. If $\alpha \to \alpha'$ and $\alpha' \to \alpha$ we say there is a *tie* and we denote it $\alpha \leftrightarrow \alpha'$. It is clear that a game having no ties is a *generic* property, meaning it holds for an open dense subset of games. Put differently, one can always find an arbitrarily small perturbation of the game which resolves ties.

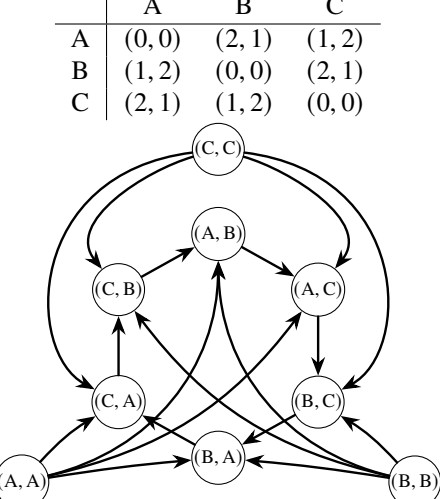

|   | A | B | C |
|---|---|---|---|
| A | (0, 0) | (2, 1) | (1, 2) |
| B | (1, 2) | (0, 0) | (2, 1) |
| C | (2, 1) | (1, 2) | (0, 0) |

**Figure 2.** Payoff table and preference graph of *Shapley's game* [60], a $3 \times 3$ game. The unique proper closed under better replies set, which is also strongly connected and hence a sink equilibrium, is the cycle of six profiles—the "hexagon"—in the middle. See [Appendix B](#) for more examples and details.

**Ordinal stability and connectedness.** We now define the preference graph notion capturing "stability under unilateral improvement":

**Definition 1.** We say $\mathcal{H} \subseteq \mathcal{A}$ is *closed under better replies* (club) if it has no outgoing arc to $\mathcal{A} \setminus \mathcal{H}$. Similarly, we say it is *closed under strict better replies* (s-club) if it has no outgoing arc unless it's a tie.

Equivalently, a club set is stable under (weakly) profitable unilateral deviation: if $\alpha \in \mathcal{H}$ and $\alpha \rightarrow \beta$ then $\beta \in \mathcal{H}$ as well. An s-club set is then stable under profitable unilateral deviations: if $\alpha \in \mathcal{H}$ and $\alpha \rightarrow \beta$ then either $\beta \in \mathcal{H}$ or $\beta \rightarrow \alpha$ as well. For concision, we will say a club / s-club set is *proper* if it is neither non-empty nor equal to $\mathcal{A}$.

We next turn from ordinal stability to a complementary structural feature: *connectedness*.

**Definition 2.** A set $\mathcal{H} \subseteq \mathcal{A}$ is *strongly connected* if any two vertices in $\mathcal{H}$ can be joined by a directed path (a unilateral improvement path) that remains in $\mathcal{H}$.

A nonempty set of pure profiles which is both club and strongly connected is called a *sink equilibrium*, it is then also a *minimal* club set, that is, a nonempty club set which contains no proper club set.

**Spans, subgames and skeletons.** Given any set of pure profiles $\mathcal{H} \subseteq \mathcal{A}$, we define its *span*[2] as the region of mixed profiles supported on $\mathcal{H}$:

$$\mathrm{span}(\mathcal{H}) := \{x \in \mathcal{X} : \forall \alpha \notin \mathcal{H}, x_\alpha = 0\}.$$

A set $\mathcal{B} \subseteq \mathcal{A}$ is a *subgame* if it has a product structure $\mathcal{B} = \prod_{i \in \mathcal{N}} \mathcal{B}_i$ with nonempty $\mathcal{B}_i \subseteq \mathcal{A}_i$. Its span is then the corresponding *face* $\mathcal{F} = \mathrm{span}(\mathcal{B}) = \prod_{i \in \mathcal{N}} \Delta(\mathcal{B}_i)$. More generally, for any $\mathcal{H} \subseteq \mathcal{A}$, its span corresponds to the union of faces of the strategy space for which all vertices are in $\mathcal{H}$. Finally, given a set $S \subseteq \mathcal{X}$ we define its *skeleton* as the set of pure profiles it contains, i.e. $S \cap \mathcal{A}$. It is clear that, given $\mathcal{H} \subseteq \mathcal{A}$, the skeleton of $\mathrm{span}(\mathcal{H})$ is $\mathcal{H}$. We give illustrations of these notions in Figure 3 below.

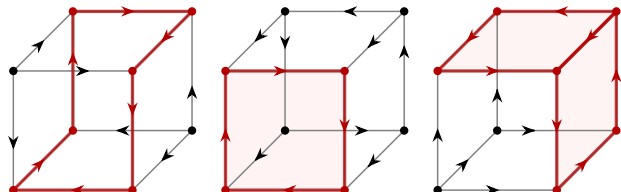

**Figure 3.** Three $2 \times 2 \times 2$ games, one player choses top or bottom, one left or right and one front or bottom. In each, the set of red vertices is both closed under better replies and strongly connected, and the red region represent its span (an edge cycle on the left, a square in the middle, and a union of two squares on the right).

*Remark* 1. Importantly, note that a pure profile $\alpha$ is a pure Nash equilibrium *if and only if* the singleton $\{\alpha\}$ is closed under strict better replies, since both are equivalent to $\alpha$

---

[2]This has also been called the *content* [8–10].

---

having no strictly profitable unilateral deviations. Similarly, $\alpha$ is a strict Nash equilibrium *if and only if* the singleton $\{\alpha\}$ is closed under better replies, in which case it will also be a sink equilibrium.

## 3.2. Regularized learning and dynamical stability

**Regret and FTRL** A central benchmark for adaptive behavior is *regret*. Fix a trajectory $t \mapsto x(t) \in \mathcal{X}$ and consider player $i$'s cumulative regret up to time $T$ against the best fixed mixed action in hindsight:

$$\mathrm{Reg}_i(T) := \max_{\alpha_i \in \mathcal{X}_i} \int_0^T \left(u_i(\alpha_i, x_{-i}(t)) - u_i(x(t))\right) dt.$$

We say that a learning rule is *no-regret* for player $i$ if its regret grows sublinearly with the horizon, i.e., $\mathrm{Reg}_i(T) = o(T)$. A broad and widely used family of no-regret dynamics is *follow-the-regularized-leader* (FTRL). The guiding principle is that each player $i$ maintains a vector of *scores*, i.e., aggregate payoffs, and selects a mixed action by trading off these scores against a regularization term that smoothens the dynamics and incentivizes exploration. Concretely, fix for each player $i$ a *regularizer* $h_i : \mathcal{X}_i \rightarrow \mathbb{R} \cup \{+\infty\}$ and define the associated *choice map* $Q_i : \mathcal{Y}_i \rightarrow \mathcal{X}_i$

$$Q_i(y_i) := \arg\max_{x_i \in \mathcal{X}_i} \{\langle y_i, x_i \rangle - h_i(x_i)\}.$$

With this notation, the FTRL dynamics can be written for each player $i$ as:

$$\underbrace{\dot{y}_i(t) = v_i(x(t))}_{\text{aggregate payoffs}} \qquad \underbrace{x_i(t) = Q_i(y_i(t))}_{\text{strategy update}}. \qquad \text{(FTRL)}$$

For the regularizer, we will assume for our purposes that it is *decomposable*, meaning that

$$h_i(x_i) = \sum_{\alpha_i \in \mathcal{A}_i} \theta_i(x_{i\alpha}), \quad \theta_i : [0,1] \rightarrow \mathbb{R},$$

where we call $\theta_i$ the *kernel* of $h_i$. To streamline our presentation, we will assume that $\theta_i$ is continuous on $[0,1]$, that $\theta_i \in C^2((0,1])$ (smoothness) and that there exists $K_i > 0$ such that, for all $p \in (0,1]$, $\theta_i''(p) \geq K_i$ (strong convexity). Moreover, for conciseness, we shall write $h := \sum_{i \in \mathcal{N}} h_i$ for the aggregate regularizer and $Q := (Q_i)_{i \in \mathcal{N}} : \mathcal{Y} \rightarrow \mathcal{X}$ for the induced choice map. This regularization viewpoint yields clean regret guarantees. Indeed [39] proved that in continuous time, along any solution of (FTRL), player $i$'s regret is uniformly bounded by the range of the regularizer:

$$\forall T \geq 0, \quad \mathrm{Reg}_i(T) \leq \max h_i - \min h_i,$$

In this sense, the regularizer influences performance, but it also plays a second, equally important role: it determines the geometry of the induced learning dynamics via the behavior of the choice map $Q_i$.

**Strategy dynamics.** We call $h_i$ *steep* if $\theta_i'(p) \to -\infty$ as $p \downarrow 0$. In that case the regularization cost becomes prohibitive near the boundary, so the regularized maximizer is always interior, that is, its image satisfies $\mathrm{Im}\, Q_i = \mathcal{X}_i^\circ$. By contrast, if $h_i$ is not steep, then boundary points may in general be selected and typically reached in finite time. For conciseness, we will say $h$ is steep if $h_i$ is steep for every player. Two canonical regularizers illustrate this dichotomy:

**Example 1.** For the *entropic* regularizer, with steep kernel $\theta_i(p) = p \log p$, the choice map is the softmax:

$$\Lambda_i(y_i) := \left( \frac{\exp(y_{i\alpha})}{\sum_{\beta_i \in \mathcal{A}_i} \exp(y_{i\beta})} \right)_{\alpha_i \in \mathcal{A}_i}.$$

which induces, in strategy space, the *replicator dynamics*

$$\dot{x}_{i\alpha} = x_{i\alpha}\big(v_{i\alpha}(x) - u_i(x)\big). \tag{RD}$$

**Example 2.** At the other extreme, for the *Euclidean* regularizer, whose kernel $\theta_i(p) = \frac{1}{2}p^2$ is not steep, the choice map is the Euclidean projection into $\mathcal{X}_i$:

$$\Pi_i(y_i) := \arg\min_{x_i \in \mathcal{X}_i} \|x_i - y_i\|_2^2.$$

Hence, the induced dynamics are the piecewise-smooth *projection dynamics* [47], and are most transparently expressed on each fixed-support face: if $x_i(t) \in \Delta(\mathcal{B}_i)^\circ$ for $t$ in some interval, then in that interval

$$\dot{x}_{i\alpha} = v_{i\alpha}(x) - \frac{1}{|\mathcal{B}_i|} \sum_{\beta_i \in \mathcal{B}_i} v_{i\beta}(x), \quad \alpha_i \in \mathcal{B}_i. \tag{PD}$$

When a coordinate hits 0, the support (hence the active face) may change in finite time, and the dynamics continue with the corresponding formula for the new support.

More generally, it is useful to eliminate the score variables and write the induced evolution directly in strategy space. On any time interval where player $i$'s support is fixed, say $x_i(t) \in \Delta(\mathcal{B}_i)^\circ$—writing

$$s_i := \frac{1}{\theta_i''}, \quad \pi_{i\alpha}(x_i) := \frac{s_i(x_{i\alpha})}{\sum_{\gamma_i \in \mathcal{B}_i} s_i(x_{i\gamma})} \text{ if } \alpha_i \in \mathcal{B}_i,$$

and $\pi_{i\alpha} = 0$ if $\alpha_i \notin \mathcal{B}_i$—the induced trajectory satisfies, for all $\alpha_i \in \mathcal{B}_i$, the *strategy dynamics*

$$\dot{x}_{i\alpha} = s_i(x_{i\alpha})\big(v_{i\alpha}(x) - \langle \pi_i(x_i), v_i(x)\rangle\big). \tag{SD}$$

Finally, fix the initial support $\mathcal{B} = \mathrm{supp}(x(0))$ and let $\mathcal{F} = \mathrm{span}(\mathcal{B})$ be the corresponding face. If the regularizer is steep, the support never changes and trajectories remain in $\mathcal{F}^\circ$ for all time. Thus, under some additional mild regularity assumptions on $s_i$, the facewise vector fields coincide on intersections of adjacent faces and therefore glue into a globally Lipschitz vector field on $\mathcal{X}$. Hence the induced strategy dynamics are globally well-posed for all $t \in \mathbb{R}$ and are *face-invariant*, meaning the support never changes along the dynamics. We call the resulting flow the *strategy flow* and denote it by $(X_t)_{t \in \mathbb{R}}$ (see Appendix E for more details).

**Dynamical stability and attraction.** Informally, stability means that trajectories starting sufficiently close to a set remain close for all future times, while attraction means that trajectories starting nearby converge to the set. Formally:

**Definition 3.** Let $x(t) = Q(y(t))$ be any solution orbit of (FTRL). Given $S \subseteq \mathcal{X}$ nonempty and closed, we will say $S$ is *(Lyapunov) stable* if for every neighborhood $U$ of $S$ there exists a neighborhood $V$ of $S$ such that

$$x(0) \in V \cap \mathrm{Im}\, Q \implies x(t) \in U \text{ for all } t \geq 0,$$

$S$ is *attracting* if there is a neighborhood $U$ of $S$ such that

$$x(0) \in U \cap \mathrm{Im}\, Q \implies \lim_{t \to +\infty} \mathrm{dist}(x(t), S) = 0,$$

and $S$ is *asymptotically stable* if it is stable and attracting.

As for the induced strategy flow $(X_t)_{t \in \mathbb{R}}$ of (SD), we shall work with the notion of *attractor*, which is an invariant[3] asymptotically stable set (see Appendix C for precise definitions and related discussions concerning attractors). Importantly, for this strategy flow notion of attractor, we allow nearby initializations on *all of* $\mathcal{X}$, including all adjacent faces, whereas for asymptotic stability under (FTRL), the only allowed initializations are on *the image of the choice map* (in particular, $\mathcal{X}^\circ$ for steep regularizers).

## 4. Preferences constrain long-run behavior

This section develops a systematic link between two levels of description: (1) combinatorial properties of subsets of the preference graph (ordinal stability notions), and (2) dynamical properties of regions of strategy space (dynamic stability notions). The guiding theme is that ordinal incentives, encoded by the preference graph, impose constraints on which regions of $\mathcal{X}$ can be long-run stable outcomes under regularized learning. All proofs are deferred to Appendix F.

We start with the first connection, which is that attraction for the strategy flow forces no outgoing better replies:

**Proposition 1.** *If $S \subseteq \mathcal{X}$ is attracting for the strategy flow induced by (SD), its skeleton is closed under better replies.*

The intuition behind this is that, under steep regularization, the dynamics are face-invariant, so starting on the 1-dimensional face spanned by two comparable vertices, the orbit stays on that segment, and the flow always follows the direction of the better reply. Outside the strategy flow regime, face-invariance is no longer available and we can no longer exploit initializations on proper subfaces of the strategy space. Nonetheless we are still able to prove, via new techniques utilizing the dual score representation of the dynamics, a strict version of the same result for stable

---

[3]For the strategy flow, invariant means that $S \subseteq \mathcal{X}$ satisfies $X_t(S) = S$ for all $t \in \mathbb{R}$.

sets, thereby showing that dynamical stability forces ordinal stability.

**Theorem 1.** *If $S \subseteq \mathcal{X}$ is stable under* (FTRL)*, then its skeleton is closed under strict better replies.*

Indeed, even without face-invariance, stability near a pure profile $\alpha$ rules out the existence of a *strict* better-reply $\alpha \rightarrow \alpha'$ because one can initialize scores so that the opponents remain close to $\alpha$ long enough for player $i$'s score difference $y_{i\alpha'} - y_{i\alpha}$ to grow linearly thanks to the strict payoff gap. Once this gap exceeds a certain carefully chosen regularizer-dependent threshold, the choice map shifts player $i$'s mixed action noticeably toward $\alpha_i'$, pushing $x(t)$ outside any prescribed small neighborhood of $\alpha$, contradicting stability. Thus, the whole difficulty lies in choosing the right initializations so that player $i$, the deviating player, has enough time to move outside the prescribed neighborhood while the other players remain (almost) frozen during that time, exploiting in particular the fact that the dynamics have *bounded speed* in score space.

The next result shows that strong connectivity in the preference graph imposes a constraint on the shape of attractors.

**Theorem 2.** *If $A \subseteq \mathcal{X}$ is an attractor of the strategy flow induced by* (SD) *and contains a strongly connected set $\mathcal{H} \subseteq \mathcal{A}$, then* span$(\mathcal{H}) \subseteq A$*. Hence, if the preference graph is strongly connected, the flow admits no proper[4] attractor.*

This theorem is a consequence of a general principle: connectedness in the preference graph propagates to a corresponding notion of connectedness for the dynamics on the associated mixed region, which generalizes the second theorem of [9]. Formally, this is captured by *chain transitivity*, which intuitively means that one can move from any point to any other by following the flow for arbitrarily long stretches, allowing only arbitrarily small perturbations in between. We refer to Appendix C for precise definitions. In order to establish this principle, we proceed by induction, proving that if $A$ contains all subfaces of a face, it must then contain the whole face as well, exploiting the fact that under the strategy flow, no asymptotically stable set can be fully contained in the interior [22].

# 5. When are preferences and long-run behavior equivalent?

Section 4 established one direction: asymptotic stability forces a range of closure and stability properties on the preference graph. We now ask when the converse holds for the natural geometric candidates arising from the graph, namely *spans of pure profiles*, that is, unions of faces of $\mathcal{X}$. In other words: when does preferential stability suffice

to guarantee dynamical stability of the region it spans, and when does it not?

## 5.1. When preferences are enough

We begin with the most simple class of spans: those corresponding to a single face, i.e., spans of subgames, which can be written as $\mathcal{B} = \prod_{i \in \mathcal{N}} \mathcal{B}_i$, with $\mathcal{B}_i \subseteq \mathcal{A}_i$. In this case, preferential stability turns out to imply asymptotic stability:

**Theorem 3.** *If $\mathcal{B} \subseteq \mathcal{A}$ is a subgame and closed under better replies, then* span$(\mathcal{B})$ *is asymptotically stable under* (FTRL)*.*

The proof (in Appendix G) relies on a recurring device in this section: constructing a suitable *energy function* which converts dissipativity in score space to asymptotic stability in strategy space. In the present case, the relevant energy is the *Fenchel gap*

$$F_{\mathcal{B}}(y) := h^*(y) - h_{\mathcal{B}}^*(y),$$

where $h^*$ is the convex conjugate of $h$ and $h_{\mathcal{B}}^*(y)$ is its restriction to the face spanned by $\mathcal{B}$. The Fenchel gap may be viewed as a *dual distance* to span$(\mathcal{B})$, since it is always nonnegative and vanishes exactly when $Q(y) \in$ span$(\mathcal{B})$. If $\mathcal{B}$ is a club set, then in a neighborhood of span$(\mathcal{B})$ every action outside $\mathcal{B}$ is uniformly worse than some action inside $\mathcal{B}$, yielding a differential inequality showing that $F_{\mathcal{B}}$ decreases at a rate proportional to the total probability mass of playing outside $\mathcal{B}$, or, equivalently, to the $\ell_1$ distance to its span. Therefore, the dynamics exhibit a sharp inward drift toward span$(\mathcal{B})$, which implies attraction and stability. Importantly, this energy function works even for regularizers that may be not steep, which makes it a substantial improvement over recent prior approaches [13, 14].

Now, combining the two theorems above with the results of Section 4, we obtain the following strong equivalences:

**Corollary 1.** *Let $\mathcal{B} \subseteq \mathcal{A}$ be a subgame and assume the game has no ties. Then $\mathcal{B}$ is closed under better replies if and only if* span$(\mathcal{B})$ *is asymptotically stable under* (FTRL)*.*

**Corollary 2.** *Let $\mathcal{B} \subseteq \mathcal{A}$ be a subgame. Then $\mathcal{B}$ is closed under better replies if and only if* span$(\mathcal{B})$ *is an attractor of the strategy flow induced by* (SD)*.*

Note that these results are subgame generalizations of the pointwise version, which has been sometimes called "the folk theorem of evolutionary game theory" [48, 58] in the literature, and says that a point is asymptotically stable if and only if it is a strict Nash equilibrium [22]. Indeed, any strict equilibrium is a vertex of the preference graph with no outgoing edge, and therefore a singleton club subgame.

We next illustrate how these equivalences specialize in games where better-reply paths always lead to equilibrium.

---

[4]Proper here is in the sense of being a strict subset, that is, the attractor is not all of $\mathcal{X}$.

This is the reachability notion of *weak acyclicity*, introduced by Young [64].

**Definition 4.** A game is *weakly acyclic* if for every $\alpha \in \mathcal{A}$ there exists a finite path of better replies

$$\alpha = \alpha^0 \to \alpha^1 \to \cdots \to \alpha^k$$

ending at a pure Nash equilibrium $\alpha^k$.

*Remark* 2. Note that weakly acyclic games are a generalization of the widely studied class of *potential games*, and more generally *ordinal potential games*, introduced by Monderer and Shapley [51], which are equivalent to *acyclic* games, games whose preference graph admits no directed cycles.

Applying our results to the class of weakly acyclic games, we obtain a sharp characterization of *minimal attractors*.[5]

**Corollary 3.** *If the game is weakly acyclic and has no ties, then $M \subseteq \mathcal{X}$ is a minimal attractor of the strategy flow induced by* (SD) *if and only if $M$ is a strict Nash equilibrium.*

An important application of this result follows from Johnston et al. [35], who show that, in the regime of many players relative to the maximal number of actions per player, weak acyclicity becomes overwhelmingly likely: among games admitting at least one pure Nash equilibrium, the probability of being weakly acyclic converges to 1 at an exponential rate in the number of players. Consequently, for such "typical" large games with a pure Nash equilibrium, Corollary 3 guarantees, under the strategy flow, that all minimal attractors coincide with strict Nash equilibria. This gives theoretical insight as to why FTRL seems in practice to work well with games with many players but (relatively) few actions.

### 5.2. When preferences are *not* enough

The preceding results rely crucially on the product structure of subgames. When $\mathcal{H} \subseteq \mathcal{A}$ is *not* a subgame, preferential stability of $\mathcal{H}$ may fail to control the dynamical stability of its span. The next proposition makes this separation explicit.

**Proposition 2.** *There is a $2 \times 2 \times 2$ game with a unique proper closed under better replies set $\mathcal{H}$ for which* span$(\mathcal{H})$ *is not stable under* (FTRL) *with entropic regularizer.*

The game is represented in Figure 1a, where $\mathcal{H}$ is the set of red vertices, and its span $\mathcal{S} = \text{span}(\mathcal{H})$ is the union of the top, right and back faces. One can observe that initializing near the center of the top face, which is a rest point of the dynamics, the third player has an incentive to deviate toward the bottom face, which is *not* included in $\mathcal{S}$, before the two other players can "snap" back into another face of $\mathcal{S}$, thus rendering it unstable. In fact, in this example, the minimal attractor containing $\mathcal{H}$ is not even a span of pure profiles

---

[5]Here, minimal is in the sense of set inclusion, that is, attractors which contain no proper attractor.

(see Remark G.4). This counterexample also refutes a recent conjecture of Biggar and Papadimitriou [8] concerning their notion of *local sources* for the replicator dynamics. Indeed, intuitively, they conjecture that any span of sink equilibrium which is not an attractor of the replicator must admit an initialization in a *proper subface* for which the trajectory drifts away from the span, whereas in our example, the escape happens in the *interior* (see Remark G.5 for details).

### 5.3. Recovering dynamic stability

By Proposition 2, we see that preferences alone may be too coarse to determine dynamic stability: recovering it requires *a fortiori* additional information, such as payoff magnitudes. For this reason, we introduce a natural and simple payoff-dependent condition which restores stability.

**Definition 5.** Given a pure profile $\alpha \in \mathcal{A}$, we define the *leakage* to $\alpha$, denoted $l_\alpha : \mathcal{X} \to \mathbb{R}$, as

$$l_\alpha(x) \coloneqq \sum_{i \in \mathcal{N}} \left( u_i(\alpha_i, x_{-i}) - u_i(x) \right).$$

Moreover, given $\mathcal{H} \subseteq \mathcal{A}$, we will say $\mathcal{H}$ is *leakless* if for every $\alpha \notin \mathcal{H}$ and every $\beta \in \mathcal{H}$, we have

$$l_\alpha(\beta) \le 0,$$

and we will say it is *strictly leakless* if the inequality is strict.

The summand $u_i(\alpha_i, x_{-i}) - u_i(x)$ is the instantaneous gain to player $i$ from deviating unilaterally to $\alpha_i$ at $x$, so $l_\alpha(x)$ aggregates these unilateral gains along the profile $\alpha$. Strict leaklessness therefore means that every outside profile $\alpha \notin \mathcal{H}$ is unilaterally unappealing in aggregate against every $\beta \in \mathcal{H}$. Note that leaklessness depends only on the magnitudes of pure deviations, which naturally correspond to the weights of the preference graph: if $\alpha \to \beta$ are *i*-comparable, then the weight of that edge is $u_i(\beta) - u_i(\alpha)$. Moreover, from a graph-theoretic perspective, the leakage from $\beta$ to $\alpha$ is obtained by summing the signed weights of the first outgoing edges from $\beta$ along shortest paths to $\alpha$. Thus, $l_\alpha(\beta)$ may be interpreted as the total "payoff flow" leaking from $\beta$ toward $\alpha$. Notice additionally that leaklessness is a cardinal strengthening of ordinal closure:

**Proposition 3.** *Let $\mathcal{H} \subseteq \mathcal{A}$. If $\mathcal{H}$ is leakless (resp. strictly leakless), then $\mathcal{H}$ is closed under strict better replies (resp. closed under better replies).*

The diagram below resumes the implication chain between these notions:

$$
\begin{array}{ccc}
\text{leakless} & \Longrightarrow & \text{s-club} \\
\Uparrow & & \Uparrow \\
\text{strictly leakless} & \Longrightarrow & \text{club}
\end{array}
$$

Note that, in games without ties, club and s-club coincide, whereas leaklessness and strict leaklessness need not. Moreover, by Remark 1, in the pointwise case $\mathcal{H} = \{\alpha\}$, leaklessness, s-club, and pure Nash equilibrium *all coincide*. Likewise, strict leaklessness, club, and strict Nash equilibrium coincide in this case. Thus, when there are no ties, all of them are equivalent for singletons. Hence, leaklessness provides a setwise *cardinal* generalization of pure Nash equilibrium, just as closedness under strict better replies provides its setwise *ordinal* counterpart.

As for complexity, it is also clear that, given $\mathcal{H}$, leaklessness can be checked in $O(|\mathcal{N}|\,|\mathcal{H}|\,|\mathcal{A} \setminus \mathcal{H}|)$ steps. Moreover, one can *find* a leakless set in $O(|\mathcal{N}|\,|\mathcal{A}|^2)$ time as follows: define the directed graph $G_l$ on vertex set $\mathcal{A}$ with an arc $\beta \to \alpha$ if and only if $l_\alpha(\beta) > 0$ (resp. $l_\alpha(\beta) \geq 0$ for strict leaklessness). Then $\mathcal{H}$ is leakless if and only if it has no outgoing arcs in $G_l$, i.e., if and only if there is no edge $\beta \to \alpha$ with $\beta \in \mathcal{H}$ and $\alpha \notin \mathcal{H}$. Constructing $G_l$ requires computing all leakages between pure profiles, which takes $O(|\mathcal{N}|\,|\mathcal{A}|^2)$ time, and finding a subset with no outgoing edge is $O(|\mathcal{A}|^2)$ (via standard graph decomposition algorithms, see e.g. [62]), hence the total complexity is $O(|\mathcal{N}|\,|\mathcal{A}|^2)$. Thus, given access to the payoff table, leaklessness can be tested and found with a polynomial number of payoff queries in the size of the explicit normal-form representation.

This stands in sharp contrast with the convex relaxation of pure Nash equilibrium, namely *mixed Nash equilibrium*. Despite being the most widely studied solution concept in finite games, non-pure, mixed Nash equilibria are computationally hard: in the same normal-form payoff-table model, computing one is well known to be PPAD-hard [18]. They are moreover dynamically unappealing: they are *never* asymptotically stable under regularized learning [22]. In comparison, the setwise relaxation given by strict leaklessness recovers asymptotic stability of the entire span:

**Theorem 4.** *If $\mathcal{H} \subseteq \mathcal{A}$ is strictly leakless, then* $\mathrm{span}(\mathcal{H})$ *is an attractor of the strategy flow induced by* (SD).

The proof (in Appendix G) is based on the construction of an energy function adapted to general spans, defined by

$$\bar{F}_{\mathcal{H}}(y) := \sum_{\alpha \notin \mathcal{H}} e^{-F_h(\alpha, y)},$$

with the convention $e^{-\infty} = 0$, where $F_h(x, y) := h(x) + h^*(y) - \langle y, x \rangle$ is the *Fenchel coupling*, a primal-dual divergence between strategies and scores. In the steep case, along any (FTRL) orbit $x(t) = Q(y(t))$, we have $F_h(\alpha, y(t)) \to -\infty$ if and only if $x_\alpha(t) \to 0$, hence the energy vanishes exactly on $\mathrm{span}(\mathcal{H})$. Additionally, the time-derivative of this energy along the same orbit is given by

$$\frac{d}{dt}\bar{F}_{\mathcal{H}}(y(t)) = \sum_{\alpha \notin \mathcal{H}} e^{-F_h(\alpha,\,y(t))}\, l_\alpha(x(t)),$$

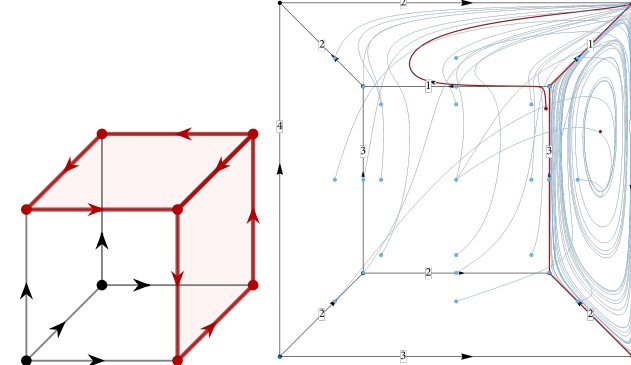

**Figure 4.** A game with a strictly leakless set (for the precise payoff structure of the game, see Appendix B). On the subfigure to the left, we plot the preference graph of the game, indicating the strictly leakless set (and its span) in red; on the subfigure to the right, we plot the evolution of the replicator dynamics in said game. We note that the (span of) the strictly leakless set is asymptotically stable, as per Theorem 5. At the same time, we also note that, even though all trajectories eventually converge to the rightmost face (a leaky subset of the original leakless set), the face itself is not asymptotically stable: a replicator trajectory may start arbitrarily close to said face and venture arbitrarily far before eventually returning to it (the trajectory highlighted in red illustrates precisely this behavior).

therefore the dynamics are dissipative near the span under strict leaklessness.

We conclude by specializing to entropic regularization, where we obtain the following refinement:

**Theorem 5.** *If $\mathcal{H} \subseteq \mathcal{A}$ is leakless and closed under better replies, then* $\mathrm{span}(\mathcal{H})$ *is an attractor of* (RD).

This result implies in particular, when combined with Proposition 3, that in any game with no ties, every span of leakless set is asymptotically stable under the replicator dynamics.

The proof (in Appendix G) relies on a strikingly simple energy function: it is given by the total outside mass under the strategy profile $x$

$$\bar{W}_{\mathcal{H}}(x) := \sum_{\alpha \notin \mathcal{H}} x_\alpha.$$

It is nonnegative and vanishes exactly on $\mathrm{span}(\mathcal{H})$. Along the replicator dynamics, the product rule gives

$$\dot{x}_\alpha = x_\alpha \sum_{i \in \mathcal{N}} \big(v_{i\alpha}(x) - u_i(x)\big) = x_\alpha l_\alpha(x),$$

so, using multilinearity of $l_\alpha(\cdot)$,

$$\frac{d}{dt}\bar{W}_{\mathcal{H}}(x(t)) = \sum_{\alpha \notin \mathcal{H}} \sum_{\beta \in \mathcal{H}} x_\alpha x_\beta l_\alpha(\beta)$$
$$+ \sum_{\alpha \notin \mathcal{H}} \sum_{\beta \notin \mathcal{H}} x_\alpha x_\beta l_\alpha(\beta).$$

The first sum is nonpositive by leaklessness, while the second is $O(\bar{W}_{\mathcal{H}}(x)^2)$. To get strict negativity near $\mathrm{span}(\mathcal{H})$,

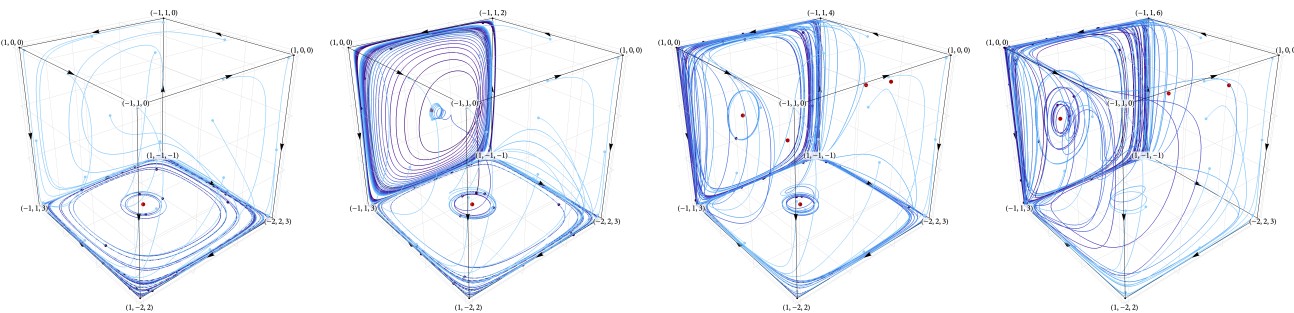

**Figure 5.** Four games with the same preference graph, but where only a single payoff has been modified. Notice how the behavior, from left to right, becomes progressively more and more chaotic and how trajectories start converging to different faces at different paces.

fix $x^* \in \mathrm{span}(\mathcal{H})$ and let $\mathcal{B} = \mathrm{supp}(x^*) \subseteq \mathcal{H}$. For every minimal profile $\alpha \notin \mathcal{H}$, in the deviation order, that can appear when moving out of $\mathcal{B}$, clubness gives a neighboring profile $\beta(\alpha) \in \mathcal{H}$ with

$$l_\alpha(\beta(\alpha)) < 0,$$

uniformly over such $\alpha$. Hence the first sum contains a strictly negative contribution of the form

$$\sum_\alpha x_\alpha x_{\beta(\alpha)} l_\alpha(\beta(\alpha)),$$

which dominates the second sum after shrinking sufficiently the neighborhood of $\mathrm{span}(\mathcal{H})$. Thus

$$\frac{d}{dt} \bar{W}_{\mathcal{H}}(x(t)) < 0$$

whenever $x(t)$ is sufficiently close to, but not in, $\mathrm{span}(\mathcal{H})$, hence $\bar{W}_{\mathcal{H}}$ is a local energy for the span.

Surprisingly, some intriguing and popular spans of club sets in the game theory literature—such as the 6-cycle in Jordan's Matching Pennies [36] or the hexagon in Shapley's game [60]—satisfy both clubness and leaklessness, yet fail to satisfy strict leaklessness (see Appendix B.2). In particular, for the Jordan game (which is represented in Figure 1b), existing proofs of asymptotic stability of the club span under the replicator dynamics [24, 37] rely on essentially ad-hoc arguments tailored to that game. It thus remained open to identify a general game-theoretic principle that can explain, beyond such special cases, the emergence of these *coordinated* and *correlated* outcomes under decentralized, uncoupled adaptive behavior. To the best of our knowledge the notion of leaklessness is the first explanation for this type of stability under regularized learning in games.

## 6. Concluding remarks

The results of this paper provide a step toward understanding the attractors of regularized payoff-based learning dynamics in games. Indeed, even though the preference graph provides a simple combinatorial blueprint that *constrains* the stable outcomes of FTRL, it is not enough to *characterize*

them. Moreover, although we have established that leaklessness is a natural sufficient condition to restore the asymptotic stability of spans, it is nonetheless *not* necessary in general (cf. the last example in Appendix B). Consequently, it remains open whether asymptotically stable spans admit a straightforward characterization in terms of payoff conditions that can be checked at the level of pure strategies, and whether they can be identified, or approximated, efficiently. Finally, while our results constrain attractors, a more complete account of the players' long-run behavior remains elusive and likely requires new ideas: for example, in Figure 5 the preference graphs are strongly connected, so no proper attractor exists, yet the observed asymptotic behavior is qualitatively different. Taken together, these questions outline a broad and ambitious agenda at the intersection of game theory and learning.

## Acknowledgments

We wish to express our deep gratitude to Josef Hofbauer for the fruitful and illuminating discussions, in particular regarding the counterexample.

This research was supported in part by the French National Research Agency (ANR) in the framework of the PEPR IA FOUNDRY project (ANR-23-PEIA-0003). PM is also a member of the Archimedes Research Unit/Athena RC, and was partially supported by project MIS 5154714 of the National Recovery and Resilience Plan Greece 2.0 funded by the European Union under the NextGenerationEU Program.

We would also like to thank the IRP Fair Games of the CNRS for supporting this project.

## Impact statement

This paper presents work whose goal is to advance the field of machine learning. There are many potential societal consequences of our work, none of which we feel must be specifically highlighted here.

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

# Appendices

# A. Related work

The study of attractors of the replicator dynamics originates in evolutionary game theory, where the replicator equation was introduced as a model of evolutionary selection [33, 58, 63]. Already in this literature, the long-run behavior of learning was understood to be set-valued, and a seminal contribution in this direction is due to Ritzberger and Weibull [57], who studied the asymptotic stability of faces under evolutionary dynamics and showed that subgames closed under better replies induce attracting faces, which is Theorem 3 for the special case of the replicator.

These questions later became central for learning in games. The replicator dynamics are the continuous-time limit of exponential weights [34, 61], while exponential weights is one of the few quintessential no-regret algorithm [15]. More generally, regularized dynamics, such as FTRL, provide a common, unified framework for no-regret learning [32, 48, 59]. Thus, the question of identifying attractors of the replicator naturally extends to the broader question of identifying the stable long-run outcomes of regularized learning dynamics.

For special classes of games, the picture is relatively well understood. In zero-sum and adversarial settings, regularized learning typically cycles rather than converges [46]. For zero-sum games, [10] showed that the span of the sink equilibrium is the unique global attractor of the replicator dynamics. Moreover, [41, 42] showed that harmonic games, a generalization of zero-sum games with interior equilibria, are Poincaré recurrent in the interior under FTRL, and therefore admit no proper attractor. By contrast, in potential games, the potential acts as a Lyapunov function, yielding convergence to Nash equilibria under broad classes of game dynamics [47]. Outside such structured classes, the long-run behavior of learning in finite games remains poorly understood.

A recent line of work studies this problem through the preference graph of the game. Papadimitriou and Piliouras [56] conjectured that, for the replicator dynamics, minimal attractors should exist and should correspond to sink equilibria of the preference graph. Biggar and Shames [9] proved that minimal attractors do exist and always contain sink equilibria. They also proved, for the replicator dynamics, a weaker form of Lemma F.5: any chain transitive set which contains a strongly connected set of pure profiles also contains its span. In the same paper, they also showed that club subgames are attracting for the replicator dynamics, essentially recovering the same result of Ritzberger and Weibull [57]. This had already been extended prior to discrete-time FTRL by [13]. However, as noted by [14], their proof contains a gap: the energy function used there does not generally induce neighborhoods of club faces, and hence does not by itself establish asymptotic stability. The energy of [14], based on the Bregman divergence, fixes this for steep regularizers (where they work with a stochastic version of continuous-time FTRL), while ours (the Fenchel gap) allows to prove the result for both steep and non-steep regularizers in continuous-time.

The conjectural correspondence between sink equilibria and minimal attractors was disproved by Biggar and Papadimitriou [8]. Their construction uses a notion they termed local sources, which certify escape from the span of a sink equilibrium through a proper face intersecting it. They conjectured that every non-attracting sink equilibrium span can be certified in this way. Our counterexample disproves this: the span can fail to be attracting even when the obstruction is not detected by any such local source. The same paper also introduced pseudoconvexity, a payoff-based condition guaranteeing attraction and stability of sink equilibrium spans in two-player games under the replicator dynamics. This condition does not directly extend to arbitrary numbers of players, whereas our leaklessness conditions apply to arbitrary finite games and arbitrary spans, and to the whole class of (steep) regularized dynamics.

Finally, [12] recently showed that the span of an induced cycle in the preference graph (a cycle for which each vertex has a unique outgoing edge) which is closed under better replies is an attractor of the replicator dynamics. The analogue of this result for more general regularizers remains open. This gives another type of span where preferences are enough for dynamic stability, and further shows that, even for spans and even for the replicator dynamics, a systematic and principled account of the attractors of learning in games is still missing.

# B. Preference graphs

We recall that the *preference graph* has vertex set $\mathcal{A}$ and an arc $\alpha \to \alpha'$ whenever the two profiles are $i$-comparable for some player $i \in N$ (i.e. $\alpha_{-i} = \alpha'_{-i}$) with $u_i(\alpha') \geq u_i(\alpha)$.

### B.1. Sink equilibria

Note that, given any directed graph, we can decompose its set of vertices into a disjoint union of *strongly connected components* (SCCs), which are strongly connected sets that admit no strongly connected superset. Given now such a decomposition, there will always exists among them some components which have no outgoing edge, known as *sink strongly connected components* (sink SCCs).

When applied to the preference graph, these sink SCCs have been called the *sink equilibria* of the game [11]. They are therefore nonempty subsets of pure profiles which are both closed under better replies and strongly connected in the preference graph. Equivalently, they are also nonempty minimal club sets, that is, club sets that contain no proper club set. Thus, once a better-reply path enters a sink equilibrium, it can keep moving within it, but it cannot leave it. In this sense, sink equilibrium represent the terminal "basins" of the ordinal deviation structure.

This interpretation is also reflected by a probabilistic viewpoint [29, 55, 56] : consider a random walk on $\mathcal{A}$ whose transitions $P$ follow the preference edges, i.e. $P(\alpha, \alpha') > 0$ if and only if $\alpha \to \alpha'$. An interpretation of such dynamics is that, at each turn, given we are on a pure profile $\alpha$, we pick a random player who has one or many profitable deviations from $\alpha$, who in turn picks at random such a deviation. Then, the classical ergodic theorem on Markov chains shows that the recurrent classes of this chain coincide with the sink SCCs of the preference graph, i.e. the sink equilibria, and every invariant probability measure $\pi$ is therefore supported on the union of sink equilibria.

### B.2. Examples

We now illustrate this with some example. In each case, the preference graph is drawn on the left, and the payoff table is given on the right. The sink equilibria vertices are colored red, and their spans are red as well.

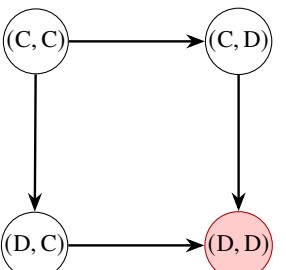

|   | C | D |
|---|---|---|
| C | $(3, 3)$ | $(0, 5)$ |
| D | $(5, 0)$ | $(1, 1)$ |

**Figure 6.** *Prisoner's Dilemma*

This is the well known *Prisoner's Dilemma*. The preference graph is acyclic: all paths flow to $(D, D)$, which is a sink vertex, and therefore the unique strict Nash equilibrium.

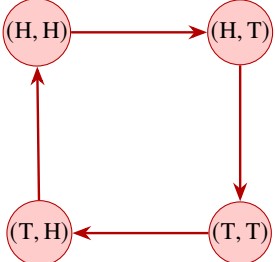

|   | H | T |
|---|---|---|
| H | $(1, -1)$ | $(-1, 1)$ |
| T | $(-1, 1)$ | $(1, -1)$ |

**Figure 7.** Matching Pennies

This is the *Matching Pennies* game. It is a zero-sum game, and its preference graph forms a directed 4–cycle, hence the graph is strongly connected and the unique club set is the whole graph.

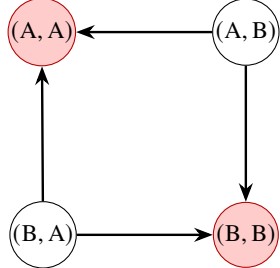

|   | A       | B       |
|---|---------|---------|
| A | $(1,1)$ | $(0,0)$ |
| B | $(0,0)$ | $(1,1)$ |

**Figure 8.** A coordination game.

The preference graph is acyclic and has two sink vertices, $(A, A)$ and $(B, B)$, so the game has two strict Nash equilibria (and hence two singleton club sets), corresponding to the two coordinated outcomes.

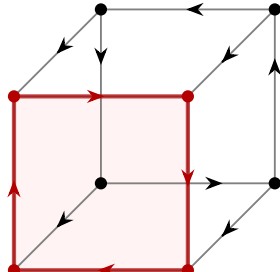

**Front**

|   | L         | R         |
|---|-----------|-----------|
| T | $(1,0,1)$ | $(0,1,1)$ |
| B | $(0,1,1)$ | $(1,0,1)$ |

**Back**

|   | L          | R          |
|---|------------|------------|
| T | $(-1,1,0)$ | $(1,0,0)$  |
| B | $(1,0,0)$  | $(-1,1,0)$ |

**Figure 9.** A cube with a club square.

A $2 \times 2 \times 2$ game, for which we represent pure profiles as the vertices of a cube. Player 1 choses to play top or bottom, player 2 left or right and player 3 front or back. Edges carry arrows for profitable unilateral deviations. The $2 \times 2$ subgame where the third player plays front only is closed under better replies.

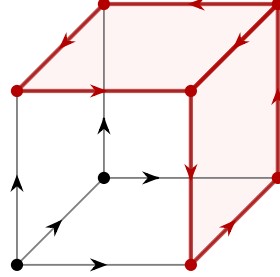

**Front**

|   | L           | R          |
|---|-------------|------------|
| T | $(2,-1,1)$  | $(0,1,1)$  |
| B | $(-2,-2,-2)$| $(1,1,-1)$ |

**Back**

|   | L           | R          |
|---|-------------|------------|
| T | $(1,0,-1)$  | $(1,-1,0)$ |
| B | $(-2,-2,0)$ | $(-2,0,1)$ |

**Figure 10.** A game with a stricly leakless set.

A $2 \times 2 \times 2$ game with a strictly leakless sink equilibrium. Indeed let $\mathcal{H}$ be that sink, a direct computation yields, $l_\alpha(\beta) \leq -1 < 0$ for all $\alpha \notin \mathcal{H}$ and all $\beta \in \mathcal{H}$.

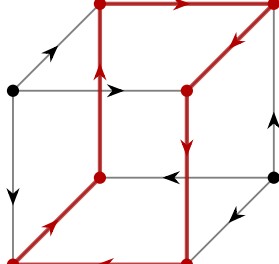

| Front | | |
|---|---|---|
| | L | R |
| T | $(-1,-1,-1)$ | $(-1,1,1)$ |
| B | $(1,1,-1)$ | $(1,-1,1)$ |

| Back | | |
|---|---|---|
| | L | R |
| T | $(1,-1,1)$ | $(1,1,-1)$ |
| B | $(-1,1,1)$ | $(-1,-1,-1)$ |

**Figure 11.** Jordan's Matching Pennies

This game, known as *Jordan's Matching Pennies*, was introduced by Jordan [36] and studied in [24] as a compact, explicit instance in which adaptive play can fail to converge to Nash equilibrium and instead exhibits persistent cycling concentrated on the red 6-cycle (the span of the sink equilibrium). It also provides a minimal example where long-run behavior is naturally described by sink equilibria rather than Nash equilibria [37].

Now, let $\mathcal{H}$ denote the sink equilibrium of Jordan's Matching Pennies (the six red vertices in the previous figure). Here $\mathcal{A} \setminus \mathcal{H} = \{(T, L, F), (B, R, B)\}$, and a direct computation gives $l_\alpha(\beta) \in \{0, -2\}$ for all such $\alpha \notin \mathcal{H}$ and $\beta \in \mathcal{H}$. Hence $\mathcal{H}$ is leakless but *not* strictly leakless.

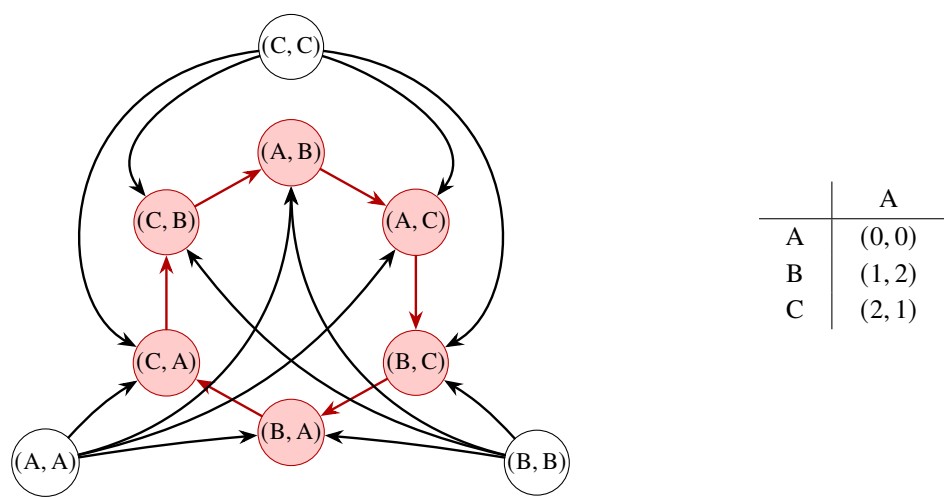

| | A | B | C |
|---|---|---|---|
| A | $(0,0)$ | $(2,1)$ | $(1,2)$ |
| B | $(1,2)$ | $(0,0)$ | $(2,1)$ |
| C | $(2,1)$ | $(1,2)$ | $(0,0)$ |

**Figure 12.** Shapley's game.

Shapley's classical $3 \times 3$ example [60] is historically important as (one of) the first explicit demonstrations that learning can fail to converge to Nash equilibrium in two-player non-zero-sum games, where it was shown specifically for the fictitious play scheme [38].

Let $\mathcal{H} := \{(i, j) : i \neq j\}$ be the sink equilibrium of Shapley's game (the red hexagon in the previous figure). Its span is exactly this directed cycle (union of one-dimensional faces). For $\alpha = (k, k) \notin \mathcal{H}$ and $\beta = (i, j) \in \mathcal{H}$,

$$l_\alpha(\beta) = \big(u_1(k, j) - u_1(i, j)\big) + \big(u_2(i, k) - u_2(i, j)\big).$$

If $k = i$ then $u_1(k, j) = u_1(i, j)$ and $u_2(i, k) = u_2(i, i) = 0$, so $l_\alpha(\beta) = -u_2(i, j) \in \{-1, -2\} < 0$. If $k = j$ similarly $l_\alpha(\beta) = -u_1(i, j) \in \{-1, -2\} < 0$. If $k \notin \{i, j\}$, then $u_1(i, j) + u_2(i, j) = 3$ and also $u_1(k, j) + u_2(i, k) = 3$, hence $l_\alpha(\beta) = 0$. Therefore $l_\alpha(\beta) \leq 0$ for all $\alpha \notin \mathcal{H}$, $\beta \in \mathcal{H}$, so $\mathcal{H}$ is leakless, but it is not strictly leakless since equality occurs (e.g. $\alpha = (A, A)$ and $\beta \in \{(B, C), (C, B)\}$).

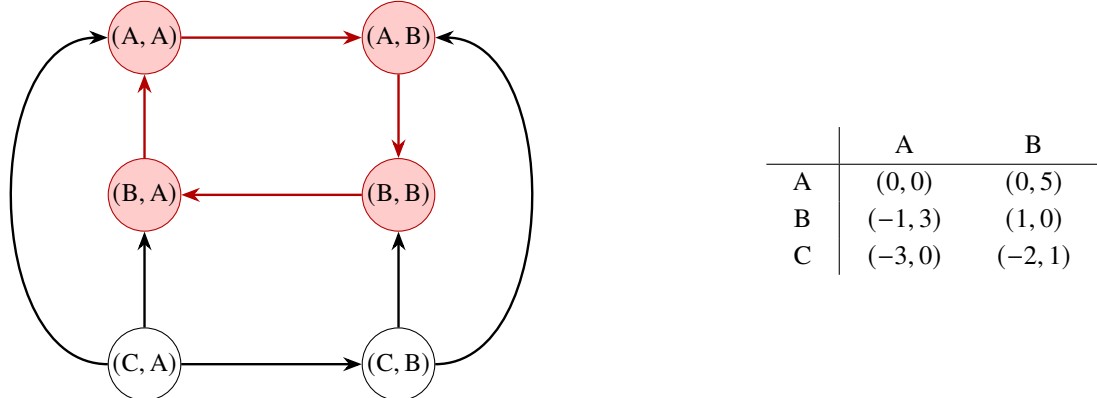

**Figure 13.** A club subgame which is not leakless.

The club subgame is $\mathcal{B} = \{A, B\} \times \{A, B\}$. Nonetheless, $\mathcal{B}$ is not leakless: with $\beta = (A, A) \in \mathcal{B}$ and $\alpha = (C, B) \notin \mathcal{B}$,

$$l_\alpha(\beta) = (u_1(C, A) - u_1(A, A)) + (u_2(A, B) - u_2(A, A)) = (-3 - 0) + (5 - 0) = 2 > 0.$$

Since club subgames are asymptotically stable, this shows leaklessness is not necessary for asymptotic stability.

## C. Attractors and chain transitivity

We collect here standard notions from dynamical systems used throughout this paper, see e.g. [1, 58].

Let $X$ be a compact subset of $\mathbb{R}^d$. We shall consider a *flow*, which is a continuous map $\phi : \mathbb{R} \times X \to X$ such that

$$\phi_0 = \mathrm{Id}_X, \qquad \phi_{t+s} = \phi_t \circ \phi_s \quad \forall\, s, t \in \mathbb{R}.$$

We write $\phi_t(x_0)$ for the state at time $t$ starting from $x_0 \in X$. All flows in this paper arise from (globally) well-posed ODEs of the form $\dot{x} = f(x)$, where $f : X \to \mathbb{R}^d$ is Lipschitz continuous.

### C.1. Attractors

A set $A \subseteq X$ is *forward-invariant* if $\phi_t(A) \subseteq A$ for all $t \geq 0$, and *invariant* if $\phi_t(A) = A$ for all $t \in \mathbb{R}$. A point $x^* \in X$ is *stationary* if $\phi_t(x^*) = x^*$ for all $t \geq 0$. Let $S \subseteq X$ be nonempty and closed and let $\mathcal{N}(S)$ denote the family of neighborhoods of $S$, then:

- *Stability.* $S$ is *stable* if for every $U \in \mathcal{N}(S)$ there exists $V \in \mathcal{N}(S)$ with

$$x \in V \implies \phi_t(x) \in U \quad \forall\, t \geq 0.$$

- *Attraction.* $S$ is *attracting* if there exists $U \in \mathcal{N}(S)$ such that

$$x \in U \implies \mathrm{dist}(\phi_t(x), S) \xrightarrow[t \to \infty]{} 0.$$

  Any such $U$ is a *basin of attraction* of $S$.

- *Asymptotic stability.* $S$ is *asymptotically stable* if it is both stable and attracting.

A nonempty, compact, invariant set $A \subseteq X$ is an *attractor* if there exists $U \in \mathcal{N}(A)$ with

$$\lim_{t \to \infty} \sup_{x \in U} \mathrm{dist}(\phi_t(x), A) = 0.$$

Equivalently, $A$ is an attractor if and only if there exists an open, forward-invariant *trapping region* $B$ and $t_0 > 0$ such that $\phi_t(\mathrm{cl}\, B) \subset B$ for all $t \geq t_0$ and $A = \bigcap_{t \geq 0} \phi_t(B)$. In particular, every attractor is asymptotically stable, and any compact, invariant and asymptotically stable set is an attractor [58]. If $A$ is an attractor with trapping region $B$, the *dual repeller* is

$$A^* := \bigcap_{t < 0} \phi_t(X \setminus B),$$

which is an attractor for the time-reversed flow $\psi_t := \phi_{-t}$ and satisfies $A \cap A^* = \varnothing$. Finally, we cite the following useful lemmas:

**Lemma C.1.** *Let $A \subseteq X$ be an attractor, let $K \subseteq X$ be nonempty, compact and invariant and assume $A \cap K \neq \varnothing$. Then $A \cap K$ is an attractor for the restricted flow $\phi|_K : \mathbb{R} \times K \to K$.*

### C.2. Chain transitivity

For $\varepsilon, T > 0$, an $(\varepsilon, T)$–*chain* from $x$ to $y$ is a finite sequence

$$x = x_0, \ x_1, \ \ldots, \ x_k = y, \qquad t_0, \ldots, t_{k-1} \geq T,$$

such that $\mathrm{dist}(\phi_{t_j}(x_j), x_{j+1}) < \varepsilon$ for all $j \in \{0, \ldots, k-1\}$. There is a *pseudo-orbit* from $x$ to $y$ (denoted $x \rightsquigarrow y$) if for every $\varepsilon, T > 0$ there exists an $(\varepsilon, T)$–chain from $x$ to $y$. Note that pseudo-orbits cannot leave, but may enter, attractors—and pseudo-orbits may leave, but cannot enter a repellor.

With this we can define the notion of *chain transitivity*. A nonempty compact set $K \subseteq X$ is *internally chain transitive* for the flow $\phi$ if it is invariant and, for every $x, y \in K$, $x \rightsquigarrow y$ for the flow *restricted to $K$*. We now mention the following alternative definition, due to [6]:

**Proposition C.2.** *Let $K \subseteq X$ be nonempty and compact, then $K$ is internally chain transitive if and only if $K$ is invariant and the flow restricted to $K$ admits no proper attractor.*

## D. Regularizers and choice maps

Throughout, fix a player $i \in \mathcal{N}$ and write $\mathcal{X}_i = \Delta(\mathcal{A}_i)$, $\mathcal{Y}_i = \mathbb{R}^{\mathcal{A}_i}$, $\langle \cdot, \cdot \rangle$ for the standard pairing on $\mathcal{Y}_i \times \mathcal{Y}_i$, and $\mathbf{1} \in \mathcal{Y}_i$ for the all-ones vector. We view $h_i : \mathcal{X}_i \to \mathbb{R} \cup \{+\infty\}$ as an extended-real convex function on $\mathcal{Y}_i$ by setting $h_i(x_i) = +\infty$ for $x_i \notin \mathcal{X}_i$. We assume throughout that $h_i$ is *decomposable*, *continuous* and *smooth on the interior* :

$$h_i(x_i) = \sum_{\alpha_i \in \mathcal{A}_i} \theta_i(x_{i\alpha}), \quad \theta_i : [0, 1] \to \mathbb{R}, \quad \theta_i \in C^0([0, 1]), \quad \theta_i \in C^2((0, 1]).$$

We will moreover assume that there exists a constant $K_i > 0$ such that

$$\theta_i''(p) \geq K_i \qquad \forall \, p \in (0, 1]. \tag{D.1}$$

As a consequence, $h_i$ is $K_i$-*strongly convex* on $\mathcal{X}_i$ (with respect to $\|\cdot\|_2$), i.e. for all $x_i, x_i' \in \mathcal{X}_i$ and all $t \in (0, 1]$,

$$h_i\big((1-t)x_i + tx_i'\big) \ \leq \ (1-t)h_i(x_i) + th_i(x_i') - \frac{K_i}{2}\, t(1-t)\, \|x_i - x_i'\|_2^2.$$

Finally, recall from the main text that $h_i$ is *steep* if $\theta_i'(p) \to -\infty$ as $p \downarrow 0$.

### D.1. Choice maps

The *convex conjugate* of $h_i$ is

$$h_i^*(y_i) \ := \ \sup_{x_i \in \mathcal{X}_i} \{\langle y_i, x_i \rangle - h_i(x_i)\}, \qquad y_i \in \mathcal{Y}_i,$$

and the regularized *choice map* (also known as the *mirror map*) is

$$Q_i(y_i) \ := \ \arg\max_{x_i \in \mathcal{X}_i} \{\langle y_i, x_i \rangle - h_i(x_i)\}.$$

By strong convexity, the maximizer is unique, so $Q_i : \mathcal{Y}_i \to \mathcal{X}_i$ is single-valued. We shall also consider the *subdifferential* of $h_i$ at $x_i \in \mathcal{X}_i$:

$$\partial h_i(x_i) \ := \ \Big\{g_i \in \mathcal{Y}_i : \ h_i(w_i) \geq h_i(x_i) + \langle g_i, w_i - x_i \rangle \ \forall w_i \in \mathcal{X}_i\Big\},$$

and we write $\mathrm{dom}\, \partial h_i := \{x_i \in \mathcal{X}_i : \partial h_i(x_i) \neq \varnothing\}$ and $\mathrm{Im}\, Q_i = Q_i(\mathcal{Y}_i)$ for the image of $Q_i$, also known as the *prox-domain*.

Under our assumptions, the choice map enjoys the following properties:

**Proposition D.1.** *For each $i \in \mathcal{N}$:*

1. $h_i^*$ *is* $C^1$ *on* $\mathcal{Y}_i$ *and* $Q_i = \nabla h_i^*$, *in particular* $Q_i$ *is Lipschitz continuous.*

2. $\operatorname{Im} Q_i = \operatorname{dom} \partial h_i$.

3. *For all* $c \in \mathbb{R}$, $Q_i(y_i + c\mathbf{1}) = Q_i(y_i)$.

4. *If* $y_i^k \in \mathcal{Y}_i$ *satisfies* $y_{i\alpha}^k - y_{i\beta}^k \to -\infty$ *for some* $\alpha \neq \beta$, *then* $Q_{i\alpha}(y_i^k) \to 0$.

5. *For* $x = Q(y)$ *there exist* $\lambda_i \in \mathbb{R}$ *and* $\nu_i \in \mathbb{R}_+^{\mathcal{A}_i}$ *such that, for all* $\alpha_i \in \mathcal{A}_i$,

$$y_{i\alpha} = \theta_i'(x_{i\alpha}) + \lambda_i - \nu_{i\alpha},$$

*and if* $x_{i\alpha} \neq 0$, *then* $\nu_{i\alpha} = 0$.

*Proof.* See [48]. $\qquad\square$

**Lemma D.2.** *If $h_i$ is steep, then $\operatorname{Im} Q_i = \mathcal{X}_i^\circ$.*

*Proof.* Direct consequence of Proposition D.1(2), see [48]. $\qquad\square$

**Proposition D.3.** *If $h_i$ is steep, then, for all $y_i, y_i' \in \mathcal{Y}_i$,*

$$Q_i(y_i) = Q_i(y_i') \quad \Longleftrightarrow \quad \exists c \in \mathbb{R}: \; y_i' = y_i + c\mathbf{1}.$$

*Equivalently, the restriction of $Q_i$ to the quotient space $\mathcal{Y}_i/\operatorname{span}\{\mathbf{1}\}$ is injective.*

*Proof.* Let $x_i := Q_i(y_i) = Q_i(y_i') \in \mathcal{X}_i^\circ$. Since $x_i$ is interior, by Proposition D.1(5)

$$y_{i\alpha} - \theta_i'(x_{i\alpha}) = \lambda, \qquad y_{i\alpha}' - \theta_i'(x_{i\alpha}) = \lambda', \qquad \forall\, \alpha_i \in \mathcal{A}_i.$$

Subtracting gives $y_{i\alpha}' - y_{i\alpha} = \lambda' - \lambda$ for all $\alpha_i$, hence $y_i' - y_i = c\mathbf{1}$ with $c := \lambda' - \lambda$. Conversely, if $y_i' = y_i + c\mathbf{1}$, then $Q_i(y_i') = Q_i(y_i)$ by Proposition D.1(3). $\qquad\square$

## D.2. Bregman divergence and Fenchel coupling

Fix $p_i \in \mathcal{X}_i$ and define the *face-neighborhood* of $p_i$ by

$$\Delta_{p_i} := \{x_i \in \mathcal{X}_i : \operatorname{supp}(x_i) \supseteq \operatorname{supp}(p_i)\}.$$

For $x_i \in \Delta_{p_i}$, the segment $x_i + t(p_i - x_i)$ stays in $\mathcal{X}_i$ for all sufficiently small $t \downarrow 0$. We therefore define the (one-sided) directional derivative

$$h_i'(x_i; z_i) := \lim_{t \downarrow 0} \frac{h_i(x_i + tz_i) - h_i(x_i)}{t}, \qquad z_i \in \mathcal{Y}_i \text{ with } x_i + tz_i \in \mathcal{X}_i \text{ for small } t > 0,$$

and the (extended) *Bregman divergence* by

$$D_i(p_i, x_i) := h_i(p_i) - h_i(x_i) - h_i'(x_i; p_i - x_i), \qquad x_i \in \Delta_{p_i}. \tag{D.2}$$

If $x_i \in \mathcal{X}_i^\circ$ then $h_i$ is differentiable at $x_i$ and $h_i'(x_i; p_i - x_i) = \langle \nabla h_i(x_i), p_i - x_i \rangle$, so (D.2) reduces to the usual Bregman divergence.

The associated *Fenchel coupling* is

$$F_i(p_i, y_i) := h_i(p_i) + h_i^*(y_i) - \langle y_i, p_i \rangle, \qquad (p_i, y_i) \in \mathcal{X}_i \times \mathcal{Y}_i,$$

and for profiles $p \in \mathcal{X}$, $y \in \mathcal{Y}$, we set $F_h(p, y) := \sum_{i \in \mathcal{N}} F_i(p_i, y_i)$.

**Proposition D.4.** *Assume $h_i$ is $K_i$-strongly convex on $\mathcal{X}_i$ w.r.t. $\|\cdot\|$. Then, for all $p_i \in \mathcal{X}_i$ and $y_i \in \mathcal{Y}_i$:*

1. $F_i(p_i, y_i) \geq \dfrac{K_i}{2} \|Q_i(y_i) - p_i\|^2.$

2. $F_i(p_i, y_i^k) \to 0$ if and only if $Q_i(y_i^k) \to p_i$.

3. If $x_i = Q_i(y_i) \in \Delta_{p_i}$, then $F_i(p_i, y_i) = D_i(p_i, x_i)$.

*Proof.* See [48]. □

In particular, $F_i(p_i, y_i) = 0$ if and only if $p_i = Q_i(y_i)$. Thus the Fenchel coupling acts as a kind of primal-dual distance between strategies and scores.

**Lemma D.5.** *Let $x = Q(y)$ be a solution orbit of* (FTRL). *Then for every fixed $p_i \in \mathcal{X}_i$,*

$$\frac{d}{dt} F_i(p_i, y_i) = \langle v_i(x), x_i - p_i \rangle.$$

*Proof.* By Proposition D.1(1), $\nabla h_i^* = Q_i$. Hence, by the chain rule,

$$\frac{d}{dt} F_i(p_i, y_i) = \langle \nabla h_i^*(y_i), \dot{y}_i \rangle - \langle \dot{y}_i(t), p_i \rangle = \langle \dot{y}_i, Q_i(y_i) - p_i \rangle.$$

Using $Q_i(y_i) = x_i$ and $\dot{y}_i = v_i(x)$ yields the claim. □

### D.3. Examples of regularizers

We give here some examples of standard regularizers [5, 13, 48, 59] which verify our assumptions. Recall $s_i(p) := 1/\theta_i''(p)$.

**Example D.6** (Entropy). Let $\theta_i(p) = p \log p$ for $p \in [0, 1]$ (with the convention $0 \log 0 = 0$). Then

$$\theta_i''(p) = \frac{1}{p} \geq 1 \quad (p \in (0, 1]),$$

so (D.1) holds with $K_i = 1$ and $s_i(p) = p$. The regularizer is steep and the choice map is the softmax

$$Q_i(y_i) = \Lambda_i(y_i) := \left( \frac{\exp(y_{i\alpha})}{\sum_{\beta_i \in \mathcal{A}_i} \exp(y_{i\beta})} \right)_{\alpha_i \in \mathcal{A}_i}, \qquad h_i^*(y_i) = \log \left( \sum_{\beta_i \in \mathcal{A}_i} \exp(y_{i\beta}) \right).$$

On $\Delta_{p_i}$, the Bregman divergence is the *Kullback–Leibler divergence*, or *relative entropy*:

$$D_i(p_i, x_i) = \sum_{\alpha_i \in \mathcal{A}_i} p_{i\alpha} \log \frac{p_{i\alpha}}{x_{i\alpha}} = D_{\mathrm{KL}}(p_i \| x_i).$$

**Example D.7** (Euclidean). Let $\theta_i(p) = \frac{1}{2} p^2$. Then $\theta_i''(p) = 1$, so (D.1) holds with $K_i = 1$ and $s_i(p) = 1$. The regularizer is non-steep and

$$Q_i(y_i) = \Pi_i(y_i) := \arg \min_{x_i \in \mathcal{X}_i} \|x_i - y_i\|_2^2.$$

Moreover,

$$h_i^*(y_i) = \sup_{x_i \in \mathcal{X}_i} \left\{ \langle y_i, x_i \rangle - \frac{1}{2} \|x_i\|_2^2 \right\} = \frac{1}{2} \|y_i\|_2^2 - \frac{1}{2} \operatorname{dist}(y_i, \mathcal{X}_i)^2,$$

and on $\mathcal{X}_i^\circ$ one has $D_i(p_i, x_i) = \frac{1}{2} \|p_i - x_i\|_2^2$.

**Example D.8** (Square-root). Let $\theta_i(p) = -4\sqrt{p}$ on $[0, 1]$. Then

$$\theta_i''(p) = p^{-3/2} \geq 1,$$

so (D.1) holds with $K_i = 1$ and $s_i(p) = p^{3/2}$. The regularizer is steep, and on $\Delta_{p_i}$ the Bregman divergence is

$$D_i(p_i, x_i) = 2 \sum_{\alpha_i \in \mathcal{A}_i} \left( \frac{(\sqrt{p_{i\alpha}} - \sqrt{x_{i\alpha}})^2}{\sqrt{x_{i\alpha}}} \right).$$

**Example D.9** (Tsallis entropy). Fix $\lambda \in (0, 1) \cup (1, 2]$ and let

$$\theta_i(p) = \frac{p^\lambda}{\lambda(\lambda - 1)}, \qquad p \in [0, 1].$$

Then, since $\lambda - 2 \leq 0$ we have

$$\theta_i''(p) = p^{\lambda-2} \geq 1 \quad (p \in (0, 1]),$$

so (D.1) holds with $K_i = 1$ and $s_i(p) = p^{2-\lambda}$. Moreover, if $\lambda \in (0, 1)$ then $\theta_i'(p) \to -\infty$ as $p \downarrow 0$, so it is steep, while if $\lambda \in (1, 2]$ then $\theta_i'(0+) = 0$, so it is non-steep. On $\Delta_{p_i}$,

$$D_i(p_i, x_i) = \frac{1}{\lambda(\lambda - 1)} \sum_{\alpha_i \in \mathcal{A}_i} \left( p_{i\alpha}^\lambda - \lambda x_{i\alpha}^{\lambda-1} p_{i\alpha} + (\lambda - 1) x_{i\alpha}^\lambda \right).$$

This family contains the Euclidean regularizer as the case $\lambda = 2$ and the square-root regularizer as the case $\lambda = 1/2$, while the entropy regularizer is recovered in the limit $\lambda \to 1$, up to affine terms on the simplex.

## E. Strategy dynamics

Consider continuous-time FTRL

$$\dot{y} = v(x), \qquad x = Q(y).$$

We start first with well-posedness in *score space*:

**Proposition E.1.** *$v \circ Q$ is Lipschitz continuous and the ODE $\dot{y} = v(Q(y))$ admits a unique global solution $y(\cdot) : \mathbb{R} \to \mathcal{X}$ from every initial condition $y(0)$.*

*Proof.* $v$ is multilinear by definition and bounded by finiteness of the game, $Q$ is Lipschitz continuous by Proposition D.1(1), hence their composition is Lipschitz continuous. Well-posedness then follows from standard Cauchy–Lipschitz / Picard–Lindelöf theorem. $\square$

We would like now to establish the existence of a flow in *strategy space*. For this purpose, fix $\mathcal{B}_i \subseteq \mathcal{A}_i$ and assume that on some interval $I$ the support of player $i$ is constant:

$$x_i(t) \in \Delta(\mathcal{B}_i)^\circ \qquad \forall t \in I.$$

Define

$$s_i(p) := \frac{1}{\theta_i''(p)}, \quad S_i(x_i) := \sum_{\beta_i \in \mathcal{B}_i} s_i(x_{i\beta}), \quad \pi_{i\alpha}(x_i) := \frac{s_i(x_{i\alpha})}{S_i(x_i)} \ (\alpha_i \in \mathcal{B}_i), \quad \pi_{i\alpha}(x_i) = 0 \ (\alpha_i \notin \mathcal{B}_i),$$

and write $\pi_i(x_i) = (\pi_{i\beta}(x_i))_{\beta_i \in \mathcal{A}_i}$.

**Proposition E.2.** *On $I$, for every $\alpha_i \in \mathcal{B}_i$,*

$$\dot{x}_{i\alpha} = s_i(x_{i\alpha}) \Big( v_{i\alpha}(x) - \langle \pi_i(x_i), v_i(x) \rangle \Big).$$

*Proof.* Following [48] and [22], on $I$, we have $x_{i\alpha}(t) > 0$ for $\alpha_i \in \mathcal{B}_i$, so $\theta_i$ is $C^2$ along these coordinates. The KKT conditions for the constrained maximization defining $x_i = Q_i(y_i)$ on the face $\Delta(\mathcal{B}_i)$ yield

$$y_{i\alpha} - \theta_i'(x_{i\alpha}) = \lambda_i, \qquad \alpha_i \in \mathcal{B}_i,$$

for some scalar multiplier $\lambda_i = \lambda_i(t)$ enforcing $\sum_{\alpha_i \in \mathcal{B}_i} x_{i\alpha} = 1$. Differentiating in time and using $\dot{y}_{i\alpha} = v_{i\alpha}(x)$ gives

$$v_{i\alpha}(x) - \theta_i''(x_{i\alpha}) \dot{x}_{i\alpha} = \dot{\lambda}_i,$$

hence

$$\dot{x}_{i\alpha} = s_i(x_{i\alpha})(v_{i\alpha}(x) - \dot{\lambda}_i).$$

Summing over $\alpha_i \in \mathcal{B}_i$ and using $\sum_{\alpha_i \in \mathcal{B}_i} \dot{x}_{i\alpha} = 0$ yields

$$0 = \sum_{\alpha_i \in \mathcal{B}_i} s_i(x_{i\alpha})(v_{i\alpha}(x) - \dot{\lambda}_i) \quad \Longrightarrow \quad \dot{\lambda}_i = \sum_{\alpha_i \in \mathcal{B}_i} \pi_{i\alpha}(x_i)\, v_{i\alpha}(x) = \langle \pi_i(x_i), v_i(x) \rangle,$$

which gives the stated formula. $\square$

*Remark* E.3. In the steep case, the choice map takes values in the relative interior, so the above facewise ODE applies globally along each trajectory (with $\mathcal{B}_i = \mathcal{A}_i$).

### E.1. Construction of the strategy flow

We now encode the induced dynamics directly on $\mathcal{X}$ by a globally defined vector field. This requires a regularity strengthening on $s_i$, which is the following Assumption.

**Assumption 1.** For each $i$, $s_i$ extends to a globally Lipschitz function on $[0, 1]$ with $s_i(0) = 0$

Notice that all the steep regularizers in Appendix D.3 verify this assumption. Note also that this assumption implies steepness:

**Proposition E.4.** *Under Assumption 1, $h_i$ is steep.*

*Proof.* Let $L > 0$ be a Lipschitz constant of $s_i$ on $[0, 1]$. Since $s_i(0) = 0$, we have $s_i(p) = |s_i(p) - s_i(0)| \le Lp$ for all $p \in [0, 1]$. Thus, for all $p \in (0, 1]$,

$$\theta_i''(p) = \frac{1}{s_i(p)} \ge \frac{1}{Lp}.$$

Fix $p_0 \in (0, 1]$. Since $\theta_i \in C^2((0, 1])$, $\theta_i'$ is absolutely continuous on $[p, p_0]$ for any $p \in (0, p_0)$, and

$$\theta_i'(p) = \theta_i'(p_0) - \int_p^{p_0} \theta_i''(t)\, dt \le \theta_i'(p_0) - \frac{1}{L} \int_p^{p_0} \frac{dt}{t} = \theta_i'(p_0) - \frac{1}{L} \log \frac{p_0}{p}.$$

Letting $p \downarrow 0$ yields $\theta_i'(p) \to -\infty$, i.e. $h_i$ is steep. $\square$

We now state a quick consequence of this, which will be useful for the contruction of the global flow.

**Lemma E.5.** *Under Assumption 1, for each $i$ there exists $\sigma_i > 0$ such that $S_i(x_i) \ge \sigma_i$ for all $x_i \in \mathcal{X}_i$. Consequently $\pi_i : \mathcal{X}_i \to \mathcal{X}_i$ is Lipschitz continuous.*

*Proof.* Let $m_i = |\mathcal{A}_i|$. For any $x_i \in \mathcal{X}_i$ there exists $\alpha$ with $x_{i\alpha} \ge 1/m_i$. Since $s_i$ is continuous on $[0, 1]$ and strictly positive on $(0, 1]$, the minimum

$$\sigma_i := \min_{p \in [1/m_i, 1]} s_i(p)$$

exists and satisfies $\sigma_i > 0$. Hence $S_i(x_i) \ge s_i(x_{i\alpha}) \ge \sigma_i$. Lipschitz continuity of $\pi_i$ follows because it is a quotient of Lipschitz continuous functions with denominator bounded away from 0. $\square$

**Definition E.6.** Define the *strategy field* $F : \mathcal{X} \to \mathcal{Y}$ by, for each $i \in \mathcal{N}$ and $\alpha_i \in \mathcal{A}_i$,

$$F_{i\alpha}(x) := s_i(x_{i\alpha})\Big(v_{i\alpha}(x) - \langle \pi_i(x_i), v_i(x) \rangle\Big).$$

**Proposition E.7.** *Under Assumption 1 the ODE*

$$\dot{x} = F(x), \qquad x(0) \in \mathcal{X},$$

*admits a unique global solution $x(\cdot) : \mathbb{R} \to \mathcal{X}$. This yields a continuous flow $(X_t)_{t \in \mathbb{R}}$ on $\mathcal{X}$—the* strategy flow—*which is invariant on each face of $\mathcal{X}$.*

*Proof.* Assumption 1 implies each $s_i$ is globally Lipschitz on $[0, 1]$ and, by Lemma E.5, the map $x_i \mapsto \pi_i(x_i)$ is also globally Lipschitz on $\mathcal{X}_i$. Moreover, by multilinearity, $v$ is globally Lipschitz on $\mathcal{X}$. Therefore each component $F_{i\alpha}(x)$ is a composition of globally Lipschitz maps, hence $F$ is globally Lipschitz on $\mathcal{X}$. Existence and uniqueness of global solutions for all $t \in \mathbb{R}$ follow from the Cauchy–Lipschitz / Picard–Lindelöf theorem. Moreover, for each $i$, using $s_i(x_{i\alpha}) = S_i(x_i)\pi_{i\alpha}(x_i)$ we obtain

$$\sum_{\alpha_i} \dot{x}_{i\alpha} = \sum_{\alpha_i} F_{i\alpha}(x) = \sum_{\alpha_i} s_i(x_{i\alpha})v_{i\alpha}(x) - \Big(\sum_{\alpha_i} s_i(x_{i\alpha})\Big)\langle \pi_i(x_i), v_i(x) \rangle = 0.$$

This shows $\sum_{\alpha_i} \dot{x}_{i\alpha} = 0$, hence $\sum_{\alpha_i} x_{i\alpha}(t) = 1$ for all $t$. Moreover, if $x_{i\alpha_i}(t) = 0$, then $s_i(0) = 0$ implies $\dot{x}_{i\alpha_i}(t) = 0$, so zero coordinates remain zero. Consequently, for every initial condition $x_0 \in \mathcal{X}$ and every $t \geq 0$,

$$\text{supp}(X_t(x_0)) \subseteq \text{supp}(x_0),$$

i.e. each face of $\mathcal{X}$ is forward-invariant. To upgrade this to invariance (two-sided), fix a face $\mathcal{F} = \prod_{i \in \mathcal{N}} \Delta(\mathcal{B}_i)$ and note that

$$x \in \mathcal{F} \quad \Longleftrightarrow \quad x_{i\alpha} = 0 \text{ for all } i \in \mathcal{N}, \ \alpha_i \in \mathcal{A}_i \setminus \mathcal{B}_i.$$

By the previous paragraph, if $x \in \mathcal{F}$ then $X_t(x) \in \mathcal{F}$ for all $t \geq 0$. Consider now the strategy flow $(X_t)_{t \in \mathbb{R}}$ which is defined for all $t \in \mathbb{R}$ and satisfies the flow property $X_{t+s} = X_t \circ X_s$ and $X_0 = \text{Id}$. Let $x \in \mathcal{F}$ and $t \in \mathbb{R}$. Put $w := X_{-t}(x)$. Then $X_t(w) = x$. Applying forward-invariance to the point $w$ over the time interval $[0, t]$ (if $t \geq 0$) gives $\text{supp}(X_t(w)) \subseteq \text{supp}(w)$, hence

$$\text{supp}(x) = \text{supp}(X_t(w)) \subseteq \text{supp}(w).$$

Thus $\text{supp}(w)$ contains only actions from $\mathcal{B}_i$ (since $\text{supp}(x) \subseteq \prod_i \mathcal{B}_i$), which means $w \in \mathcal{F}$. Equivalently, $X_{-t}(\mathcal{F}) \subseteq \mathcal{F}$. Applying $X_t$ to both sides yields $\mathcal{F} \subseteq X_t(\mathcal{F})$. Since we already have $X_t(\mathcal{F}) \subseteq \mathcal{F}$ for $t \geq 0$, it follows that $X_t(\mathcal{F}) = \mathcal{F}$ for all $t \geq 0$. Replacing $t$ by $-t$ gives $X_t(\mathcal{F}) = \mathcal{F}$ for all $t \in \mathbb{R}$. □

*Remark* E.8. Since $F$ is globally Lipschitz, the time-reversed flow is obtained by solving $\dot{x} = -F(x)$, i.e. $X_{-t}$ is the flow map of $-F$. Moreover, if one considers the negated game with payoffs $-u_i$ (so the payoff field becomes $-v$), then the induced strategy field is exactly $-F$ (replace $v$ by $-v$ in Definition E.6). Hence $(X_{-t})_{t \in \mathbb{R}}$ coincides with the strategy flow of the negated game.

Naturally, when $h$ is steep, we shall always assume Assumption 1 whenever we mention the strategy flow induced by (SD).

*Remark* E.9. Consider a subgame $\mathcal{B} = \prod_{i \in \mathcal{N}} \mathcal{B}_i$ and its face $\mathcal{F} = \text{span}(\mathcal{B})$, let $Q_{\mathcal{B}} : \mathcal{Y} \to \mathcal{F}^\circ$ be the face-restricted choice map $Q_{\mathcal{B}}(y) := \arg\max_{w \in \mathcal{F}} \{\langle y, w \rangle - h(w)\}$, and consider the facewise FTRL dynamics

$$\dot{y} = v(x), \qquad x = Q_{\mathcal{B}}(y). \tag{FTRL-$\mathcal{B}$}$$

Now, given $x_0 \in \mathcal{F}$, the strategy flow trajectory $(X_t)$ induced by (SD) with $X_0 = x_0$ coincides, by construction, with the (FTRL-$\mathcal{B}$) orbit $x(\cdot)$ with $Q_{\mathcal{B}}(y(0)) = x_0$ (which indeed exists, by Lemma D.2, and is unique by Proposition D.3). Hence the strategy flow can be seen as a globally Lipschitz continuous patchwork of these facewise FTRL dynamics.

## F. Omitted proofs from Section 4

We start by stating two properties of FTRL which will be useful for this section. We start with a powerful rationality property, proved in [48]:

**Theorem F.1.** *Under* (FTRL)*, any dominated mixed strategy becomes extinct along every orbit $x(t) = Q(y(t))$.*

Where we say that $x_i \in \mathcal{X}_i$ is *dominated* by $x_i' \in \mathcal{X}_i$ if $x_i'$ yields a strictly higher payoff against every opponents' profile: $u_i(x_i', w_{-i}) > u_i(x_i, w_{-i})$ for all $w_{-i} \in \mathcal{X}_{-i}$.

We will also use the following fact, proved by Flokas et al. [22].

**Theorem F.2.** *Let $A \subseteq \mathcal{X}^\circ$. Then $A$ is not asymptotically stable under* (FTRL)*. Consequently, if $h$ is steep, no subset of the relative interior of any face is asymptotically stable under the strategy flow induced by* (SD)*. In particular, every asymptotically stable set intersects the vertex set $\mathcal{A}$.*

### F.1. Stability and attraction

Recall first that we identify each pure action $\alpha_i \in \mathcal{A}_i$ with the corresponding vertex of the simplex $\mathcal{X}_i$ (and likewise each pure profile $\alpha \in \mathcal{A}$ with the corresponding vertex of $\mathcal{X}$).

The next argument hinges on two structural properties of the strategy flow $(X_t)$: *face-invariance* and *elimination of dominated strategies*. In fact, the same reasoning applies verbatim to any flow on $\mathcal{X}$ that satisfies these two properties. Together, they imply that on each one-dimensional face (edge) of the strategy space, the induced motion is aligned with player deviations: trajectories are confined to the edge by face-invariance, and elimination of dominated strategies forces them to drift toward the endpoint corresponding to the profitable unilateral deviation. In this sense, the flow on edges follows the orientation prescribed by the preference graph.

***Proof of [Proposition 1].*** Suppose, toward a contradiction, that $\alpha \in S$ and $\alpha \to \alpha'$ in the preference graph, but $\alpha' \notin S$. There exists then a unique player $i$ such that $\alpha$ and $\alpha'$ are $i$–comparable and $u_i(\alpha') \geq u_i(\alpha)$. Let $U$ be an attracting neighborhood of $S$. Because $\alpha \in S \subset U$ and $\alpha' \notin S$, shrinking $U$ if necessary we may assume $\alpha' \notin U$. By connectedness of the segment $[\alpha, \alpha']$, there exists $\beta \in (\alpha, \alpha') \cap U$. By face invariance under steep regularizers ([Proposition E.7]), $X_t(\beta) \in [\alpha, \alpha']$ for all $t \geq 0$.

If $u_i(\alpha') > u_i(\alpha)$, on the segment $[\alpha, \alpha']$, player $i$'s action $\alpha_i'$ strictly dominates $\alpha_i$, so by elimination of dominated strategies applied to this one–player game (see [Theorem F.1]) we have $X_t(\beta) \longrightarrow \alpha'$ as $t \to \infty$. Since $\beta \in U$ and $U$ is an attracting neighborhood of $S$, every limit point of $(X_t)$ starting from $\beta$ lies in $S$, hence $\alpha' \in S$, a contradiction.

If $u_i(\alpha') = u_i(\alpha)$, then every point of $[\alpha, \alpha']$ is stationary by face-invariance and the fact that a game with constant payoffs admits constant dynamics under ([FTRL]). Define now $\alpha(\lambda) := (1 - \lambda)\alpha + \lambda\alpha'$, $\lambda \in [0, 1]$, and set

$$\Lambda := \{\lambda \in [0, 1] : \ \alpha(\lambda) \in S\}.$$

Then $0 \in \Lambda$ and $\Lambda$ is closed (since $S$ is closed). Let $\lambda^* = \sup \Lambda$. If $\lambda^* < 1$, choose $\lambda \in (\lambda^*, 1)$ sufficiently close to $\lambda^*$ so that $\alpha(\lambda) \in U$. But $\alpha(\lambda)$ is stationary, and by attraction, this forces $\alpha(\lambda) \in S$, contradicting the choice of $\lambda^*$. Hence $\lambda^* = 1$ and $\alpha' = \alpha(1) \in S$, again a contradiction. $\qquad\square$

With this, we obtain the following useful corollary:

**Corollary F.3.** *Every asymptotically stable set for the strategy flow induced by* ([SD]) *contains a set of pure profiles which is closed under better replies.*

*Proof.* By theorem [Theorem F.2], every asymptotically stable set $S$ must intersect $\mathcal{A}$. [Proposition 1] then forces $S \cap \mathcal{A}$ to be closed under better replies. $\qquad\square$

To prove the next results, we will need the following lemma:

**Lemma F.4.** *Let $i \in \mathcal{N}$ and $\alpha_i \in \mathcal{A}_i$. For $G \geq 0$ define the gap region*

$$\Gamma_i(\alpha_i, G) := \{y_i \in \mathcal{Y}_i : \ y_{i\alpha} \geq y_{i\beta} + G \ \forall \beta \neq \alpha\},$$

*then, for every $r > 0$, there exists a threshold $G_i(\alpha_i, r) > 0$ such that for every $y_i \in \Gamma_i(\alpha_i, G_i(\alpha_i, r))$ one has $\|Q_i(y_i) - \alpha_i\|_1 \leq r$.*

*Proof.* Fix $r > 0$ and assume, toward a contradiction, that no such $G_i(\alpha_i, r)$ exists. Then for each integer $m \geq 1$ we can find $y_i^m \in \Gamma_i(\alpha_i, m)$ such that $\|Q_i(y_i^m) - \alpha_i\|_1 > r$. By [Proposition D.1(3)], replacing $y_i^m$ by $z_i^m := y_i^m - y_{i\alpha}^m \mathbf{1}$ does not change $Q_i$ and yields $z_{i\alpha}^m = 0$ and $z_{i\beta}^m \leq -m$ for all $\beta_i \neq \alpha_i$. Hence $z_{i\beta}^m - z_{i\alpha}^m \to -\infty$ for every $\beta_i \neq \alpha_i$. By [Proposition D.1(4)], this implies $Q_{i\beta}(z_i^m) \to 0$ for every $\beta_i \neq \alpha_i$, and therefore $Q_i(z_i^m) \to \alpha_i$ in $\|\cdot\|_1$, contradicting $\|Q_i(y_i^m) - \alpha_i\|_1 > r$. $\qquad\square$

In the next proof, due to the lack of face-invariance, we instead initialize the score dynamics so that not only play starts arbitrarily close to $\alpha$, but also all opponents are forced to remain close to $\alpha_{-i}$ over a prescribed time window by imposing large initial score gaps. During that window, the deviator's score difference between $\alpha_i$ and the strict better reply $\alpha_i'$ drifts at a uniform negative rate, and we pick the window length so that the choice map flips (up to a prescribed tolerance) toward $\alpha_i'$, forcing the orbit to exit a stable neighborhood.

***Proof of [Theorem 1].*** Let $S \subseteq \mathcal{X}$ be nonempty, closed, and stable under ([FTRL]), i.e.

$$x(0) \in V \cap \operatorname{Im} Q \implies x(t) \in U \quad \forall t \geq 0$$

for every neighborhood $U$ of $S$ and some neighborhood $V$ of $S$. Assume toward a contradiction that $\alpha \in S \cap \mathcal{A}$ and $\alpha \to \alpha'$ is a strict better reply but $\alpha' \notin S$. Let $i$ be the deviating player, so $\alpha' = (\alpha_i', \alpha_{-i})$, and set

$$\delta := u_i(\alpha_i', \alpha_{-i}) - u_i(\alpha_i, \alpha_{-i}) > 0.$$

Since $S$ is closed and $\alpha' \notin S$, let $r := \operatorname{dist}(\alpha', S) > 0$ and define the open neighborhood

$$U := \{x \in \mathcal{X} : \operatorname{dist}(x, S) < r/2\},$$

so $\alpha' \notin U$. By stability, there exists an open neighborhood $V \supseteq S$ such that every solution orbit $x = Q(y)$ with $x(0) \in V \cap \operatorname{Im} Q$ satisfies $x(t) \in U$ for all $t \geq 0$. Because $\alpha \in S \subseteq V$ and $V$ is open, pick $r_0 \in (0, r/8)$ such that $B_1(\alpha, r_0) \subseteq V$. Define the continuous function

$$g(x_{-i}) := u_i(\alpha_i', x_{-i}) - u_i(\alpha_i, x_{-i}), \qquad x_{-i} \in \mathcal{X}_{-i}.$$

Since $g(\alpha_{-i}) = \delta$, there exists $\kappa_0 > 0$ such that

$$\|x_{-i} - \alpha_{-i}\|_1 \leq \kappa_0 \implies g(x_{-i}) \geq \delta/2. \tag{F.1}$$

Set $\kappa := \min\{\kappa_0, r/8, r_0/4\}$. If $|\mathcal{N}| \geq 2$, choose $\rho := \min\{\kappa/2, r_0/2\}$ and numbers $(\rho_j)_{j \neq i}$ with $\rho_j > 0$ and $\sum_{j \neq i} \rho_j = \rho$. If $|\mathcal{N}| = 1$ set $\rho := 0$ and ignore all $j \neq i$ below. Next, we invoke Lemma F.4: for each $j \neq i$ choose $G_j > 0$ such that

$$y_j \in \Gamma_j(\alpha_j, G_j) \implies \|Q_j(y_j) - \alpha_j\|_1 \leq \rho_j. \tag{F.2}$$

Likewise, apply the same property twice for player $i$ to obtain $B > 0$ and $G_i' > 0$ such that

$$y_i \in \Gamma_i(\alpha_i, B) \implies \|Q_i(y_i) - \alpha_i\|_1 \leq r_0/2, \tag{F.3}$$
$$y_i \in \Gamma_i(\alpha_i', G_i') \implies \|Q_i(y_i) - \alpha_i'\|_1 \leq r/8. \tag{F.4}$$

Fix now the horizon

$$T_* := \frac{2(B + G_i')}{\delta}, \qquad \text{and set } T := T_*.$$

For each player $j \neq i$ define the payoff spread

$$M_j := \max_{x \in \mathcal{X}} \max_{\beta_j, \gamma_j \in \mathcal{A}_j} |v_{j\beta}(x) - v_{j\gamma}(x)| < \infty.$$

Initialize the opponents' scores by

$$y_{j\alpha}(0) = G_j + M_j T, \qquad y_{j\beta}(0) = 0, \quad \beta_j \neq \alpha_j.$$

For each $\beta_j \neq \alpha_j$ consider the gap $z_{j\beta}(t) := y_{j\alpha}(t) - y_{j\beta}(t)$. Then

$$\dot{z}_{j\beta}(t) = v_{j\alpha}(x(t)) - v_{j\beta}(x(t)) \geq -M_j,$$

so for $t \in [0, T]$,

$$z_{j\beta}(t) \geq z_{j\beta}(0) - M_j t \geq (G_j + M_j T) - M_j T = G_j.$$

Thus $y_j(t) \in \Gamma_j(\alpha_j, G_j)$ for all $t \in [0, T]$, and (F.2) yields

$$\|x_j(t) - \alpha_j\|_1 = \|Q_j(y_j(t)) - \alpha_j\|_1 \leq \rho_j, \qquad \forall t \in [0, T], \ \forall j \neq i. \tag{F.5}$$

In particular,

$$\|x_{-i}(t) - \alpha_{-i}\|_1 \leq \sum_{j \neq i} \rho_j = \rho \leq \kappa \leq \kappa_0, \qquad \forall t \in [0, T]. \tag{F.6}$$

By (F.1) and (F.6), for $t \in [0, T]$ we have

$$u_i(\alpha_i', x_{-i}(t)) - u_i(\alpha_i, x_{-i}(t)) = g(x_{-i}(t)) \geq \delta/2.$$

Now initialize player $i$'s scores by

$$y_{i\alpha}(0) = B, \qquad y_{i\alpha'}(0) = 0, \qquad y_{i\gamma}(0) = -R \quad (\gamma_i \notin \{\alpha_i, \alpha_i'\}),$$

where $R$ will be chosen momentarily. Then $y_i(0) \in \Gamma_i(\alpha_i, B)$, so by (F.3) we have $\|x_i(0) - \alpha_i\|_1 \leq r_0/2$. Combining with (F.5) at $t = 0$ gives

$$\|x(0) - \alpha\|_1 \leq \|x_i(0) - \alpha_i\|_1 + \sum_{j \neq i} \|x_j(0) - \alpha_j\|_1 \leq r_0/2 + \rho \leq r_0,$$

hence $x(0) \in B_1(\alpha, r_0) \subseteq V$. Since also $x(0) = Q(y(0)) \in \operatorname{Im} Q$, stability implies $x(t) \in U$ for all $t \geq 0$.

Define the deviator's score difference $z(t) := y_{i\alpha}(t) - y_{i\alpha'}(t)$. For $t \in [0, T]$,

$$\dot{z}(t) = v_{i\alpha}(x(t)) - v_{i\alpha'}(x(t)) = u_i(\alpha_i, x_{-i}(t)) - u_i(\alpha_i', x_{-i}(t)) \leq -\delta/2,$$

so

$$z(T) \leq z(0) - (\delta/2)T = B - (\delta/2) \cdot \frac{2(B + G_i')}{\delta} = -G_i',$$

i.e. $y_{i\alpha'}(T) \geq y_{i\alpha}(T) + G_i'$.

It remains to ensure that $\alpha_i'$ dominates every other action at time $T$ by margin $G_i'$. Let

$$M_i := \max_{x \in \mathcal{X}} \max_{\beta_i, \gamma_i \in \mathcal{A}_i} |v_{i\beta}(x) - v_{i\gamma}(x)|,$$

and for each $\gamma_i \notin \{\alpha_i, \alpha_i'\}$ define $w_\gamma(t) := y_{i\alpha'}(t) - y_{i\gamma}(t)$. Then $\dot{w}_\gamma(t) = v_{i\alpha'}(x(t)) - v_{i\gamma}(x(t)) \geq -M_i$, so for $t \in [0, T]$,

$$w_\gamma(t) \geq w_\gamma(0) - M_i t = R - M_i t.$$

Choose $R \geq G_i' + M_i T$. Then $w_\gamma(T) \geq G_i'$ for all $\gamma_i \notin \{\alpha_i, \alpha_i'\}$, and we already have $y_{i\alpha'}(T) \geq y_{i\alpha}(T) + G_i'$. Hence $y_i(T) \in \Gamma_i(\alpha_i', G_i')$, and (F.4) yields

$$\|x_i(T) - \alpha_i'\|_1 \leq r/8. \tag{F.7}$$

Also, by (F.5) at $t = T$,

$$\|x_{-i}(T) - \alpha_{-i}\|_1 \leq \sum_{j \neq i} \rho_j = \rho \leq r/8.$$

Therefore,

$$\|x(T) - \alpha'\|_1 \leq \|x_i(T) - \alpha_i'\|_1 + \|x_{-i}(T) - \alpha_{-i}\|_1 \leq r/8 + r/8 = r/4.$$

Since $x \mapsto \text{dist}(x, S)$ is 1-Lipschitz in $\|\cdot\|_1$, we have

$$\text{dist}(x(T), S) \geq \text{dist}(\alpha', S) - \|x(T) - \alpha'\|_1 \geq r - r/4 = 3r/4 > r/2,$$

so $x(T) \notin U$. This contradicts stability. $\square$

### F.2. Connectedness

The next lemma works by showing that the span of a strongly connected set cannot contain a proper attractor. Indeed, we proceed by contradiction, showing first that such an attractor would have to contain all vertices of the set, and then, by induction, show how that would force the existence of a fully interior repellor, which is impossible.

The next theorem shows that strong connectivity of pure profiles propagates to dynamical connectedness of their mixed span.

**Lemma F.5.** *If $\mathcal{H} \subseteq \mathcal{A}$ is strongly connected, then* $\text{span}(\mathcal{H})$ *is internally chain transitive for the strategy flow induced by* (SD).

*Proof.* Let $\mathcal{H} \subseteq \mathcal{A}$ be strongly connected. By Proposition C.2, it suffices to show that $\text{span}(\mathcal{H})$ is invariant and that the flow restricted to $\text{span}(\mathcal{H})$ admits no proper attractor.

By Proposition E.7, every face of $\mathcal{X}$ is invariant under the strategy flow. Since $\text{span}(\mathcal{H})$ is the union of all faces whose vertices lie in $\mathcal{H}$, it follows that $\text{span}(\mathcal{H})$ is invariant. Let $A \subseteq \text{span}(\mathcal{H})$ be an attractor for the restricted flow on $\text{span}(\mathcal{H})$. We show that $A = \text{span}(\mathcal{H})$.

Pick a face $\mathcal{F} \subseteq \text{span}(\mathcal{H})$ with $A \cap \mathcal{F} \neq \varnothing$. Because $\mathcal{F}$ is invariant, Lemma C.1 implies that $A \cap \mathcal{F}$ is an attractor for the flow restricted to $\mathcal{F}$. Hence $A \cap \mathcal{F}$ is asymptotically stable for the dynamics on $\mathcal{F}$, so by Theorem F.2 it intersects the vertex set of $\mathcal{F}$. Choose $\alpha \in (A \cap \mathcal{F}) \cap \mathcal{A}$. Since $\alpha \in \text{span}(\mathcal{H})$ and $\alpha$ is a vertex, we have $\alpha \in \mathcal{H}$.

We claim that $\mathcal{H} \subseteq A$. If not, pick $\alpha' \in \mathcal{H} \setminus A$. By strong connectivity of $\mathcal{H}$, there exists a directed path

$$\alpha = \alpha^0 \to \alpha^1 \to \cdots \to \alpha^k = \alpha' \quad \text{with all } \alpha^j \in \mathcal{H}.$$

Let $j$ be the smallest index such that $\alpha^j \in A$ and $\alpha^{j+1} \notin A$, and set $E := [\alpha^j, \alpha^{j+1}]$. By Proposition E.7, $E$ is invariant, and since $\alpha^j \in A \cap E$, Lemma C.1 shows that $A \cap E$ is an attractor for the restricted flow on $E$. Applying Proposition 1 to this

(one-dimensional) restricted flow yields that $(A \cap E) \cap \mathcal{A}$ is closed under better replies on $E$ and since $\alpha^j \to \alpha^{j+1}$, we get $\alpha^{j+1} \in A \cap E$, a contradiction. Thus $\mathcal{H} \subseteq A$.

Next, we prove that every face $\mathcal{F} \subseteq \mathrm{span}(\mathcal{H})$ satisfies $\mathcal{F} \subseteq A$, by induction on $d := \dim \mathcal{F}$. For $d = 0$, $\mathcal{F}$ is a vertex in $\mathcal{A}$, hence lies in $\mathcal{H} \subseteq A$. Assume $d \geq 1$ and the claim holds for all proper subfaces $\mathcal{F}' \subsetneq \mathcal{F}$. Note first that the boundary of $\mathcal{F}$ verifies $\partial \mathcal{F} = \bigcup_{\mathcal{F}' \subsetneq \mathcal{F}} \mathcal{F}' \subseteq A$. Let $A_{\mathcal{F}} := A \cap \mathcal{F}$, by Lemma C.1, $A_{\mathcal{F}}$ is an attractor for the restricted flow on $\mathcal{F}$. If $A_{\mathcal{F}} \neq \mathcal{F}$, consider the corresponding dual repellor $R_{\mathcal{F}}$ which is nonempty, compact, invariant in $\mathcal{F}$, and disjoint from $A_{\mathcal{F}}$. Since $\partial \mathcal{F} \subseteq A_{\mathcal{F}}$, we have $R_{\mathcal{F}} \cap \partial \mathcal{F} = \varnothing$, hence $R_{\mathcal{F}} \subseteq \mathcal{F}^{\circ}$. For the time-reversed flow on $\mathcal{F}$, the set $R_{\mathcal{F}}$ is an attractor, and by Remark E.8, this time-reversed flow is itself a strategy flow (of the negated game) on $\mathcal{F}$. This contradicts Theorem F.2, therefore $A_{\mathcal{F}} = \mathcal{F}$, i.e. $\mathcal{F} \subseteq A$.

By induction, every face $\mathcal{F} \subseteq \mathrm{span}(\mathcal{H})$ is contained in $A$, so $\mathrm{span}(\mathcal{H}) \subseteq A$, since always $A \subseteq \mathrm{span}(\mathcal{H})$, we conclude $A = \mathrm{span}(\mathcal{H})$. Thus the restricted flow on $\mathrm{span}(\mathcal{H})$ admits no proper attractor, and Proposition C.2 yields that $\mathrm{span}(\mathcal{H})$ is internally chain transitive. $\qquad \square$

***Proof of Theorem 2.*** Let $\mathcal{H} \subseteq \mathcal{A}$ be again strongly connected. Since $\mathrm{span}(\mathcal{H})$ is invariant, Lemma C.1 applied to the attractor $A$ and $\mathrm{span}(\mathcal{H})$ shows that $A \cap \mathrm{span}(\mathcal{H})$ is an attractor for the flow restricted to $\mathrm{span}(\mathcal{H})$ (nonempty because $\mathcal{H} \subseteq A \cap \mathrm{span}(\mathcal{H})$). By Lemma F.5, $\mathrm{span}(\mathcal{H})$ is internally chain transitive, hence the restricted flow on $\mathrm{span}(\mathcal{H})$ admits no proper attractor. Therefore $A \cap \mathrm{span}(\mathcal{H}) = \mathrm{span}(\mathcal{H})$, i.e. $\mathrm{span}(\mathcal{H}) \subseteq A$.

In the case the whole preference graph is strongly connected, applying this to $\mathcal{H} = \mathcal{A}$ yields that $\mathrm{span}(\mathcal{H}) = \mathcal{X}$ admits no proper attractor. $\qquad \square$

# G. Omitted proofs from Section 5

### G.1. Energy functions and asymptotic stability

Before we start, we shall first define the notion of *energy functions*, introduced in [45], which allow us to convert local dissipation in *score space* into asymptotic stability in *strategy space*. They are defined as follows:

**Definition G.1.** Let $S \subseteq \mathcal{X}$ be a nonempty closed set. Under (FTRL), a map $E : \mathcal{Y} \to [0, \infty)$ is a *local energy function* for $S$ if:

1. $E$ is Lipschitz and $C^1$ on $\mathcal{Y}$.
2. For every sequence $(y^k)_{k \in \mathbb{N}} \subseteq \mathcal{Y}$,
$$Q(y^k) \to S \iff E(y^k) \to 0.$$

3. There exists $\bar{E} > 0$ such that for all $0 < E^- < E^+ \leq \bar{E}$,
$$\sup\{\dot{E}(y) : E^- < E(y) < E^+\} < 0,$$

where $\dot{E}(y) := \langle \nabla E(y), v(Q(y)) \rangle$ is the derivative of $E$ along (FTRL).

The link with the long-run behavior of the continuous-time dynamics, which we prove, is as follows:

**Theorem G.2.** *Let $S \subseteq \mathcal{X}$ be nonempty and closed, if $S$ admits a local energy function, then $S$ is asymptotically stable under* (FTRL).

*Proof.* Let $y(\cdot)$ be a solution of $\dot{y} = v(Q(y))$ and set $x(t) = Q(y(t))$ and $\mathcal{E}(t) := E(y(t))$. By Definition G.1(1) and the chain rule, $\mathcal{E}$ is $C^1$ and $\dot{\mathcal{E}}(t) = \dot{E}(y(t))$.

First, for every neighborhood $U$ of $S$ in $\mathcal{X}$ there exists $\delta_U > 0$ such that
$$E(y) < \delta_U \implies Q(y) \in U. \tag{G.1}$$

Otherwise, there exist $U$ and $y_k$ with $E(y_k) \to 0$ but $Q(y_k) \notin U$, contradicting $Q(y_k) \to S$ from Definition G.1(2). Second, for every $\delta > 0$ there exists a neighborhood $V_\delta$ of $S$ such that
$$Q(y) \in V_\delta \implies E(y) < \delta. \tag{G.2}$$

Otherwise, there exist $\delta > 0$ and $y_m$ with $\text{dist}(Q(y_m), S) < 1/m$ but $E(y_m) \geq \delta$, contradicting Definition G.1(2).

By Definition G.1(3), fix $\bar{E} > 0$ such that for every $0 < E^- < E^+ \leq \bar{E}$,

$$\kappa(E^-, E^+) := -\sup\{\dot{E}(y) : E^- < E(y) < E^+\} > 0. \tag{$*$}$$

Hence, whenever $\mathcal{E}(t) \in (E^-, E^+)$, we have $\dot{\mathcal{E}}(t) \leq -\kappa(E^-, E^+)$. In particular, if $0 < a < b \leq \bar{E}$ and $\mathcal{E}(t_0) \leq a$, then $\mathcal{E}(t) \leq a$ for all $t \geq t_0$: otherwise, letting $\tau = \inf\{t \geq t_0 : \mathcal{E}(t) > a\}$ gives $\mathcal{E}(\tau) = a$ by continuity, and then continuity yields $\varepsilon > 0$ with $\mathcal{E}(t) \in (a, b)$ for $t \in (\tau, \tau + \varepsilon)$, so $\dot{\mathcal{E}}(t) \leq -\kappa(a, b) < 0$ on $(\tau, \tau + \varepsilon)$ by $(*)$, contradicting $\mathcal{E}(t) > a$ for $t > \tau$ close enough.

For stability, let $U$ be any neighborhood of $S$. Choose $\delta_U \in (0, \bar{E}]$ so that (G.1) holds, pick $a \in (0, \delta_U)$, and apply (G.2) with $\delta = a$ to obtain a neighborhood $V$ of $S$ such that $x(0) = Q(y(0)) \in V$ implies $\mathcal{E}(0) < a$. The previous paragraph (with $(a, b) = (a, \delta_U)$) yields $\mathcal{E}(t) \leq a < \delta_U$ for all $t \geq 0$, hence $x(t) \in U$ for all $t \geq 0$ by (G.1), hence $S$ is Lyapunov stable.

For attraction, fix $b \in (0, \bar{E}]$ and assume $\mathcal{E}(0) < b$. Let $a \in (0, b)$ be arbitrary. If $\mathcal{E}(0) \leq a$, set $T_a = 0$. Otherwise, if $\mathcal{E}(0) > a$ and $\mathcal{E}(t) > a$ for all $t \geq 0$, then $\mathcal{E}(t) \in (a, b)$ for all $t \geq 0$, so $\dot{\mathcal{E}}(t) \leq -\kappa(a, b)$ for all $t \geq 0$ and thus

$$\mathcal{E}(t) \leq \mathcal{E}(0) - \kappa(a, b)\, t,$$

which is impossible for $t > \mathcal{E}(0)/\kappa(a, b)$ because $\mathcal{E}(t) \geq 0$. Hence there exists $T_a$ with $\mathcal{E}(T_a) \leq a$, and the previous paragraph implies $\mathcal{E}(t) \leq a$ for all $t \geq T_a$. Since this holds for every $a \in (0, b)$, we obtain $\mathcal{E}(t) \to 0$ as $t \to \infty$. If $\text{dist}(x(t), S) \not\to 0$, there exist $\varepsilon > 0$ and $t_k \to \infty$ with $\text{dist}(x(t_k), S) \geq \varepsilon$ for all $k$. But $\mathcal{E}(t_k) = E(y(t_k)) \to 0$, so Definition G.1(2) implies $x(t_k) = Q(y(t_k)) \to S$, a contradiction, therefore $\text{dist}(x(t), S) \to 0$. Finally, applying (G.2) with $\delta = b$ yields a neighborhood $U_0$ of $S$ such that $x(0) \in U_0 \cap \text{Im}\, Q$ implies $\mathcal{E}(0) < b$, the above then shows $\text{dist}(x(t), S) \to 0$ for all such initial conditions, so $S$ is attracting. $\qquad\square$

## G.2. When preferences are enough

We start by constructing an energy function for spans of club subgames.

**Lemma G.3.** *Let $\mathcal{B} \subseteq \mathcal{A}$ be a subgame. If $\mathcal{B}$ is closed under better replies, then the Fenchel gap*

$$F_{\mathcal{B}}(y) := h^*(y) - h_{\mathcal{B}}^*(y), \qquad h_{\mathcal{B}}^*(y) := \max_{w \in \text{span}(\mathcal{B})} \{\langle y, w \rangle - h(w)\},$$

*is a local energy function for* $\text{span}(\mathcal{B})$.

*Proof.* Write $\bar{\mathcal{B}}_i := \mathcal{A}_i \setminus \mathcal{B}_i$ and $\mathcal{F} := \text{span}(\mathcal{B}) = \prod_{i \in \mathcal{N}} \Delta(\mathcal{B}_i)$. For each $i \in \mathcal{N}$, define the restricted choice map[6]

$$Q_{\mathcal{B}_i}(y_i) := \arg \max_{x_i \in \Delta(\mathcal{B}_i)} \{\langle y_i, x_i \rangle - h_i(x_i)\}, \qquad Q_{\mathcal{B}}(y) := (Q_{\mathcal{B}_i}(y_i))_{i \in \mathcal{N}} \in \mathcal{F},$$

and set, throughout the proof,

$$x := Q(y), \qquad x^{\mathcal{B}} := Q_{\mathcal{B}}(y), \qquad \Delta_i := x_i - x_i^{\mathcal{B}}, \qquad m_i := \sum_{\alpha \in \bar{\mathcal{B}}_i} x_{i\alpha}.$$

We divide the proof into 6 steps.

*Step 1:* By Proposition D.1(1), $h^*$ is $C^1$ and $\nabla h^*(y) = Q(y)$. Applying the same result playerwise to the reduced simplex $\Delta(\mathcal{B}_i)$ shows that the restricted conjugate

$$h_{\mathcal{B}_i}^*(y_i) := \max_{w_i \in \Delta(\mathcal{B}_i)} \{\langle y_i, w_i \rangle - h_i(w_i)\}$$

is $C^1$ with $\nabla h_{\mathcal{B}_i}^*(y_i) = Q_{\mathcal{B}_i}(y_i)$. Since $h = \sum_i h_i$ and $\mathcal{F}$ is a product face, we have $h_{\mathcal{B}}^*(y) = \sum_i h_{\mathcal{B}_i}^*(y_i)$ and thus

$$\nabla h_{\mathcal{B}}^*(y) = Q_{\mathcal{B}}(y).$$

---

[6]By abuse, we denote here $\Delta(\mathcal{B}_i)$ as the subset of mixed strategies of $w_i \in \mathcal{X}_i$ with $\text{supp}(w_i) \subseteq \mathcal{B}_i$.

Consequently,

$$\nabla F_{\mathcal{B}}(y) = \nabla h^*(y) - \nabla h^*_{\mathcal{B}}(y) = Q(y) - Q_{\mathcal{B}}(y) = x - x^{\mathcal{B}}. \tag{G.3}$$

Because $x, x^{\mathcal{B}} \in \mathcal{X}$ for all $y$, the gradient (G.3) is bounded on $\mathcal{Y}$, hence $F_{\mathcal{B}}$ is globally Lipschitz and $C^1$, verifying Definition G.1(1).

*Step 2:* By definition,

$$h^*(y) = \max_{x \in \mathcal{X}}\{\langle y, x \rangle - h(x)\}, \qquad h^*_{\mathcal{B}}(y) = \max_{w \in \mathcal{F}}\{\langle y, w \rangle - h(w)\},$$

so

$$F_{\mathcal{B}}(y) = h^*(y) - h^*_{\mathcal{B}}(y) = \min_{w \in \mathcal{F}}\big(h(w) + h^*(y) - \langle y, w \rangle\big) = \min_{w \in \mathcal{F}} F_h(w, y), \tag{G.4}$$

where $F_h(w, y) = \sum_{i \in \mathcal{N}} F_i(w_i, y_i)$ is the (playerwise) Fenchel coupling already defined in Appendix D. By Fenchel–Young, $F_i(\cdot, y_i) \geq 0$ for each $i$, hence $F_{\mathcal{B}}(y) \geq 0$ for all $y$. Moreover, $F_{\mathcal{B}}(y) = 0$ if and only if there exists $w \in \mathcal{F}$ with $F_h(w, y) = 0$, then each $F_i(w_i, y_i) = 0$, so $w_i = Q_i(y_i)$ for all $i$ by Proposition D.4(2), i.e. $Q(y) = w \in \mathcal{F}$. Conversely, if $Q(y) \in \mathcal{F}$, choosing $w = Q(y)$ in (G.4) gives $F_{\mathcal{B}}(y) \leq F_h(Q(y), y) = 0$, hence $F_{\mathcal{B}}(y) = 0$. Thus,

$$F_{\mathcal{B}}(y) = 0 \iff Q(y) \in \mathcal{F}. \tag{G.5}$$

*Step 3:* Let $K_{\min} := \min_{i \in \mathcal{N}} K_i$. Fix $y \in \mathcal{Y}$ and $x = Q(y)$. Using (G.4) and the quadratic lower bound of the Fenchel coupling from Proposition D.4(1),

$$F_{\mathcal{B}}(y) = \min_{w \in \mathcal{F}} \sum_{i \in \mathcal{N}} F_i(w_i, y_i) \geq \min_{w \in \mathcal{F}} \sum_{i \in \mathcal{N}} \frac{K_i}{2} \|x_i - w_i\|_2^2 \geq \frac{K_{\min}}{2} \operatorname{dist}(x, \mathcal{F})^2.$$

Therefore, if $F_{\mathcal{B}}(y^k) \to 0$ along some sequence $(y^k)$, then $\operatorname{dist}(Q(y^k), \mathcal{F}) \to 0$, i.e. $Q(y^k) \to \mathcal{F}$ because $\mathcal{F}$ is closed. Conversely, assume $\operatorname{dist}(Q(y^k), \mathcal{F}) \to 0$ and set $x^k := Q(y^k)$. Choose $w^k \in \mathcal{F}$ such that $\|x^k - w^k\|_2 \leq \operatorname{dist}(x^k, \mathcal{F}) + 1/k$, hence $\|x^k - w^k\|_2 \to 0$. By compactness of $\mathcal{F}$, pass to a subsequence (not relabeled) with $w^k \to w \in \mathcal{F}$, then also $x^k \to w$. For each $i$, $Q_i(y_i^k) = x_i^k \to w_i$, so Proposition D.4(2) yields $F_i(w_i, y_i^k) \to 0$. Summing over $i$, we get $F_h(w, y^k) \to 0$ and then, by (G.4),

$$0 \leq F_{\mathcal{B}}(y^k) = \min_{u \in \mathcal{F}} F_h(u, y^k) \leq F_h(w, y^k) \to 0.$$

Hence $F_{\mathcal{B}}(y^k) \to 0$. This proves Definition G.1(2).

*Step 4:* Fix $i \in \mathcal{N}$, $\beta_i \in \mathcal{B}_i$ and $\alpha_i \in \bar{\mathcal{B}}_i$. For every pure opponents' profile $\gamma_{-i} \in \prod_{j \neq i} \mathcal{B}_j$, closure of $\mathcal{B}$ under better replies means that $(\alpha_i, \gamma_{-i})$ cannot be a weakly improving deviation from $(\beta_i, \gamma_{-i})$, hence

$$u_i(\beta_i, \gamma_{-i}) > u_i(\alpha_i, \gamma_{-i}) \qquad \forall \gamma_{-i} \in \prod_{j \neq i} \mathcal{B}_j.$$

By multilinearity, the same strict inequality holds for all mixed opponents' profiles $w_{-i} \in \prod_{j \neq i} \Delta(\mathcal{B}_j)$, i.e., for all $w \in \mathcal{F}$:

$$v_{i\beta}(w) > v_{i\alpha}(w).$$

Consider now the gap $v_{i\beta}(w) - v_{i\alpha}(w)$, this is continuous on the compact set $\mathcal{F}$ and strictly positive everywhere, so[7]

$$\delta'_i := \min_{\beta_i \in \mathcal{B}_i, \ \alpha_i \in \bar{\mathcal{B}}_i} \min_{w \in \mathcal{F}} v_{i\beta}(w) - v_{i\alpha}(w) > 0$$

Set $\delta_i := \delta'_i/2$ for those $i$ with $\bar{\mathcal{B}}_i \neq \emptyset$. By uniform continuity of $v$ on compact $\mathcal{X}$, there exists $\varepsilon > 0$ such that whenever $\operatorname{dist}(x, \mathcal{F}) < \varepsilon$, we have, for all $i \in \mathcal{N}$ with $\bar{\mathcal{B}}_i \neq \emptyset$, $\beta_i \in \mathcal{B}_i$, $\alpha_i \in \bar{\mathcal{B}}_i$,

$$v_{i\beta}(x) - v_{i\alpha}(x) \geq \delta_i. \tag{G.6}$$

*Step 5:* By Proposition D.1(5), there exist $\lambda_i \in \mathbb{R}$ and multipliers $\nu_i \in \mathbb{R}_+^{\mathcal{A}_i}$ such that for all $\gamma_i \in \mathcal{A}_i$,

$$y_{i\gamma} = \theta'_i(x_{i\gamma}) + \lambda_i - \nu_{i\gamma}, \qquad x_{i\gamma} > 0 \implies \nu_{i\gamma} = 0.$$

---

[7]With the convention that the minimum over an empty index set is $+\infty$.

Likewise, applying the same KKT statement to the restricted maximization over $\Delta(\mathcal{B}_i)$, there exist $\lambda_i^{\mathcal{B}} \in \mathbb{R}$ and $v_i^{\mathcal{B}} \in \mathbb{R}_+^{\mathcal{B}_i}$ such that for all $\beta_i \in \mathcal{B}_i$,

$$y_{i\beta} = \theta_i'(x_{i\beta}^{\mathcal{B}}) + \lambda_i^{\mathcal{B}} - v_{i\beta}^{\mathcal{B}}, \qquad x_{i\beta}^{\mathcal{B}} > 0 \implies v_{i\beta}^{\mathcal{B}} = 0, \qquad x_{i\alpha}^{\mathcal{B}} = 0 \ (\alpha_i \in \mathcal{A}_i \setminus \mathcal{B}_i).$$

Define $\theta_i'(0+) := \lim_{p \downarrow 0} \theta_i'(p) \in [-\infty, \infty)$ and the truncated inverse

$$\varphi_i(z) := \begin{cases} 0, & z \leq \theta_i'(0+), \\ (\theta_i')^{-1}(z), & z > \theta_i'(0+), \end{cases}$$

which is nondecreasing. We claim that

$$x_{i\gamma} = \varphi_i(y_{i\gamma} - \lambda_i) \quad (\gamma_i \in \mathcal{A}_i), \qquad x_{i\beta}^{\mathcal{B}} = \varphi_i(y_{i\beta} - \lambda_i^{\mathcal{B}}) \quad (\beta_i \in \mathcal{B}_i). \tag{G.7}$$

Indeed, if $x_{i\gamma} > 0$, then $v_{i\gamma} = 0$ and $y_{i\gamma} - \lambda_i = \theta_i'(x_{i\gamma})$, so $x_{i\gamma} = (\theta_i')^{-1}(y_{i\gamma} - \lambda_i) = \varphi_i(y_{i\gamma} - \lambda_i)$. If $x_{i\gamma} = 0$, then $y_{i\gamma} - \lambda_i = \theta_i'(0+) - v_{i\gamma} \leq \theta_i'(0+)$ (interpreting $\theta_i'(x_{i\gamma})$ as $\theta_i'(0+)$ at $x_{i\gamma=0}$), hence $\varphi_i(y_{i\gamma} - \lambda_i) = 0 = x_{i\gamma}$. The same reasoning applies to $x^{\mathcal{B}}$ on $\mathcal{B}_i$. Summing (G.7) over $\mathcal{A}_i$ and $\mathcal{B}_i$ respectively gives

$$\sum_{\gamma \in \mathcal{A}_i} \varphi_i(y_{i\gamma} - \lambda_i) = 1, \qquad \sum_{\beta \in \mathcal{B}_i} \varphi_i(y_{i\beta} - \lambda_i^{\mathcal{B}}) = 1.$$

Since $\varphi_i \geq 0$,

$$\sum_{\beta \in \mathcal{B}_i} \varphi_i(y_{i\beta} - \lambda_i) \leq \sum_{\gamma \in \mathcal{A}_i} \varphi_i(y_{i\gamma} - \lambda_i) = 1.$$

Because $\varphi_i$ is nondecreasing, the function $\tau \mapsto \sum_{\beta \in \mathcal{B}_i} \varphi_i(y_{i\beta} - \tau)$ is nonincreasing, thus the equality $\sum_{\beta \in \mathcal{B}_i} \varphi_i(y_{i\beta} - \lambda_i^{\mathcal{B}}) = 1$ and the inequality at $\tau = \lambda_i$ imply $\lambda_i^{\mathcal{B}} \leq \lambda_i$.[8] Therefore, for every $\beta \in B$,

$$y_{i\beta} - \lambda_i \leq y_{i\beta} - \lambda_i^{\mathcal{B}} \implies x_{i\beta} = \varphi_i(y_{i\beta} - \lambda_i) \leq \varphi_i(y_{i\beta} - \lambda_i^{\mathcal{B}}) = x_{i\beta}^{\mathcal{B}}.$$

In summary,

$$x_{i\beta} \leq x_{i\beta}^{\mathcal{B}} \ (\beta \in \mathcal{B}_i), \qquad x_{i\alpha}^{\mathcal{B}} = 0 \ (\alpha \in \bar{\mathcal{B}}_i). \tag{G.8}$$

Consequently, we have the sign and mass identities

$$\Delta_{i\alpha} \geq 0 \ (\alpha \in \bar{\mathcal{B}}_i), \qquad \Delta_{i\beta} \leq 0 \ (\beta \in \mathcal{B}_i), \qquad m_i = \sum_{\alpha \in \bar{\mathcal{B}}_i} \Delta_{i\alpha} = -\sum_{\beta_i \in \mathcal{B}_i} \Delta_{i\beta}. \tag{G.9}$$

*Step 6:* Along continuous-time FTRL, $\dot{y} = v(Q(y)) = v(x)$, so by (G.3),

$$\dot{F}_{\mathcal{B}}(y) = \langle \nabla F_{\mathcal{B}}(y), \dot{y} \rangle = \langle x - x^{\mathcal{B}}, v(x) \rangle = \sum_{i \in \mathcal{N}} \langle \Delta_i, v_i(x) \rangle. \tag{G.10}$$

Assume now that $\text{dist}(x, \mathcal{F}) < \varepsilon$, with $\varepsilon$ from (G.6). Fix $i$ with $\bar{\mathcal{B}}_i \neq \varnothing$, then (G.6) yields $v_{i\alpha}(x) \leq \min_{\beta_i \in \mathcal{B}_i} v_{i\beta}(x) - \delta_i$ for all $\alpha_i \in \bar{\mathcal{B}}_i$ and $v_{i\beta}(x) \geq \min_{\beta_i \in \mathcal{B}_i} v_{i\beta}(x)$ for all $\beta_i \in \mathcal{B}_i$. Using (G.9),

$$\begin{aligned}
\langle \Delta_i, v_i(x) \rangle &= \sum_{\alpha_i \in \bar{\mathcal{B}}_i} v_{i\alpha}(x)\Delta_{i\alpha} + \sum_{\beta_i \in \mathcal{B}_i} v_{i\beta}(x)\Delta_{i\beta} \\
&\leq (\min_{\beta_i \in \mathcal{B}_i} v_{i\beta}(x) - \delta_i) \sum_{\alpha_i \in \bar{\mathcal{B}}_i} \Delta_{i\alpha} + \min_{\beta_i \in \mathcal{B}_i} v_{i\beta}(x) \sum_{\beta_i \in \mathcal{B}_i} \Delta_{i\beta} \\
&= (\min_{\beta_i \in \mathcal{B}_i} v_{i\beta}(x) - \delta_i)m_i - \min_{\beta_i \in \mathcal{B}_i} v_{i\beta}(x) \, m_i = -\delta_i m_i = -\delta_i \sum_{\alpha_i \in \bar{\mathcal{B}}_i} x_{i\alpha}.
\end{aligned}$$

Summing over $i$ in (G.10) gives, whenever $\text{dist}(x, \mathcal{F}) < \varepsilon$,

$$\dot{F}_{\mathcal{B}}(y) \leq -\sum_{i \in \mathcal{N}: \bar{\mathcal{B}}_i \neq \varnothing} \delta_i \sum_{\alpha \in \bar{\mathcal{B}}_i} x_{i\alpha}. \tag{G.11}$$

---

[8]Unless both sums happen to equal 1 at $\tau = \lambda_i$, in which case the maximizer of the unrestricted problem is in $\mathcal{F}$ anyway and one might take $\lambda_i = \lambda_i^{\mathcal{B}}$.

Now set $K_{\min} := \min_i K_i$ and define

$$\bar{E} := \frac{K_{\min}}{2} \varepsilon^2.$$

By Step 3, $F_{\mathcal{B}}(y) \leq \bar{E}$ implies $\operatorname{dist}(Q(y), \mathcal{F}) \leq \varepsilon$, so the drift bound (G.11) holds on the entire sublevel set $\{F_{\mathcal{B}} \leq \bar{E}\}$. Fix $0 < E^- < E^+ \leq \bar{E}$ and consider the domain $\mathcal{D} := \{y \in \mathcal{Y} : E^- < F_{\mathcal{B}}(y) < E^+\}$. Define the continuous outside-mass functional

$$M(y) := \sum_{i \in \mathcal{N}} \sum_{\alpha_i \in \bar{\mathcal{B}}_i} Q_{i\alpha}(y_i) = \sum_{i \in \mathcal{N}} m_i.$$

Let $\delta_{\min} := \min_{i: \bar{\mathcal{B}}_i \neq \varnothing} \delta_i > 0$ (if $\bar{\mathcal{B}}_i = \varnothing$ for all $i$, then $\mathcal{F} = \mathcal{X}$ and $F_{\mathcal{B}} = 0$ everywhere, so the dissipation requirement is vacuous). Then (G.11) gives, for all $y \in \mathcal{D}$,

$$\dot{F}_{\mathcal{B}}(y) \leq -\delta_{\min} M(y).$$

It remains to show that $M$ is bounded away from 0 on $\mathcal{D}$. Suppose not, then there exists a sequence $y^k \in \mathcal{D}$ with $M(y^k) \to 0$. Let $x^k := Q(y^k)$ and define $w^k \in \mathcal{F}$ by renormalizing $x_i^k$ on $\mathcal{B}_i$: if $m_i^k := \sum_{\alpha_i \in \bar{\mathcal{B}}_i} x_{i\alpha}^k$ and $1 - m_i^k > 0$, set

$$w_{i\beta}^k := \frac{x_{i\beta}^k}{1 - m_i^k} \ (\beta_i \in \mathcal{B}_i), \qquad w_{i\alpha}^k := 0 \ (\alpha_i \in \bar{\mathcal{B}}_i).$$

Then $w^k \in \mathcal{F}$ and, for each $i$,

$$\|x_i^k - w_i^k\|_2^2 = \sum_{\beta_i \in \mathcal{B}_i} \left(x_{i\beta}^k - \frac{x_{i\beta}^k}{1 - m_i^k}\right)^2 + \sum_{\alpha_i \in \bar{\mathcal{B}}_i} (x_{i\alpha}^k)^2 = \left(\frac{m_i^k}{1 - m_i^k}\right)^2 \sum_{\beta_i \in \mathcal{B}_i} (x_{i\beta}^k)^2 + \sum_{\alpha_i \in \bar{\mathcal{B}}_i} (x_{i\alpha}^k)^2.$$

Using $\sum_{\beta_i \in \mathcal{B}_i} (x_{i\beta}^k)^2 \leq (\sum_{\beta_i \in \mathcal{B}_i} x_{i\beta}^k)^2 = (1 - m_i^k)^2$ and $\sum_{\alpha_i \in \bar{\mathcal{B}}_i} (x_{i\alpha}^k)^2 \leq (\sum_{\alpha_i \in \bar{\mathcal{B}}_i} x_{i\alpha}^k)^2 = (m_i^k)^2$, we obtain

$$\|x_i^k - w_i^k\|_2^2 \leq (m_i^k)^2 + (m_i^k)^2 = 2(m_i^k)^2, \qquad \text{so} \qquad \|x_i^k - w_i^k\|_2 \leq \sqrt{2}\, m_i^k.$$

Therefore,

$$\operatorname{dist}(x^k, \mathcal{F}) \leq \|x^k - w^k\|_2 \leq \sqrt{\sum_{i \in \mathcal{N}} \|x_i^k - w_i^k\|_2^2} \leq \sqrt{2} \sum_{i \in \mathcal{N}} m_i^k = \sqrt{2}\, M(y^k) \to 0.$$

By Step 3, this implies $F_{\mathcal{B}}(y^k) \to 0$, contradicting $F_{\mathcal{B}}(y^k) > E^-$ for all $k$. Hence $\inf_{y \in \mathcal{D}} M(y) =: \kappa(E^-, E^+) > 0$, and thus

$$\sup_{y \in \mathcal{D}} \dot{F}_{\mathcal{B}}(y) \leq -\delta_{\min} \kappa(E^-, E^+) < 0,$$

which is exactly Definition G.1(3).

All three items of Definition G.1 are verified, so $F_{\mathcal{B}}$ is a local energy function for $\mathcal{F} = \operatorname{span}(\mathcal{B})$. $\qquad \square$

We may now harvest what we sowed:

***Proof of Theorem 3.*** Combine Theorem G.2 with Lemma G.3. $\qquad \square$

***Proof of Corollary 1.*** Combine Theorem 1 with Theorem 3. $\qquad \square$

***Proof of Corollary 2.*** We shall construct a basin of attraction on all of $\mathcal{X}$ by gluing together all the facewise basins of (FTRL) restricted to each face.

Let $\mathcal{F} = \operatorname{span}(\mathcal{B}) = \prod_{i \in \mathcal{N}} \Delta(\mathcal{B}_i)$. First, assume that $\mathcal{B}$ is closed under better replies. We show that $\mathcal{F}$ is an attractor for the strategy flow induced by (SD), i.e., $\mathcal{F}$ is invariant and asymptotically stable. Let $\mathcal{F}' = \operatorname{span}(\mathcal{B}') = \prod_{i \in \mathcal{N}} \Delta(\mathcal{B}_i')$ be any face of $\mathcal{X}$ such that $\mathcal{F} \cap \mathcal{F}' \neq \varnothing$. Then $\mathcal{B}_i \cap \mathcal{B}_i' \neq \varnothing$ for all $i$, and

$$\mathcal{F} \cap \mathcal{F}' = \prod_{i \in \mathcal{N}} \Delta(\mathcal{B}_i \cap \mathcal{B}_i') = \operatorname{span}(\mathcal{B}|_{\mathcal{B}'}), \qquad \mathcal{B}|_{\mathcal{B}'} := \prod_{i \in \mathcal{N}} (\mathcal{B}_i \cap \mathcal{B}_i'). \tag{G.12}$$

Indeed, $x \in \mathcal{F} \cap \mathcal{F}'$ if and only if for every $i$, $x_i \in \Delta(\mathcal{B}_i) \cap \Delta(\mathcal{B}'_i) = \Delta(\mathcal{B}_i \cap \mathcal{B}'_i)$. Moreover, $\mathcal{B}|_{\mathcal{B}'}$ is closed under better replies in the preference graph restricted to the vertex set $\mathcal{B}'$: if $\alpha \in \mathcal{B}|_{\mathcal{B}'}$ and $\alpha \to \beta$ with $\beta \in \mathcal{B}'$, then $\alpha \in \mathcal{B}$ and the arc $\alpha \to \beta$ is also an arc of the full preference graph, hence $\beta \in \mathcal{B}$ by closedness of $\mathcal{B}$, so $\beta \in \mathcal{B} \cap \mathcal{B}' = \mathcal{B}|_{\mathcal{B}'}$.

Fix such a face $\mathcal{F}' = \mathrm{span}(\mathcal{B}')$ and consider the facewise FTRL dynamics (FTRL-$\mathcal{B}'$) on $\mathcal{F}'$. Applying Theorem 3 to the restricted game on action sets $\mathcal{B}'$ yields that

$$\mathcal{F} \cap \mathcal{F}' = \mathrm{span}(\mathcal{B}|_{\mathcal{B}'})$$

is asymptotically stable under (FTRL-$\mathcal{B}'$). Under Assumption 1, Lemma D.2 gives $\mathrm{Im}\, Q_{\mathcal{B}'} = \mathcal{F}'^{\circ}$, and by Remark E.9 the restriction of the strategy flow to $\mathcal{F}'$ coincides with the induced strategy orbit of (FTRL-$\mathcal{B}'$) for initial conditions in $\mathcal{F}'^{\circ}$. Hence, for every $\varepsilon > 0$ and every such $\mathcal{F}'$, there exists a neighborhood $U^{\varepsilon}_{\mathcal{F}'} \subseteq \mathcal{F}'$ of $\mathcal{F} \cap \mathcal{F}'$ in the relative topology of $\mathcal{F}'$ such that every strategy flow trajectory with initial condition $x_0 \in U^{\varepsilon}_{\mathcal{F}'} \cap \mathcal{F}'^{\circ}$ satisfies

$$\mathrm{dist}(X_t(x_0), \mathcal{F} \cap \mathcal{F}') < \varepsilon \ \forall t \geq 0, \qquad \lim_{t \to +\infty} \mathrm{dist}(X_t(x_0), \mathcal{F} \cap \mathcal{F}') = 0. \tag{G.13}$$

Let now $\mathscr{F}$ be the (finite) set of faces $\mathcal{F}'$ of $\mathcal{X}$ such that $\mathcal{F} \cap \mathcal{F}' \neq \emptyset$. Fix $\varepsilon > 0$. For each $\mathcal{F}' \in \mathscr{F}$, let $U^{\varepsilon}_{\mathcal{F}'}$ be as in (G.13). Fix $x \in \mathcal{F}$ and define $\mathscr{F}(x) := \{\mathcal{F}' \in \mathscr{F} : x \in \mathcal{F}'\}$. Since $\mathcal{X}$ has finitely many faces, set

$$r_x := \frac{1}{2} \min\{\mathrm{dist}(x, \mathcal{F}'') : \ \mathcal{F}'' \text{ a face of } \mathcal{X}, \ x \notin \mathcal{F}''\} > 0,$$

so that $B_{r_x}(x) \cap \mathcal{X}$ meets only faces $\mathcal{F}'$ with $x \in \mathcal{F}'$ (equivalently, only faces in $\mathscr{F}(x)$). Next, for each $\mathcal{F}' \in \mathscr{F}(x)$, since $U^{\varepsilon}_{\mathcal{F}'}$ is a neighborhood of $\mathcal{F} \cap \mathcal{F}'$ in $\mathcal{F}'$ and $x \in \mathcal{F} \cap \mathcal{F}'$, there exists $\rho^x_{\mathcal{F}'} > 0$ such that

$$B_{\rho^x_{\mathcal{F}'}}(x) \cap \mathcal{F}' \subseteq U^{\varepsilon}_{\mathcal{F}'}.$$

Define

$$\rho_x := \min\left(r_x, \min_{\mathcal{F}' \in \mathscr{F}(x)} \rho^x_{\mathcal{F}'}\right) > 0, \qquad W_x := B_{\rho_x}(x) \cap \mathcal{X},$$

so $W_x$ is open in $\mathcal{X}$. Let $z \in W_x$ and let $\mathcal{F}_z$ denote the (unique) minimal face of $\mathcal{X}$ containing $z$. By construction of $r_x$, every face of $\mathcal{X}$ intersecting $W_x$ contains $x$, so in particular $x \in \mathcal{F}_z$, hence $\mathcal{F}_z \in \mathscr{F}(x) \subseteq \mathscr{F}$. Moreover,

$$z \in B_{\rho_x}(x) \cap \mathcal{F}_z \subseteq U^{\varepsilon}_{\mathcal{F}_z}$$

by construction of $\rho_x$. By definition of minimal face, $z \in \mathcal{F}_z^{\circ}$ and by Proposition E.7, faces are invariant for the strategy flow, so $X_t(z) \in \mathcal{F}_z$ for all $t \geq 0$. Applying (G.13) with $\mathcal{F}' = \mathcal{F}_z$ yields

$$\mathrm{dist}(X_t(z), \mathcal{F} \cap \mathcal{F}_z) < \varepsilon \ \forall t \geq 0, \qquad \mathrm{dist}(X_t(z), \mathcal{F} \cap \mathcal{F}_z) \to 0.$$

Since $\mathcal{F} \cap \mathcal{F}_z \subseteq \mathcal{F}$, we have $\mathrm{dist}(X_t(z), \mathcal{F}) \leq \mathrm{dist}(X_t(z), \mathcal{F} \cap \mathcal{F}_z)$, hence

$$\mathrm{dist}(X_t(z), \mathcal{F}) < \varepsilon \ \forall t \geq 0, \qquad \mathrm{dist}(X_t(z), \mathcal{F}) \to 0.$$

Because $\mathcal{F}$ is compact, the open cover $\{W_x\}_{x \in \mathcal{F}}$ admits a finite subcover $\mathcal{F} \subseteq \bigcup_{k=1}^m W_{x^k}$. Set

$$W^{\varepsilon} := \bigcup_{k=1}^m W_{x^k},$$

an open neighborhood of $\mathcal{F}$ in $\mathcal{X}$. Then for all $z \in W^{\varepsilon}$,

$$\mathrm{dist}(X_t(z), \mathcal{F}) < \varepsilon \ \forall t \geq 0, \qquad \mathrm{dist}(X_t(z), \mathcal{F}) \to 0.$$

Since $\varepsilon > 0$ was arbitrary, this proves that $\mathcal{F}$ is stable and attracting for the strategy flow. Moreover, $\mathcal{F}$ is invariant by Proposition E.7. Hence $\mathcal{F}$ is an invariant asymptotically stable set, i.e., an attractor.

Conversely, if $\mathcal{F}$ is an attractor for the strategy flow, then in particular it is attracting. By Proposition 1, $\mathcal{F} \cap \mathcal{A}$ is closed under better replies. Since $\mathcal{F} = \mathrm{span}(\mathcal{B})$, we have $\mathcal{F} \cap \mathcal{A} = \mathcal{B}$, so $\mathcal{B}$ is closed under better replies. $\qquad \square$

***Proof of Corollary 3.*** First, we show strict Nash equilibria are attractors. Let $\alpha^* \in \mathcal{A}$ be such an equilibrium and identify it with the corresponding vertex of $\mathcal{X}$. Then for every player $i$ and every $\alpha_i \neq \alpha_i^*$ we have $u_i(\alpha_i, \alpha_{-i}^*) < u_i(\alpha^*)$, hence there is no arc $\alpha^* \to \beta$ in the preference graph with $\beta \neq \alpha^*$. Therefore $\{\alpha^*\}$ is a club subgame. By Corollary 2, $\mathrm{span}(\{\alpha^*\}) = \{\alpha^*\}$ is an attractor for the strategy flow induced by (SD). Since it is a singleton, it contains no proper (nonempty) subset, hence it is minimal among attractors.

Conversely, let $M \subseteq \mathcal{X}$ be a minimal attractor for the strategy flow. In particular $M$ is asymptotically stable. By Corollary F.3, there exists a nonempty set of pure profiles $\mathcal{H} \subseteq M \cap \mathcal{A}$ which is closed under better replies. Fix $\alpha^0 \in \mathcal{H}$. Since the game is weakly acyclic, there exists a finite better-reply path

$$\alpha^0 \to \alpha^1 \to \cdots \to \alpha^k$$

ending at a pure Nash equilibrium $\alpha^k$. Because $\mathcal{H}$ is closed under better replies and $\alpha^0 \in \mathcal{H}$, an immediate induction gives $\alpha^j \in \mathcal{H}$ for all $j$, hence $\alpha^k \in \mathcal{H} \subseteq M$. Moreover, since the game has no ties, $\alpha^k$ must be a strict Nash equilibrium. Now, by the first implication, strictness implies that $\{\alpha^k\}$ is an attractor. Since $\{\alpha^k\} \subseteq M$ and $M$ is minimal among attractors, we conclude $M = \{\alpha^k\}$, i.e. $M$ is a strict Nash equilibrium. $\qquad\square$

***Proof of Remark 2.*** Note that, by definition, a finite game is an *ordinal potential game* if there exists $\Phi : \mathcal{A} \to \mathbb{R}$ such that for every player $i$ and every $i$-comparable $\alpha, \alpha' \in \mathcal{A}$,

$$u_i(\alpha') > u_i(\alpha) \quad \Longleftrightarrow \quad \Phi(\alpha') > \Phi(\alpha).$$

Now, if $\Phi$ is an ordinal potential, every arc $\alpha \to \alpha'$ yields $u_i(\alpha') > u_i(\alpha)$ (no ties), hence $\Phi(\alpha') > \Phi(\alpha)$. Thus $\Phi$ strictly increases along directed edges, so no directed cycle exists. Conversely, if the preference graph is acyclic, fix a topological ordering $\alpha^1, \ldots, \alpha^{|\mathcal{A}|}$ and set $\Phi(\alpha^k) = k$. Then $\alpha \to \alpha'$ implies $\Phi(\alpha') > \Phi(\alpha)$. For $i$-comparable $\alpha, \alpha'$, the unique arc between them points toward the larger payoff. $\qquad\square$

## G.3. When preference are *not* enough

We start with the counterexample.

***Proof of Proposition 2.*** Players have two actions each:

$$\mathcal{A}_1 = \{F, B\} \quad (\text{Front/Back}), \qquad \mathcal{A}_2 = \{L, R\} \quad (\text{Left/Right}), \qquad \mathcal{A}_3 = \{T, B\} \quad (\text{Top/Bottom}).$$

We abbreviate a pure profile $\alpha = (\alpha_1, \alpha_2, \alpha_3) \in \mathcal{A}$ by the three-letter word $\alpha_1\alpha_2\alpha_3$, e.g. FLT = (F, L, T).

Payoffs $u(\alpha) = (u_1, u_2, u_3)$ are:

|  |  | L | R |
|---|---|---|---|
| **Front:** | T | $(1, 0, 0)$ | $(0, 1, 0)$ |
|  | B | $(2, 0, 10)$ | $(0, 2, 0)$ |

|  |  | L | R |
|---|---|---|---|
| **Back:** | T | $(0, 1, 1)$ | $(1, 0, 1)$ |
|  | B | $(0, 0, 0)$ | $(1, 1, 0)$ |

Let $\mathcal{H} := \mathcal{A} \setminus \{BLB\}$ and $\mathcal{S} := \mathrm{span}(\mathcal{H}) \subseteq \mathcal{X}$. In the preference graph of this game, BLB has no incoming arc and $\mathcal{H}$ is the unique club set (see Figure 14). In the cube $\mathcal{X} = \Delta(\mathcal{A}_1) \times \Delta(\mathcal{A}_2) \times \Delta(\mathcal{A}_3)$, write the coordinates

$$p := x_{1F} \in [0, 1], \qquad q := x_{2R} \in [0, 1], \qquad r := x_{3T} \in [0, 1].$$

Then BLB corresponds to $(p, q, r) = (0, 0, 0)$ and one checks

$$\mathcal{S} = \{p = 1\} \cup \{q = 1\} \cup \{r = 1\}, \tag{G.14}$$

i.e. the union of the three square faces not containing BLB.

Let $\mathcal{F}_T := \{r = 1\}$, i.e. player 3 plays T purely. Restricting to $\mathcal{F}_T$ gives the 2-player game between players $(1, 2)$:

|  | L | R |
|---|---|---|
| B | $(0, 1)$ | $(1, 0)$ |
| F | $(1, 0)$ | $(0, 1)$ |

hence the unique mixed equilibrium is

$$p^* = \tfrac{1}{2}, \qquad q^* = \tfrac{1}{2}.$$

Define

$$x^* := (p^*, q^*, 1) = \left(\tfrac{1}{2}, \tfrac{1}{2}, 1\right) \in \mathcal{F}_{\mathrm{T}}.$$

At $x^*$, player 3 has a strict profitable deviation to B:

$$v_{3\mathrm{T}}(x^*) = 1 - p^* = \tfrac{1}{2}, \qquad v_{3\mathrm{B}}(x^*) = 10\, p^*(1 - q^*) = 10 \cdot \tfrac{1}{4} = \tfrac{5}{2},$$

so $v_{3\mathrm{B}}(x^*) - v_{3\mathrm{T}}(x^*) = 2 > 0$.

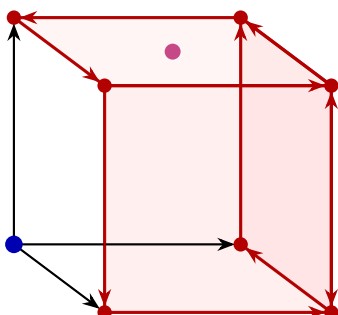

**Figure 14.** Cube $\mathcal{X}$ with coordinates $(p, q, r) = (x_{1\mathrm{F}}, x_{2\mathrm{R}}, x_{3\mathrm{T}})$. The red vertices represent the unique club set, and the red shaded faces are its span. The blue vertex is a *source vertex*, meaning it has no incoming edge. In particular it is a repellor of the dynamics. The purple point $x^* = (\tfrac{1}{2}, \tfrac{1}{2}, 1)$ is a Nash equilibrium of the game restricted to the face $\mathcal{F}_{\mathrm{T}}$, hence it is a stationary points of the dynamics. Nonetheless, Player 3 has a deviation to the bottom in the neighborhood of that point, hence the dynamics are locally tangent to $\mathcal{F}_{\mathrm{T}}$ near $x^*$. This is what shall drive instability.

We consider now the replicator dynamic on $\mathcal{X}$ (which correspond to entropic (FTRL)):

$$\dot{x}_{i\alpha} = x_{i\alpha}\big(v_{i\alpha}(x) - u_i(x)\big).$$

In the coordinates $(p, q, r)$ this becomes

$$\dot{p} = p(1 - p)\big(v_{1\mathrm{F}}(x) - v_{1\mathrm{B}}(x)\big), \quad \dot{q} = q(1 - q)\big(v_{2\mathrm{R}}(x) - v_{2\mathrm{L}}(x)\big), \quad \dot{r} = r(1 - r)\big(v_{3\mathrm{T}}(x) - v_{3\mathrm{B}}(x)\big). \tag{G.15}$$

Write $s := 1 - r$ and centered variables

$$a := p - \tfrac{1}{2}, \qquad b := q - \tfrac{1}{2}, \qquad m := |a| + |b|.$$

We show that, along (RD), the variables $(a, b, s)$ satisfy

$$\dot{a} = \left(\tfrac{1}{4} - a^2\right)\left(\tfrac{s}{2} - (2 + s)b\right), \tag{G.16}$$

$$\dot{b} = \left(\tfrac{1}{4} - b^2\right)\left(\tfrac{3}{2}s + (2 - s)a\right), \tag{G.17}$$

$$\dot{s} = (1 - s)s\left(2 + 6a - 5b - 10ab\right). \tag{G.18}$$

Indeed, direct calculation from the payoff table gives

$$v_{1\mathrm{F}}(x) = (1 - q)(2 - r), \qquad v_{1\mathrm{B}}(x) = q, \qquad v_{2\mathrm{R}}(x) = 1 + p - r, \qquad v_{2\mathrm{L}}(x) = r(1 - p),$$

$$v_{3\mathrm{T}}(x) = 1 - p, \qquad v_{3\mathrm{B}}(x) = 10p(1 - q).$$

Substituting in (G.15) and using $p(1 - p) = \tfrac{1}{4} - a^2$, $q(1 - q) = \tfrac{1}{4} - b^2$, $r(1 - r) = (1 - s)s$ yields (G.16)–(G.18).

Now, on the face $\mathcal{F}_{\mathrm{T}}$, one has $\dot{r} = 0$, so the vector field is tangent to $\mathcal{F}_{\mathrm{T}}$. At $x^* = (\tfrac{1}{2}, \tfrac{1}{2}, 1)$, players 1 and 2 are indifferent between their actions on $\mathcal{F}_{\mathrm{T}}$, hence $\dot{p} = \dot{q} = 0$. However, for $r < 1$ close to 1 and $(p, q)$ near $(\tfrac{1}{2}, \tfrac{1}{2})$, player 3 strictly prefers B, so $s = 1 - r$ increases. We shall quantify this transverse drift and show how it destroys stability of the span $\mathcal{S}$.

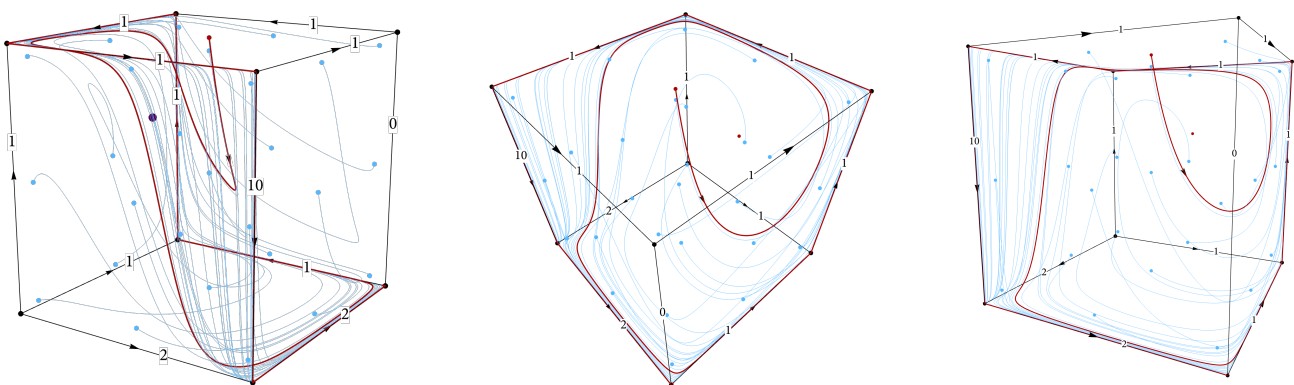

**Figure 15.** Three perspectives of the replicator dynamics for this counterexample game.

For $x = (p, q, r)$, the Euclidean distance (in $\mathcal{X}$ which is a subset of $\mathbb{R}^2 \times \mathbb{R}^2 \times \mathbb{R}^2$) to the face $\{p = 1\}$ (resp. $\{q = 1\}$, $\{r = 1\}$) equals $\sqrt{2}(1 - p)$ (resp. $\sqrt{2}(1 - q)$, $\sqrt{2}(1 - r)$), so with $s := 1 - r$,

$$\text{dist}(x, \mathcal{S}) = \min\{\, 1 - p,\; 1 - q,\; s \,\}. \tag{G.19}$$

Fix $\eta$ and let

$$U := \left\{ x \in \mathcal{X} : \text{dist}(x, \mathcal{S}) < \frac{\sqrt{2}}{2}\eta \right\}.$$

Let $V$ be any neighborhood of $\mathcal{S}$. Since $x^* = (\frac{1}{2}, \frac{1}{2}, 1) \in \mathcal{S}$, there exists $\varepsilon_0 > 0$ such that for $0 < \varepsilon < \varepsilon_0$,

$$x^\varepsilon(0) := \left( \tfrac{1}{2}, \tfrac{1}{2}, 1 - \varepsilon \right) \in V.$$

Fix $\varepsilon \in (0, \min\{\varepsilon_0, \eta/2\})$ and let $x^\varepsilon(t)$ be the corresponding replicator trajectory. Writing $a = p - \frac{1}{2}$, $b = q - \frac{1}{2}$, $m := |a| + |b|$, we have $a(0) = b(0) = 0$ and $s(0) = \varepsilon$, hence by (G.19), $x^\varepsilon(0) \in U$. The idea is that near $(a, b) = (0, 0)$ player 3 has a *uniform* incentive to increase $s$ (move away from the face $r = 1$), while the drift of $(a, b)$ is of order $s$, consequently $s$ reaches size $\eta$ before $(p, q)$ can move close to $p = 1$ or $q = 1$, so the minimum in (G.19) becomes $s$ and the orbit leaves $U$. Formally, define the cone region

$$\mathcal{R} := \{(a, b, s) : 0 < s \le \eta, \; m \le s\}.$$

Since the vector field in (G.16)–(G.18) is polynomial, solutions are $C^1$ in time. Given a function $f$, denote the upper right derivative by

$$f'_+(t) := \limsup_{h \downarrow 0} \frac{f(t + h) - f(t)}{h},$$

we have for $m(t) = |a(t)| + |b(t)|$:

$$m'_+(t) \le |\dot{a}(t)| + |\dot{b}(t)|. \tag{G.20}$$

On $\mathcal{R}$ we have $|a| \le m \le s$, $|b| \le s$, and by $|ab| \le m^2/4 \le s^2/4$. Hence

$$2 + 6a - 5b - 10ab \ge 2 - 6|a| - 5|b| - 10|ab| \ge 2 - 11s - \tfrac{5}{2}s^2.$$

With $\eta$ sufficiently small, the right-hand side is $\ge 3/2$ for all $s \in (0, \eta]$, so (G.18) yields

$$\dot{s} \ge \frac{3}{2}(1 - \eta)s \qquad \text{on } \mathcal{R}. \tag{G.21}$$

In particular, $s$ is strictly increasing as long as the trajectory remains in $\mathcal{R}$. Using $\frac{1}{4} - a^2 \le \frac{1}{4}$ and $\frac{1}{4} - b^2 \le \frac{1}{4}$ in (G.16)–(G.17), and assuming $s \le \eta$, we obtain

$$|\dot{a}| \le \frac{1}{4}\left(\frac{s}{2} + (2 + s)|b|\right) \le \frac{s}{8} + \left(\frac{1}{2} + \frac{\eta}{4}\right)|b|, \qquad |\dot{b}| \le \frac{1}{4}\left(\frac{3}{2}s + (2 - s)|a|\right) \le \frac{3s}{8} + \frac{1}{2}|a|.$$

Summing and using $|a| + |b| = m$ gives

$$|\dot{a}| + |\dot{b}| \leq \frac{s}{2} + \left(\frac{1}{2} + \frac{\eta}{4}\right)m,$$

and with (G.20):

$$m'_+ \leq \frac{s}{2} + \left(\frac{1}{2} + \frac{\eta}{4}\right)m \qquad \text{whenever } s \leq \eta. \tag{G.22}$$

Let $w := m - s$. Since $s$ is $C^1$, $w'_+(t) \leq m'_+(t) - \dot{s}(t)$. On $\mathcal{R}$, combining (G.22) with (G.21) yields

$$w'_+ \leq \frac{s}{2} + \left(\frac{1}{2} + \frac{\eta}{4}\right)m - \frac{3}{2}(1 - \eta)s.$$

If at some time $t$ we have $w(t) = 0$ and $s(t) \leq \eta$, then $m(t) = s(t)$ and therefore

$$w'_+(t) \leq \left(\frac{1}{2} + \frac{1}{2} + \frac{\eta}{4} - \frac{3}{2}(1 - \eta)\right)s(t) = \left(-\frac{1}{2} + \frac{7}{4}\eta\right)s(t) < 0,$$

for $\eta$ sufficiently small. Hence $w$ cannot cross from $< 0$ to $> 0$ while $s \leq \eta$. Because $w(0) = m(0) - s(0) = -\varepsilon < 0$, we conclude

$$m(t) \leq s(t) \qquad \text{for all } t \text{ such that } s(t) \leq \eta. \tag{G.23}$$

Let $t_\eta := \inf\{t \geq 0 : s(t) = \eta\}$. By (G.21) and $s(0) = \varepsilon < \eta$, we have $t_\eta < \infty$. At time $t_\eta$, (G.23) gives $m(t_\eta) \leq \eta$, hence

$$1 - p(t_\eta) = \tfrac{1}{2} - a(t_\eta) \geq \tfrac{1}{2} - |a(t_\eta)| \geq \tfrac{1}{2} - m(t_\eta) \geq \tfrac{1}{2} - \eta > \eta,$$

and similarly $1 - q(t_\eta) > \eta$. Since $s(t_\eta) = \eta$, the minimum in (G.19) equals $\eta$, so

$$\text{dist}(x(t_\eta), \mathcal{S}) = \sqrt{2}\eta > \frac{\sqrt{2}}{2}\eta,$$

i.e. $x(t_\eta) \notin U$ for some sufficiently small $\eta$. $\qquad\square$

*Remark* G.4. Note that in the game considered in the above proof, the vertex BLB is a repellor. Indeed, it has no incoming edge (see the blue vertex in Figure 14), so it is a strict Nash equilibrium of the negated game, hence an attractor for that game, and therefore (see Remark E.8) a repellor for our game. Consider then the dual attractor of BLB (see Appendix C.1 for definition of dual repellor/attractor). This attractor must contain the span of the unique club set (combining Theorem 2 and Corollary F.3), but it cannot coincide with that span since the latter is not stable by the previous proposition. Moreover, any larger span containing it would necessarily contain BLB, hence the dual attractor cannot be such a span either. Consequently, this attractor is not a span of pure profiles.

*Remark* G.5. Biggar and Papadimitriou [8] introduce the notion of a *local source*, defined as follows. Let $x \in \mathcal{X}$, let $\mathcal{H} \subseteq \mathcal{A}$ be a sink equilibrium (recall: a minimal nonempty club set), and set $\mathcal{S} = \text{span}(\mathcal{H})$. Let $\mathcal{B}$ be a subgame and let $\mathcal{F} = \text{span}(\mathcal{B})$ be its span. Then $x$ is a local source of $\mathcal{H}$ in $\mathcal{F}$ if: (1) $x \in \mathcal{S} \cap \mathcal{F}$, (2) $\mathcal{F} \nsubseteq \mathcal{S}$, and (3) $x$ is a quasi-strict Nash equilibrium of the negated game $-u$ restricted to $\mathcal{F}$.

They show that if a sink equilibrium admits a local source, then its span cannot be an attractor for the replicator dynamics, and they conjecture that the converse holds. However, in our example, the only face satisfying both (1) and (2) is the full cube $\mathcal{X}$, and $\mathcal{S}$ contains no strict Nash equilibrium of the negated game. Hence $\mathcal{H}$ has no local sources, yet its span is unstable, refuting their conjecture.

Nevertheless, in a loose sense, the point $x^*$ (and essentially all of the face $\mathcal{F}_\text{T}$) still behaves like a "local source", except that the leakage is onto the interior rather than onto the span of another subgame. This motivates the term "leakless" for the condition we introduce to preclude this type of instability.

## G.4. Restoring dynamic stability

We first prove that leaklessness implies closedness under better replies:

***Proof of Proposition 3.*** We prove the elementary identity from which both claims follow directly. Let $\beta \in \mathcal{H}$ and let $\alpha \in \mathcal{A} \setminus \mathcal{H}$ be $i$-comparable with $\beta$, so $\alpha_{-i} = \beta_{-i}$ and $\alpha_i \neq \beta_i$. Then, for every $j \neq i$, we have $\alpha_j = \beta_j$, and hence

$$u_j(\alpha_j, \beta_{-j}) = u_j(\beta_j, \beta_{-j}) = u_j(\beta).$$

Therefore

$$l_\alpha(\beta) = \sum_{j \in \mathcal{N}} \left( u_j(\alpha_j, \beta_{-j}) - u_j(\beta) \right)$$
$$= u_i(\alpha_i, \beta_{-i}) - u_i(\beta)$$
$$= u_i(\alpha) - u_i(\beta),$$

where the last equality uses $\alpha_{-i} = \beta_{-i}$. $\qquad \square$

We have also an upgrade to mixed profiles:

**Lemma G.6.** *Let $\mathcal{H} \subseteq \mathcal{A}$. Then $\mathcal{H}$ is leakless (resp. strictly leakless) if and only if for every $\alpha \notin \mathcal{H}$, and for every face $\mathcal{F} \subseteq \operatorname{span}(\mathcal{H})$, we have $\sup_{x \in \mathcal{F}} l_\alpha(x) \leq 0$ (resp. $\sup_{x \in \mathcal{F}} l_\alpha(x) < 0$).*

*Proof.* The forward direction is trivial: if $\alpha \notin \mathcal{H}$ and $\beta \in \mathcal{H}$, since the singleton $\{\beta\}$ is a (zero-dimensional) face of $\mathcal{X}$ and $\{\beta\} \subseteq \operatorname{span}(\mathcal{H})$, leaklessness yields

$$l_\alpha(\beta) = \sup_{x \in \{\beta\}} l_\alpha(x) \leq 0.$$

(respectively $< 0$ for strict leaklessness).

For the other direction, fix $\alpha \notin \mathcal{H}$ and a face $\mathcal{F} \subseteq \operatorname{span}(\mathcal{H})$ and $\mathcal{F} = \operatorname{span}(\mathcal{B})$ for some subgame $\mathcal{B}$, so $\mathcal{B} \subseteq \mathcal{H}$. We have

$$l_\alpha(x) = \sum_{i \in \mathcal{N}} \left( v_{i\alpha}(x) - u_i(x) \right),$$

so $l_\alpha$ is multilinear. Next, we claim that for any multilinear $f$ on $\mathcal{F}$,

$$\sup_{x \in \mathcal{F}} f(x) = \max_{\beta \in \mathcal{B}} f(\beta). \tag{G.24}$$

This is classic, to see why it's true, simply fix $x_{-1}$ and consider $x_1 \mapsto f(x_1, x_{-1})$, which is affine on $\Delta(\mathcal{B}_1)$, thus it attains its maximum at a vertex $\beta_1$ with $\beta_1 \in \mathcal{B}_1$, giving $f(x) \leq f(\beta_1, x_{-1})$. Iterating this argument over $i = 2, \ldots, n$ yields some $\beta = (\beta_1, \ldots, \beta_n) \in \mathcal{B}$ with $f(x) \leq f(\beta)$ for all $x \in \mathcal{F}$. Applying (G.24) to $f = l_\alpha$ and using $\mathcal{B} \subseteq \mathcal{H}$, we get by hypothesis

$$\sup_{x \in \mathcal{F}} l_\alpha(x) = \max_{\beta \in \mathcal{B}} l_\alpha(\beta) \leq 0 \quad (\text{resp.} < 0).$$

Since $\alpha \notin \mathcal{H}$ and $\mathcal{F} \subseteq \operatorname{span}(\mathcal{H})$ were arbitrary, $\mathcal{H}$ is leakless (resp. strictly leakless). $\qquad \square$

To proceed, we shall prove the following lemma.

**Lemma G.7.** *Let $\mathcal{H} \subseteq \mathcal{A}$ be strictly leakless and assume $h$ is steep. Define the function*

$$\bar{F}_{\mathcal{H}}(y) \coloneqq \sum_{\alpha \notin \mathcal{H}} e^{-F_h(\alpha, y)} \in [0, \infty), \; y \in \mathcal{Y},$$

*with the convention $e^{-\infty} = 0$. Then $\bar{F}_{\mathcal{H}}$ is a local energy function for $\operatorname{span}(\mathcal{H})$.*

*Proof.* Set $\mathcal{S} \coloneqq \operatorname{span}(\mathcal{H})$ and define $E : \mathcal{Y} \to [0, \infty)$ by

$$E(y) \coloneqq \bar{F}_{\mathcal{H}}(y) = \sum_{\alpha \notin \mathcal{H}} e^{-F_h(\alpha, y)}.$$

If $\mathcal{A} \setminus \mathcal{H} = \emptyset$, then $\mathcal{S} = \mathcal{X}$ and $E = 0$, so all three items of Definition G.1 hold trivially. Assume henceforth that $\mathcal{A} \setminus \mathcal{H} \neq \emptyset$.

We start with regularity. By Proposition D.1(1), $h^*$ is $C^1$ on $\mathcal{Y}$ and $\nabla h^*(y) = Q(y)$, hence, for every fixed $\alpha \in \mathcal{A}$,

$$F_h(\alpha, y) = h(\alpha) + h^*(y) - \langle y, \alpha \rangle \quad \text{is } C^1 \text{ with} \quad \nabla F_h(\alpha, y) = Q(y) - \alpha.$$

Moreover, $F_h(\alpha, y) \geq 0$ by Fenchel–Young, so $\exp(-F_h(\alpha, y)) \in (0, 1]$. Differentiating gives

$$\nabla \left( e^{-F_h(\alpha, \cdot)} \right)(y) = -e^{-F_h(\alpha, y)} (Q(y) - \alpha).$$

Since $Q(y) \in \mathcal{X}$ for all $y$ and $\alpha \in \mathcal{X}$ is fixed, the vectors $Q(y) - \alpha$ are uniformly bounded, therefore the gradients above are uniformly bounded as well (because $e^{-F_h(\alpha,y)} \le 1$). It follows that each map $y \mapsto e^{-F_h(\alpha,y)}$ is globally Lipschitz and $C^1$ on $\mathcal{Y}$. Summing over the finite set $\mathcal{A} \setminus \mathcal{H}$ shows that $E$ is globally Lipschitz and $C^1$, proving [Definition G.1(1)](#).

Now positive semi-definiteness. Let $(y^k)$ be any sequence in $\mathcal{Y}$ and set $x^k := Q(y^k)$. We first show that $x^k \to \mathcal{S}$ implies $E(y^k) \to 0$. Fix $\alpha \notin \mathcal{H}$. Since $x \mapsto x_\alpha = \prod_i x_{i\alpha}$ is continuous and $x_\alpha = 0$ for all $x \in \mathcal{S}$ by definition of $\mathrm{span}(\mathcal{H})$, we have $x_\alpha^k \to 0$. Let

$$\delta_\alpha^k := \min_{i \in \mathcal{N}} x_{i\alpha_i}^k.$$

Because $0 \le x_{i\alpha_i}^k \le 1$ and $x_\alpha^k = \prod_i x_{i\alpha_i}^k \to 0$, we must have $\delta_\alpha^k \to 0$. We claim that $F_h(\alpha, y^k) \to +\infty$, which implies $e^{-F_h(\alpha,y^k)} \to 0$. Indeed, Assumption [1](#) implies that $h$ is steep, so $Q(y^k) \in \mathcal{X}^\circ$ for all $k$, in particular all coordinates of $x^k$ are strictly positive. Thus, for each $k$ and each player $i$, the KKT conditions in [Proposition D.1(5)](#) hold with $\nu_i \equiv 0$, yielding a scalar $\lambda_i^k \in \mathbb{R}$ such that

$$y_{i\beta}^k = \theta_i'(x_{i\beta}^k) + \lambda_i^k, \qquad \forall \beta \in \mathcal{A}_i.$$

Fix $k$ and choose an index $j$ attaining the minimum in $\delta_\alpha^k$, i.e. $x_{j\alpha}^k = \delta_\alpha^k$. Also choose $\beta_j \in \mathcal{A}_j$ with $x_{i\beta}^k \ge 1/|\mathcal{A}_j|$. Using the definition of $h_j^*$ and testing it at the pure action $\beta_j$ gives

$$h_j^*(y_j^k) \ge \langle y_j^k, \beta_j \rangle - h_j(\beta_j) = y_j^k - h_j(\beta_j),$$

so

$$F_j(\alpha_j, y_j^k) = h_j(\alpha_j) + h_j^*(y_j^k) - y_{j\alpha}^k \ge h_j(\alpha_j) - h_j(\beta_j) + (y_{j\beta}^k - y_{j\alpha}^k).$$

By the KKT identity, $y_{j\beta}^k - y_{j\alpha}^k = \theta_j'(x_{j\beta}^k) - \theta_j'(x_{j\alpha}^k)$. Since $\theta_j'$ is increasing and $x_{j\beta}^k \ge 1/|\mathcal{A}_j|$, we have $\theta_j'(x_{j\beta}^k) \ge \theta_j'(1/|\mathcal{A}_j|)$, hence

$$F_j(\alpha_j, y_j^k) \ge \left( \theta_j'(1/|\mathcal{A}_j|) - \left( \max_{\gamma \in \mathcal{A}_j} h_j(\gamma_j) - \min_{\gamma \in \mathcal{A}_j} h_j(\gamma_j) \right) \right) - \theta_j'(\delta_\alpha^k).$$

Define for each $i$ the constant

$$C_i := \theta_i'(1/|\mathcal{A}_i|) - \left( \max_{\gamma_i \in \mathcal{A}_i} h_i(\gamma_i) - \min_{\gamma_i \in \mathcal{A}_i} h_i(\gamma_i) \right) \in \mathbb{R}.$$

Then the preceding inequality yields

$$F_h(\alpha, y^k) = \sum_{j \in \mathcal{N}} F_j(\alpha_j, y_j^k) \ge F_j(\alpha_j, y_j^k) \ge \min_{i \in \mathcal{N}} (C_i - \theta_i'(\delta_\alpha^k)).$$

Since we are in the steep regime, $\theta_i'(p) \to -\infty$ as $p \downarrow 0$ for each $i$. Because $\delta_\alpha^k \to 0$ and $\mathcal{N}$ is finite, the right-hand side diverges to $+\infty$. Hence $F_h(\alpha, y^k) \to +\infty$ and therefore $e^{-F_h(\alpha,y^k)} \to 0$. As $\mathcal{A} \setminus \mathcal{H}$ is finite, summing over $\alpha \notin \mathcal{H}$ yields $E(y^k) \to 0$.

Conversely, assume $E(y^k) \to 0$. We show that $x^k \to \mathcal{S}$. Fix $\alpha \notin \mathcal{H}$ and suppose, for contradiction, that $x_\alpha^k \not\to 0$. Then there exist $\varepsilon > 0$ and a subsequence (not relabeled) such that $x_\alpha^k \ge \varepsilon$ for all $k$, so in particular $x_{i\alpha_i}^k \ge \varepsilon$ for all players $i$ (since each factor is $\le 1$). Fix such an $i$. By the same KKT identity as above (valid because $x^k \in \mathcal{X}^\circ$), we have

$$\max_{\beta_i \in \mathcal{A}_i} y_{i\beta}^k - y_{i\alpha_i}^k = \max_{\beta_i \in \mathcal{A}_i} \theta_i'(x_{i\beta}^k) - \theta_i'(x_{i\alpha_i}^k) \le \theta_i'(1) - \theta_i'(\varepsilon) =: M_i(\varepsilon) < \infty.$$

Hence $y_{i\alpha}^k \ge \max_\beta y_{i\beta}^k - M_i(\varepsilon)$. On the other hand, by definition of $h_i^*$,

$$h_i^*(y_i^k) = \sup_{w_i \in \mathcal{X}_i} \left( \langle y_i^k, w_i \rangle - h_i(w_i) \right) \le \max_{\beta_i \in \mathcal{A}_i} y_{i\beta}^k - \min_{w_i \in \mathcal{X}_i} h_i(w_i).$$

Combining these two bounds gives

$$F_i(\alpha_i, y_i^k) = h_i(\alpha_i) + h_i^*(y_i^k) - y_{i\alpha}^k \le h_i(\alpha_i) - \min_{\mathcal{X}_i} h_i + M_i(\varepsilon) =: C_i'(\varepsilon) < \infty.$$

Summing over $i$ yields a finite constant $C'(\varepsilon) = \sum_i C_i'(\varepsilon)$ such that $F_h(\alpha, y^k) \le C'(\varepsilon)$ along the subsequence, hence $e^{-F_h(\alpha,y^k)} \ge e^{-C'(\varepsilon)} > 0$ along the same subsequence. This contradicts $E(y^k) \to 0$, since $E(y^k) \ge e^{-F_h(\alpha,y^k)}$. Therefore

$x_\alpha^k \to 0$ for every $\alpha \notin \mathcal{H}$. Now define the continuous function $g : \mathcal{X} \to [0, \infty)$ by $g(x) := \sum_{\alpha \notin \mathcal{H}} x_\alpha$. We have $g(x) = 0$ if and only if $x \in \mathcal{S}$, and the above shows $g(x^k) \to 0$. If $x^k \not\to \mathcal{S}$, there exist $\varepsilon_0 > 0$ and a subsequence $x^{k_j}$ with $\mathrm{dist}(x^{k_j}, \mathcal{S}) \geq \varepsilon_0$. By compactness of $\mathcal{X}$, pass to a further subsequence with $x^{k_j} \to \bar{x} \in \mathcal{X}$. Continuity of $g$ gives $g(\bar{x}) = 0$, hence $\bar{x} \in \mathcal{S}$, contradicting $\mathrm{dist}(x^{k_j}, \mathcal{S}) \geq \varepsilon_0$ for $j$ large. Thus $x^k \to \mathcal{S}$. This proves Definition G.1(2).

Finally, dissipativity. Let $y$ be any solution orbit of (FTRL) and set $x := Q(y)$. For each $\alpha \notin \mathcal{H}$, summing Lemma D.5 over players gives

$$\frac{d}{dt} F_h(\alpha, y) = \sum_{i \in \mathcal{N}} \langle v_i(x), x_i - \alpha_i \rangle = \sum_{i \in \mathcal{N}} \big( u_i(x) - v_{i\alpha}(x) \big) = - l_\alpha(x),$$

where we used $u_i(x) = \langle v_i(x), x_i \rangle$ and the definition of $l_\alpha$. Therefore,

$$\frac{d}{dt} e^{-F_h(\alpha, y)} = -e^{-F_h(\alpha, y)} \frac{d}{dt} F_h(\alpha, y) = e^{-F_h(\alpha, y)} l_\alpha(x).$$

Summing over $\alpha \notin \mathcal{H}$ yields the derivative of $E$ along (FTRL):

$$\dot{E}(y) = \sum_{\alpha \notin \mathcal{H}} e^{-F_h(\alpha, y)} l_\alpha(Q(y)). \tag{G.25}$$

We now exploit strict leaklessness. Fix $\alpha \notin \mathcal{H}$. For any $x \in \mathcal{S}$, let $\mathcal{F}$ be the (minimal) face of $\mathcal{X}$ containing $x$, then $\mathcal{F} \subseteq \mathcal{S}$. By Lemma G.6, $\sup_{z \in \mathcal{F}} l_\alpha(z) < 0$, hence $l_\alpha(x) \leq \sup_{z \in \mathcal{F}} l_\alpha(z) < 0$. Thus $l_\alpha < 0$ on $\mathcal{S}$. Since $l_\alpha$ is continuous and $\mathcal{S}$ is compact, the maximum $m_\alpha := \max_{x \in \mathcal{S}} l_\alpha(x)$ exists and satisfies $m_\alpha < 0$. Because $\mathcal{A} \setminus \mathcal{H}$ is finite, the constant

$$c := - \max_{\alpha \notin \mathcal{H}} m_\alpha > 0$$

is well-defined and satisfies $l_\alpha(x) \leq -c$ for all $x \in \mathcal{S}$ and all $\alpha \notin \mathcal{H}$. By continuity, there exists a neighborhood $U$ of $\mathcal{S}$ in $\mathcal{X}$ such that

$$l_\alpha(x) \leq -\frac{c}{2} \qquad \forall x \in U, \ \forall \alpha \notin \mathcal{H}.$$

By Definition G.1(2) already established, there exists $\bar{E} > 0$ such that $E(y) < \bar{E}$ implies $Q(y) \in U$ (otherwise, one could find $y^k$ with $E(y^k) \to 0$ but $Q(y^k) \notin U$, contradicting $Q(y^k) \to \mathcal{S}$). Hence, for any $y$ with $E(y) \leq \bar{E}$, we have $x = Q(y) \in U$ and thus, using (G.25),

$$\dot{E}(y) = \sum_{\alpha \notin \mathcal{H}} e^{-F_h(\alpha, y)} l_\alpha(x) \leq -\frac{c}{2} \sum_{\alpha \notin \mathcal{H}} e^{-F_h(\alpha, y)} = -\frac{c}{2} E(y).$$

Consequently, if $0 < E^- < E^+ \leq \bar{E}$ and $E^- < E(y) < E^+$, then $\dot{E}(y) \leq -(c/2) E^- < 0$. Taking the supremum over $\{y : E^- < E(y) < E^+\}$ gives Definition G.1(3).

All three conditions of Definition G.1 are satisfied, so $\bar{F}_{\mathcal{H}}$ is a local energy function for $\mathrm{span}(\mathcal{H})$. $\square$

We may now use this to deduce the main theorem.

***Proof of Theorem 4.*** We shall proceed again by gluing all the facewise basins into a basin on all of $\mathcal{X}$. Let $\mathcal{S} := \mathrm{span}(\mathcal{H})$. Under Assumption 1, the regularizer is steep, so the facewise FTRL dynamics are well-defined. Moreover, $\mathcal{S}$ is a union of faces, so compact and each face is invariant under the strategy flow, hence $\mathcal{S}$ is invariant.

It remains to prove that $\mathcal{S}$ is asymptotically stable for the strategy flow. Fix any face $\mathcal{F}$ of $\mathcal{X}$ with $\mathcal{S}_{\mathcal{F}} := \mathcal{S} \cap \mathcal{F} \neq \varnothing$, and write $\mathcal{F} = \mathrm{span}(\mathcal{B})$ for the corresponding subgame $\mathcal{B} = \prod_i \mathcal{B}_i \subseteq \mathcal{A}$. Define $\mathcal{H}_{\mathcal{F}} := \mathcal{H} \cap \mathcal{B}$. We claim

$$\mathcal{S}_{\mathcal{F}} = \mathrm{span}(\mathcal{H}_{\mathcal{F}}). \tag{G.26}$$

Indeed, if $x \in \mathcal{S} \cap \mathcal{F}$, then $x_\alpha = 0$ for every $\alpha \notin \mathcal{H}$ (since $x \in \mathrm{span}(\mathcal{H})$) and also $x_\alpha = 0$ for every $\alpha \notin \mathcal{B}$ (since $x \in \mathrm{span}(\mathcal{B})$), therefore $x_\alpha = 0$ for every $\alpha \notin (\mathcal{H} \cap \mathcal{B}) = \mathcal{H}_{\mathcal{F}}$, i.e. $x \in \mathrm{span}(\mathcal{H}_{\mathcal{F}})$. Conversely, $\mathrm{span}(\mathcal{H}_{\mathcal{F}}) \subseteq \mathrm{span}(\mathcal{H}) \cap \mathrm{span}(\mathcal{B}) = \mathcal{S} \cap \mathcal{F}$, proving (G.40).

Next, we show that $\mathcal{H}_{\mathcal{F}}$ is strictly leakless for the restricted game on $\mathcal{B}$. Fix $\alpha \in \mathcal{B} \setminus \mathcal{H}_{\mathcal{F}}$. Then $\alpha \notin \mathcal{H}$. Let $\mathcal{G}$ be any face with $\mathcal{G} \subseteq \mathrm{span}(\mathcal{H}_{\mathcal{F}})$. By (G.40), $\mathcal{G} \subseteq \mathrm{span}(\mathcal{H})$. By strict leaklessness of $\mathcal{H}$ and the characterization of strict leaklessness (cf. Lemma G.6), we have

$$\sup_{x \in \mathcal{G}} l_\alpha(x) < 0.$$

Since $\alpha \in \mathcal{B}$ and $\mathcal{G} \subseteq \text{span}(\mathcal{B})$, this is exactly the strict leaklessness condition for $\mathcal{H}_{\mathcal{F}}$ within $\mathcal{B}$.

Now consider the face-restricted choice map $Q_{\mathcal{B}} : \mathcal{Y} \rightarrow \mathcal{F}^{\circ}$ and the facewise FTRL dynamics

$$\dot{y} = v(x), \qquad x = Q_{\mathcal{B}}(y), \qquad\qquad \text{(FTRL-}\mathcal{B})$$

as in Remark E.9. Since $h$ is steep and $\mathcal{H}_{\mathcal{F}}$ is strictly leakless, the leakless energy construction (cf. Lemma G.7) provides a local energy function for $\text{span}(\mathcal{H}_{\mathcal{F}}) = \mathcal{S}_{\mathcal{F}}$ of the form

$$\bar{F}_{\mathcal{H}_{\mathcal{F}}}(y) = \sum_{\alpha \in \mathcal{B} \backslash \mathcal{H}_{\mathcal{F}}} e^{-F_{h|\mathcal{F}}(\alpha, y)}.$$

Therefore, by the continuous-time energy theorem (cf. Theorem G.2), $\mathcal{S}_{\mathcal{F}}$ is asymptotically stable under the facewise FTRL dynamics. Finally, by Remark E.9, for every initial condition $x_0 \in \mathcal{F}$ the strategy flow trajectory starting at $x_0$ coincides with the appropriate facewise FTRL orbit on the minimal face of $\mathcal{F}$ containing $x_0$. It follows that $\mathcal{S}_{\mathcal{F}}$ is asymptotically stable for the restriction of the strategy flow to $\mathcal{F}$. Concretely: for each such face $\mathcal{F}$ there exists a neighborhood $U_{\mathcal{F}} \subseteq \mathcal{F}$ of $\mathcal{S}_{\mathcal{F}}$ (relative topology) such that every strategy flow trajectory starting in $U_{\mathcal{F}}$ remains in $U_{\mathcal{F}}$ and satisfies $\text{dist}(X_t, \mathcal{S}_{\mathcal{F}}) \rightarrow 0$ as $t \rightarrow \infty$.

We now patch these facewise basins into a basin for $\mathcal{S}$ in $\mathcal{X}$. Let $\mathscr{F}_{\mathcal{S}}$ be the (finite) family of faces $\mathcal{F}$ of $\mathcal{X}$ with $\mathcal{S}_{\mathcal{F}} \neq \varnothing$, and fix for each $\mathcal{F} \in \mathscr{F}_{\mathcal{S}}$ a neighborhood $U_{\mathcal{F}} \subseteq \mathcal{F}$ with the invariance/attraction property above. Fix $x \in \mathcal{S}$ and set

$$\mathscr{F}(x) := \{\mathcal{F} \in \mathscr{F}_{\mathcal{S}} : x \in \mathcal{F}\}.$$

Because $\mathcal{X}$ has finitely many faces, the number

$$r_x := \tfrac{1}{2} \min\{\text{dist}(x, \mathcal{F}') : \mathcal{F}' \text{ a face of } \mathcal{X}, x \notin \mathcal{F}'\}$$

is well-defined and strictly positive. In particular, $B_{r_x}(x) \cap \mathcal{X}$ meets only faces that contain $x$. Moreover, for each $\mathcal{F} \in \mathscr{F}(x)$, since $U_{\mathcal{F}}$ is a neighborhood of $x$ in $\mathcal{F}$ (relative topology), there exists $\rho_{\mathcal{F}}^x > 0$ such that $B_{\rho_{\mathcal{F}}^x}(x) \cap \mathcal{F} \subseteq U_{\mathcal{F}}$. Define

$$\rho_x := \min\Big(r_x, \min_{\mathcal{F} \in \mathscr{F}(x)} \rho_{\mathcal{F}}^x\Big) > 0.$$

Then for every $z \in B_{\rho_x}(x) \cap \mathcal{X}$, every face of $\mathcal{X}$ containing $z$ must contain $x$ (since it meets $B_{r_x}(x) \cap \mathcal{X}$), so the minimal face $\mathcal{F}_z$ containing $z$ contains $x$, hence $\mathcal{F}_z \in \mathscr{F}(x)$, by construction, this implies $z \in U_{\mathcal{F}_z}$. Consequently, the family $\{B_{\rho_x}(x) \cap \mathcal{X}\}_{x \in \mathcal{S}}$ covers $\mathcal{S}$. By compactness of $\mathcal{S}$, choose $x^1, \ldots, x^m \in \mathcal{S}$ with

$$\mathcal{S} \subseteq U := \bigcup_{k=1}^{m} \big(B_{\rho_{x^k}}(x^k) \cap \mathcal{X}\big),$$

so $U$ is an open neighborhood of $\mathcal{S}$ in $\mathcal{X}$.

*Attraction.* Take any $X_0 \in U$ and let $\mathcal{F}$ be the minimal face of $\mathcal{X}$ containing $X_0$. By construction of $U$, there exists some $k$ such that $X_0 \in B_{\rho_{x^k}}(x^k) \cap \mathcal{X}$, hence $X_0 \in U_{\mathcal{F}}$ by the preceding argument. Since $\mathcal{F}$ is invariant under the strategy flow, the trajectory $t \mapsto X_t(X_0)$ remains in $\mathcal{F}$. Since $X_0 \in U_{\mathcal{F}}$, it remains in $U_{\mathcal{F}}$ and satisfies

$$\text{dist}\big(X_t(X_0), \mathcal{S}_{\mathcal{F}}\big) \xrightarrow[t \rightarrow \infty]{} 0.$$

Because $\mathcal{S}_{\mathcal{F}} \subseteq \mathcal{S}$, we have $\text{dist}(X_t(X_0), \mathcal{S}) \leq \text{dist}(X_t(X_0), \mathcal{S}_{\mathcal{F}}) \rightarrow 0$, proving that $\mathcal{S}$ attracts $U$.

*Stability.* Let $W$ be an arbitrary neighborhood of $\mathcal{S}$ in $\mathcal{X}$. For each $\mathcal{F} \in \mathscr{F}_{\mathcal{S}}$, the set $W \cap \mathcal{F}$ is a neighborhood of $\mathcal{S}_{\mathcal{F}}$ in the relative topology of $\mathcal{F}$. Since $\mathcal{S}_{\mathcal{F}}$ is stable for the restricted flow on $\mathcal{F}$, there exists a neighborhood $V_{\mathcal{F}} \subseteq \mathcal{F}$ of $\mathcal{S}_{\mathcal{F}}$ such that $x_0 \in V_{\mathcal{F}}$ implies $X_t(x_0) \in W \cap \mathcal{F}$ for all $t \geq 0$. Repeating the compactness/ball-cover construction above with $(V_{\mathcal{F}})_{\mathcal{F} \in \mathscr{F}_{\mathcal{S}}}$ in place of $(U_{\mathcal{F}})_{\mathcal{F} \in \mathscr{F}_{\mathcal{S}}}$ yields an open neighborhood $V$ of $\mathcal{S}$ in $\mathcal{X}$ such that every $x_0 \in V$ lies in $V_{\mathcal{F}_{x_0}}$, where $\mathcal{F}_{x_0}$ is the minimal face containing $x_0$, hence $X_t(x_0) \in W$ for all $t \geq 0$. Thus $\mathcal{S}$ is stable.

We have shown that $\mathcal{S}$ is compact, invariant, and asymptotically stable for the strategy flow. It follows then that $\text{span}(\mathcal{H}) = \mathcal{S}$ is an attractor for the strategy flow induced by (SD). $\qquad\square$

In the entropic setting, we shall work with a much simpler energy function:

**Lemma G.8.** *Let $\mathcal{H} \subseteq \mathcal{A}$ and assume $h$ is entropic. Define*

$$\bar{W}_{\mathcal{H}}(x) := \sum_{\alpha \notin \mathcal{H}} x_\alpha, \qquad x \in \mathcal{X}.$$

*Then, if $\mathcal{H}$ is leakless and closed under better replies, $\bar{W}_{\mathcal{H}} \circ Q$ is a local energy function for* $\mathrm{span}(\mathcal{H})$.

*Proof.* Set $\mathcal{S} := \mathrm{span}(\mathcal{H})$ and define

$$E : \mathcal{Y} \to [0, 1], \qquad E(y) := \bar{W}_{\mathcal{H}}(Q(y)).$$

If $\mathcal{A} \setminus \mathcal{H} = \varnothing$, then $\mathcal{S} = \mathcal{X}$ and $\bar{W}_{\mathcal{H}} = 0$, hence $E = 0$, and the three items of Definition G.1 hold trivially. Assume henceforth $\mathcal{A} \setminus \mathcal{H} \neq \varnothing$.

We start with regularity. Since $h$ is entropic, $Q$ is the softmax, hence $C^\infty$. Moreover by Proposition D.1(1), it is globally Lipschitz. Next, $\bar{W}_{\mathcal{H}}$ is a polynomial in the coordinates $x_{i\alpha}$ because each monomial $x_\alpha = \prod_i x_{i\alpha}$ is polynomial, thus $\bar{W}_{\mathcal{H}} \in C^\infty(\mathcal{X})$ and is Lipschitz on compact $\mathcal{X}$. Therefore $E = \bar{W}_{\mathcal{H}} \circ Q$ is $C^1$ and globally Lipschitz on $\mathcal{Y}$.

For positive semi-definiteness, $\bar{W}_{\mathcal{H}}(x)$ is a sum of nonnegative terms, so

$$\bar{W}_{\mathcal{H}}(x) = 0 \quad \Longleftrightarrow \quad x_\alpha = 0 \ \forall \alpha \notin \mathcal{H} \quad \Longleftrightarrow \quad x \in \mathcal{S},$$

hence

$$\mathcal{S} = \bar{W}_{\mathcal{H}}^{-1}(0). \tag{G.27}$$

We do again the basic subsequence argument. Let $(y^k)$ be any sequence and set $x^k := Q(y^k) \in \mathcal{X}$. If $x^k \to \mathcal{S}$, then by continuity of $\bar{W}_{\mathcal{H}}$ and (G.27), $E(y^k) = \bar{W}_{\mathcal{H}}(x^k) \to 0$. Conversely, if $E(y^k) = \bar{W}_{\mathcal{H}}(x^k) \to 0$ but $x^k \not\to \mathcal{S}$, then there exist $\varepsilon_0 > 0$ and a subsequence $x^{k_j}$ with $\mathrm{dist}(x^{k_j}, \mathcal{S}) \geq \varepsilon_0$ for all $j$. By compactness of $\mathcal{X}$, pass to a further subsequence (not relabeled) with $x^{k_j} \to \bar{x} \in \mathcal{X}$. Continuity gives $\bar{W}_{\mathcal{H}}(\bar{x}) = 0$, so $\bar{x} \in \mathcal{S}$ by (G.27). Since $x \mapsto \mathrm{dist}(x, \mathcal{S})$ is continuous and $\mathrm{dist}(\bar{x}, \mathcal{S}) = 0$, we get $\mathrm{dist}(x^{k_j}, \mathcal{S}) \to 0$, contradicting $\mathrm{dist}(x^{k_j}, \mathcal{S}) \geq \varepsilon_0$ for $j$ large. Hence $x^k \to \mathcal{S}$.

We now turn to dissipativity. Let $y$ be any solution orbit of (FTRL) and set $x := Q(y)$. In the entropic case, the induced strategy flow coincides with the replicator dynamics

$$\dot{x}_{i\alpha} = x_{i\alpha}\big(v_{i\alpha}(x) - u_i(x)\big), \qquad i \in \mathcal{N}, \ \alpha_i \in \mathcal{A}_i. \tag{G.28}$$

By steepness, $x_{i\alpha} > 0$ for all $i$ and $\alpha_i \in \mathcal{A}_i$. The product rule and (G.28) then give

$$\frac{d}{dt} x_\alpha = x_\alpha \sum_{i \in \mathcal{N}} \frac{\dot{x}_{i\alpha_i}}{x_{i\alpha_i}} = x_\alpha \sum_{i \in \mathcal{N}} \big(v_{i\alpha}(x) - u_i(x)\big).$$

Also

$$l_\alpha(x) = \sum_{i \in \mathcal{N}} \big(v_{i\alpha}(x) - u_i(x)\big),$$

so we obtain the pointwise identity

$$\dot{x}_\alpha = x_\alpha \, l_\alpha(x). \tag{G.29}$$

Summing (G.29) over $\alpha \notin \mathcal{H}$ yields, along the orbit,

$$\frac{d}{dt} \bar{W}_{\mathcal{H}}(x) = \sum_{\alpha \notin \mathcal{H}} x_\alpha \, l_\alpha(x). \tag{G.30}$$

Equivalently, for every $y \in \mathcal{Y}$,

$$\dot{E}(y) = \sum_{\alpha \notin \mathcal{H}} x_\alpha \, l_\alpha(x), \qquad x = Q(y). \tag{G.31}$$

Define the continuous function

$$\Phi(x) := \sum_{\alpha \notin \mathcal{H}} x_\alpha \, l_\alpha(x), \qquad x \in \mathcal{X}.$$

By (G.31), it is enough to show that $\Phi$ is strictly negative whenever the "mass outside $\mathcal{H}$", namely $\bar{W}_{\mathcal{H}}(x)$, is positive but small: we will prove that there exists $\bar{E} > 0$ such that

$$0 < \bar{W}_{\mathcal{H}}(x) \le \bar{E} \quad \implies \quad \Phi(x) < 0. \tag{G.32}$$

Fix $x^\star \in \mathcal{S}$. Set $\mathcal{B}_i := \text{supp}(x_i^\star)$ and $\mathcal{B} := \prod_{i \in \mathcal{N}} \mathcal{B}_i$. Since $x^\star \in \mathcal{S}$, we have $\mathcal{B} \subseteq \mathcal{H}$. Let $\mathcal{F} := \text{span}(\mathcal{B})$, then $x^\star \in \mathcal{F}^\circ$. For $\alpha \in \mathcal{A}$ define the *deviating coalition* from $\mathcal{B}$:

$$D_{\mathcal{B}}(\alpha) := \{i \in \mathcal{N} : \alpha_i \notin \mathcal{B}_i\},$$

and on $\mathcal{A} \setminus \mathcal{H}$ introduce the preorder $\preceq_{\mathcal{B}}$ by: $\alpha \preceq_{\mathcal{B}} \nu$ if and only if $D_{\mathcal{B}}(\alpha) \subseteq D_{\mathcal{B}}(\nu)$ and $\alpha_i = \nu_i$ for all $i \in D_{\mathcal{B}}(\alpha)$. Let $M_{\mathcal{B}} \subseteq \mathcal{A} \setminus \mathcal{H}$ be the set of $\preceq_{\mathcal{B}}$-minimal excluded profiles, and set

$$\bar{W}_{\mathcal{B}}(x) := \sum_{\alpha \in M_{\mathcal{B}}} x_\alpha.$$

The role of $M_{\mathcal{B}}$ is that it collects those excluded profiles that are "closest" to $\mathcal{B}$ in the sense that one cannot replace any off-$\mathcal{B}$ coordinate by an element of $\mathcal{B}_i$ and remain excluded, this is exactly what we use to obtain a strictly negative contribution coming from clubness.

*Claim 1.* If $\alpha \in M_{\mathcal{B}}$, $i \in D_{\mathcal{B}}(\alpha)$, and $\beta_i \in \mathcal{B}_i$, then $(\beta_i, \alpha_{-i}) \in \mathcal{H}$. Indeed, let $\gamma = (\beta_i, \alpha_{-i})$. Then $D_{\mathcal{B}}(\gamma) = D_{\mathcal{B}}(\alpha) \setminus \{i\} \subsetneq D_{\mathcal{B}}(\alpha)$ and $\gamma_j = \alpha_j$ for all $j \in D_{\mathcal{B}}(\gamma)$, hence $\gamma \preceq_{\mathcal{B}} \alpha$ with $\gamma \ne \alpha$. If $\gamma \notin \mathcal{H}$, this contradicts the $\preceq_{\mathcal{B}}$-minimality of $\alpha \in M_{\mathcal{B}}$, therefore $\gamma \in \mathcal{H}$.

*Claim 2.* There exist a neighborhood $U$ of $x^\star$ in $\mathcal{X}$ and $\gamma \in (0, 1)$ such that

$$x \in U \setminus \mathcal{S} \quad \implies \quad \bar{W}_{\mathcal{B}}(x) \ge \gamma \bar{W}_{\mathcal{H}}(x). \tag{G.33}$$

Indeed, let $n := |\mathcal{N}|$ and set

$$\delta := \frac{1}{2} \min_{i \in \mathcal{N}, \, \beta_i \in \mathcal{B}_i} x_{i\beta}^\star > 0.$$

By continuity of $x \mapsto x_{i\beta}$, there exists a neighborhood $U_0$ of $x^\star$ such that

$$x_{i\beta} \ge \delta \qquad \forall x \in U_0, \ \forall i \in \mathcal{N}, \ \forall \beta_i \in \mathcal{B}_i. \tag{G.34}$$

Define

$$\varepsilon(x) := \max_{i \in \mathcal{N}} \sum_{\alpha_i \notin \mathcal{B}_i} x_{i\alpha_i}.$$

Since $\varepsilon(x^\star) = 0$, shrink $U_0$ so that $\varepsilon(x) \le \varepsilon_0 \le \delta$ for all $x \in U_0$, where $\varepsilon_0 > 0$ will be fixed below. Fix $\nu \in (\mathcal{A} \setminus \mathcal{H}) \setminus M_{\mathcal{B}}$. Let $\mu = \mu(\nu)$ be any $\preceq_{\mathcal{B}}$-minimal element of the nonempty set $\{\alpha \in \mathcal{A} \setminus \mathcal{H} : \alpha \preceq_{\mathcal{B}} \nu\}$. Then $\mu \in M_{\mathcal{B}}$ and $\mu \preceq_{\mathcal{B}} \nu$. Moreover $D_{\mathcal{B}}(\mu) \subsetneq D_{\mathcal{B}}(\nu)$: if $D_{\mathcal{B}}(\mu) = D_{\mathcal{B}}(\nu)$, then $\mu_i = \nu_i$ for all $i \in D_{\mathcal{B}}(\nu)$ and also $\mu \preceq_{\mathcal{B}} \nu$ and $\nu \preceq_{\mathcal{B}} \mu$, so $\nu$ would be $\preceq_{\mathcal{B}}$-minimal as well, contradicting $\nu \notin M_{\mathcal{B}}$. We claim that for all $x \in U_0$,

$$x_\nu \le C \varepsilon(x) x_\mu, \qquad C := \delta^{-n}. \tag{G.35}$$

If $x_\mu = 0$, then $x_\nu = 0$ as well, hence (G.35) holds: indeed, if $x_\mu = 0$ then some index $j$ has $x_{j\mu} = 0$. By (G.34) this forces $j \in D_{\mathcal{B}}(\mu)$, and since $\mu_j = \nu_j$ for all $j \in D_{\mathcal{B}}(\mu)$ we get $x_{j\nu} = 0$ and thus $x_\nu = 0$. Assume now $x_\mu > 0$ and write

$$\frac{x_\nu}{x_\mu} = \prod_{i \in \mathcal{N}} \frac{x_{i\nu}}{x_{i\mu}}.$$

If $i \in D_{\mathcal{B}}(\mu)$ then $\mu_i = \nu_i$, so the factor equals 1. If $i \in D_{\mathcal{B}}(\nu) \setminus D_{\mathcal{B}}(\mu)$ then $\mu_i \in \mathcal{B}_i$ and $\nu_i \notin \mathcal{B}_i$, hence by (G.34) and the definition of $\varepsilon(x)$,

$$x_{i\mu} \ge \delta, \qquad x_{i\nu} \le \sum_{\alpha_i \notin \mathcal{B}_i} x_{i\alpha} \le \varepsilon(x), \quad \Rightarrow \quad \frac{x_{i\nu}}{x_{i\mu}} \le \frac{\varepsilon(x)}{\delta}.$$

Finally, if $i \notin D_{\mathcal{B}}(\nu)$ then $\nu_i \in \mathcal{B}_i$, and also $\mu_i \in \mathcal{B}_i$ (else $i \in D_{\mathcal{B}}(\mu) \subseteq D_{\mathcal{B}}(\nu)$), so $x_{i\mu} \geq \delta$ by (G.34) and trivially $x_{i\nu} \leq 1$, hence $\frac{x_{i\nu}}{x_{i\mu}} \leq 1/\delta$. Since $\nu \notin \mathcal{H}$ and $\mathcal{B} \subseteq \mathcal{H}$, we have $\nu \notin \mathcal{B}$ and thus $D_{\mathcal{B}}(\nu) \neq \varnothing$, implying $n - |D_{\mathcal{B}}(\nu)| \leq n - 1$. Also $D_{\mathcal{B}}(\mu) \subsetneq D_{\mathcal{B}}(\nu)$ implies $|D_{\mathcal{B}}(\nu) \setminus D_{\mathcal{B}}(\mu)| \geq 1$. Using $\varepsilon(x) \leq \delta$ on $U_0$ gives

$$\frac{x_\nu}{x_\mu} \leq \left( \frac{\varepsilon(x)}{\delta} \right) \left( \frac{1}{\delta} \right)^{n-1} = \delta^{-n} \varepsilon(x) = C \varepsilon(x),$$

proving (G.35). Summing (G.35) over $\nu \in (\mathcal{A} \setminus \mathcal{H}) \setminus M_{\mathcal{B}}$ yields, for all $x \in U_0$,

$$\bar{W}_{\mathcal{H}}(x) = \bar{W}_{\mathcal{B}}(x) + \sum_{\nu \in (\mathcal{A} \setminus \mathcal{H}) \setminus M_{\mathcal{B}}} x_\nu \leq \bar{W}_{\mathcal{B}}(x) + C \varepsilon(x) |\mathcal{A} \setminus \mathcal{H}| \bar{W}_{\mathcal{B}}(x).$$

Choose $\varepsilon_0$ so that $C \varepsilon_0 |\mathcal{A} \setminus \mathcal{H}| \leq \frac{1}{2}$, and set $U := U_0$ with this choice. Then for all $x \in U$ we have $\bar{W}_{\mathcal{H}}(x) \leq \frac{3}{2} \bar{W}_{\mathcal{B}}(x)$, i.e. (G.33) holds with $\gamma := 2/3$.

Next, since $\mathcal{H}$ is club, every one-step deviation from $\mathcal{H}$ to $\mathcal{A} \setminus \mathcal{H}$ is strictly unprofitable. Formally, let $\mathcal{E}$ be the finite set of triples $(\beta, \alpha, i)$ such that $\beta \in \mathcal{H}$, $\alpha \notin \mathcal{H}$, and $\alpha$ and $\beta$ $i$-comparable. For $(\beta, \alpha, i) \in \mathcal{E}$, closure under better replies implies $u_i(\alpha) < u_i(\beta)$, hence define

$$\eta := \min_{(\beta, \alpha, i) \in \mathcal{E}} \left( u_i(\beta) - u_i(\alpha) \right) > 0.$$

If $\alpha, \beta$ differ only in $i$, then for $j \neq i$ we have $u_j(\alpha_j, \beta_{-j}) = u_j(\beta)$, so

$$l_\alpha(\beta) = \sum_{j \in \mathcal{N}} \left( u_j(\alpha_j, \beta_{-j}) - u_j(\beta) \right) = u_i(\alpha) - u_i(\beta) \leq -\eta. \tag{G.36}$$

We now estimate $\Phi(x)$ for $x$ near $\mathcal{S}$. Fix $\alpha \in \mathcal{A}$. Since $x \mapsto l_\alpha(x)$ is multilinear (being a finite sum of multilinear payoff terms), for every $x \in \mathcal{X}$ we have the multilinear interpolation identity

$$l_\alpha(x) = \sum_{\beta \in \mathcal{A}} x_\beta l_\alpha(\beta), \tag{G.37}$$

obtained by expanding successively in each block $x_i$. Using (G.37),

$$\Phi(x) = \sum_{\alpha \notin \mathcal{H}} \sum_{\beta \in \mathcal{H}} x_\alpha x_\beta l_\alpha(\beta) + \sum_{\alpha \notin \mathcal{H}} \sum_{\beta \notin \mathcal{H}} x_\alpha x_\beta l_\alpha(\beta). \tag{G.38}$$

Let $D := \max_{\alpha, \beta \in \mathcal{A}} |l_\alpha(\beta)| < \infty$. Then the second term in (G.38) is bounded above by

$$\sum_{\alpha \notin \mathcal{H}} \sum_{\beta \notin \mathcal{H}} x_\alpha x_\beta |l_\alpha(\beta)| \leq D \left( \sum_{\alpha \notin \mathcal{H}} x_\alpha \right)^2 = D \bar{W}_{\mathcal{H}}(x)^2.$$

For the first term, leaklessness gives $l_\alpha(\beta) \leq 0$ for all $\alpha \notin \mathcal{H}$ and all $\beta \in \mathcal{H}$, hence the first term is $\leq 0$. To obtain strict negativity when $x \notin \mathcal{S}$ but is close to $x^\star$, we focus on the minimal excluded profiles. Fix $\alpha \in M_{\mathcal{B}}$ and choose any $i(\alpha) \in D_{\mathcal{B}}(\alpha)$. Pick $\beta_{i(\alpha)} \in \mathcal{B}_{i(\alpha)}$ maximizing $x_{i(\alpha)\beta}$ over $\mathcal{B}_{i(\alpha)}$, and set

$$\beta(\alpha) := (\beta_{i(\alpha)}, \alpha_{-i(\alpha)}).$$

By Claim 1, $\beta(\alpha) \in \mathcal{H}$, and $\alpha$ and $\beta(\alpha)$ differ only in player $i(\alpha)$, so by (G.36), $l_\alpha(\beta(\alpha)) \leq -\eta$. Therefore, using again $l_\alpha(\beta) \leq 0$ for $\beta \in \mathcal{H}$,

$$\sum_{\alpha \notin \mathcal{H}} \sum_{\beta \in \mathcal{H}} x_\alpha x_\beta l_\alpha(\beta) \leq \sum_{\alpha \in M_{\mathcal{B}}} x_\alpha x_{\beta(\alpha)} l_\alpha(\beta(\alpha)) \leq -\eta \sum_{\alpha \in M_{\mathcal{B}}} x_\alpha x_{\beta(\alpha)}.$$

For $x \in U$ from Claim 2 and $\alpha \in M_{\mathcal{B}}$, we have $\beta_{i(\alpha)} \in \mathcal{B}_{i(\alpha)}$, so (G.34) gives $x_{i(\alpha)\beta} \geq \delta$, while $\alpha_{i(\alpha)} \notin \mathcal{B}_{i(\alpha)}$ implies $x_{i(\alpha)\alpha} \leq \varepsilon(x)$. Hence, whenever $x_\alpha > 0$,

$$x_{\beta(\alpha)} = x_\alpha \frac{x_{i(\alpha)\beta}}{x_{i(\alpha)\alpha}} \geq x_\alpha \frac{\delta}{\varepsilon(x)},$$

and the same inequality holds trivially if $x_\alpha = 0$. Thus

$$x_\alpha x_{\beta(\alpha)} \geq \frac{\delta}{\varepsilon(x)} x_\alpha^2, \qquad \alpha \in M_{\mathcal{B}}.$$

Consequently,

$$\sum_{\alpha \notin \mathcal{H}} \sum_{\beta \in \mathcal{H}} x_\alpha x_\beta \, l_\alpha(\beta) \leq -\eta \, \frac{\delta}{\varepsilon(x)} \sum_{\alpha \in M_\mathcal{B}} x_\alpha^2 \leq -\eta \, \frac{\delta}{|M_\mathcal{B}| \, \varepsilon(x)} \, \bar{W}_\mathcal{B}(x)^2,$$

where we used $\sum_{\alpha \in M_\mathcal{B}} x_\alpha^2 \geq |M_\mathcal{B}|^{-1} \big( \sum_{\alpha \in M_\mathcal{B}} x_\alpha \big)^2 = |M_\mathcal{B}|^{-1} \bar{W}_\mathcal{B}(x)^2$. Combining with the bound on the second term in (G.38) and (G.33) yields, for all $x \in U \setminus \mathcal{S}$,

$$\Phi(x) \leq -\eta \, \frac{\delta}{|M_\mathcal{B}| \, \varepsilon(x)} \, \bar{W}_\mathcal{B}(x)^2 + D \, \bar{W}_\mathcal{H}(x)^2 \leq \bar{W}_\mathcal{B}(x)^2 \Big( -\eta \, \frac{\delta}{|M_\mathcal{B}| \, \varepsilon(x)} + D \, \gamma^{-2} \Big).$$

Note that on $U \setminus \mathcal{S}$ we have $\varepsilon(x) > 0$ (if $\varepsilon(x) = 0$ then $x_i$ is supported on $\mathcal{B}_i$ for every $i$, and since $\mathcal{B} \subseteq \mathcal{H}$ this forces $x \in \mathcal{S}$). Since $\varepsilon(x^\star) = 0$ and $\varepsilon$ is continuous, we may shrink $U$ (keeping $x^\star \in U$) so that

$$\varepsilon(x) \leq \frac{\eta \, \delta \, \gamma^2}{2D \, |M_\mathcal{B}|} \qquad \forall x \in U.$$

Then the bracket is strictly negative for all $x \in U \setminus \mathcal{S}$, hence $\Phi(x) < 0$ on $U \setminus \mathcal{S}$.

Since $x^\star \in \mathcal{S}$ was arbitrary, these neighborhoods form an open cover of compact $\mathcal{S}$. Extracting a finite subcover and letting $U$ be the union, we obtain an open neighborhood $U$ of $\mathcal{S}$ such that

$$\Phi(x) < 0 \qquad \forall x \in U \setminus \mathcal{S}. \tag{G.39}$$

Because $\bar{W}_\mathcal{H}$ is continuous, $\mathcal{S} = \bar{W}_\mathcal{H}^{-1}(0)$, and $\mathcal{X} \setminus U$ is compact and disjoint from $\mathcal{S}$,

$$\bar{E} := \inf_{x \in \mathcal{X} \setminus U} \bar{W}_\mathcal{H}(x) > 0.$$

Then $\{x \in \mathcal{X} : 0 < \bar{W}_\mathcal{H}(x) \leq \bar{E}\} \subseteq U \setminus \mathcal{S}$, and (G.32) follows from (G.39). Finally, let $0 < E^- < E^+ \leq \bar{E}$ and consider the compact set

$$K := \{x \in \mathcal{X} : E^- \leq \bar{W}_\mathcal{H}(x) \leq E^+\} \subseteq \{x : 0 < \bar{W}_\mathcal{H}(x) \leq \bar{E}\}.$$

By (G.32), $\Phi(x) < 0$ for all $x \in K$, hence $\max_{x \in K} \Phi(x) < 0$ by continuity. Using (G.31), for any $y$ with $E^- < E(y) < E^+$ we have $x = Q(y) \in K$ and $\dot{E}(y) = \Phi(x)$, so

$$\sup\{\dot{E}(y) : E^- < E(y) < E^+\} \leq \max_{x \in K} \Phi(x) < 0,$$

which is exactly Definition G.1(3). Hence $E = \bar{W}_\mathcal{H} \circ Q$ is a local energy function for $\mathcal{S} = \text{span}(\mathcal{H})$. $\qquad \square$

From this, as usual, we get asymptotic stability.

***Proof of Theorem 5.*** If $\mathcal{A} \setminus \mathcal{H} = \varnothing$, then $\text{span}(\mathcal{H}) = \mathcal{X}$, hence it is (trivially) an attractor for (RD). Assume henceforth $\mathcal{A} \setminus \mathcal{H} \neq \varnothing$, and set $\mathcal{S} := \text{span}(\mathcal{H})$. Since $h$ is entropic the induced strategy flow $(X_t)_{t \in \mathbb{R}}$ is well-defined and face-invariant, moreover it coincides with the replicator dynamics (RD). In particular, by Proposition E.7 every face of $\mathcal{X}$ is invariant for $(X_t)$. As $\mathcal{S}$ is a union of faces, it follows that $\mathcal{S}$ is compact and invariant. We will prove that $\mathcal{S}$ is asymptotically stable for $(X_t)$, hence an attractor.

Comes again the time for some gluing: we shall patch the facewise basins into a basin on all of $\mathcal{X}$. Let $\mathcal{F} = \prod_{i \in \mathcal{N}} \Delta(\mathcal{B}_i)$ be a face of $\mathcal{X}$ and write $\mathcal{A}(\mathcal{F}) := \prod_{i \in \mathcal{N}} \mathcal{B}_i$ for its vertex set. Define

$$\mathcal{H}_\mathcal{F} := \mathcal{H} \cap \mathcal{A}(\mathcal{F}), \qquad \mathcal{S}_\mathcal{F} := \mathcal{S} \cap \mathcal{F}.$$

Then

$$\mathcal{S}_\mathcal{F} = \text{span}(\mathcal{H}_\mathcal{F}). \tag{G.40}$$

Indeed, if $x \in \mathcal{S}_\mathcal{F}$, then $\text{supp}(x) \subseteq \mathcal{A}(\mathcal{F})$ (since $x \in \mathcal{F}$) and $\text{supp}(x) \subseteq \mathcal{H}$ (since $x \in \text{span}(\mathcal{H})$), so $\text{supp}(x) \subseteq \mathcal{H} \cap \mathcal{A}(\mathcal{F}) = \mathcal{H}_\mathcal{F}$, i.e. $x \in \text{span}(\mathcal{H}_\mathcal{F})$, and the reverse inclusion is immediate. We claim now that $\mathcal{H}_\mathcal{F}$ is leakless and closed under

better replies for the dynamics restricted to $\mathcal{F}$. For leaklessness, let $\mathcal{F}' \subseteq \mathcal{F}$ be any face with $\mathcal{F}' \subseteq \text{span}(\mathcal{H}_\mathcal{F})$ and let $\alpha \in \mathcal{A}(\mathcal{F}) \setminus \mathcal{H}_\mathcal{F}$. Then $\alpha \notin \mathcal{H}$ and $\mathcal{F}' \subseteq \text{span}(\mathcal{H}) = \mathcal{S}$, so by leaklessness of $\mathcal{H}$ and Lemma G.6,

$$\sup_{x \in \mathcal{F}'} l_\alpha(x) \leq 0.$$

the preceding inequality is exactly the leaklessness condition for $\mathcal{H}_\mathcal{F}$ on $\mathcal{F}$. For closure under better replies, let $\beta \in \mathcal{H}_\mathcal{F}$ and suppose $\beta \rightarrow \alpha$ is a better-reply edge with $\alpha \in \mathcal{A}(\mathcal{F})$. Then also $\beta \in \mathcal{H}$ and $\beta \rightarrow \alpha$ is a better-reply edge in the full game, so $\alpha \in \mathcal{H}$ because $\mathcal{H}$ is closed under better replies. As $\alpha \in \mathcal{A}(\mathcal{F})$, this gives $\alpha \in \mathcal{H} \cap \mathcal{A}(\mathcal{F}) = \mathcal{H}_\mathcal{F}$.

Fix a face $\mathcal{F}$ with $\mathcal{S}_\mathcal{F} \neq \varnothing$, and consider the face-restricted choice map $Q_\mathcal{B} : \mathcal{Y} \rightarrow \mathcal{F}^\circ$ and the corresponding facewise FTRL dynamics (FTRL-$\mathcal{B}$). Apply Lemma G.8 to the subgame on $\mathcal{F}$ (action sets $\mathcal{B}_i$) and the set $\mathcal{H}_\mathcal{F} \subseteq \mathcal{A}(\mathcal{F})$: by the previous paragraph, $\mathcal{H}_\mathcal{F}$ is leakless and closed under better replies on $\mathcal{F}$, so

$$E_\mathcal{F}(y) := \bar{W}_{\mathcal{H}_\mathcal{F}}\big(Q_\mathcal{B}(y)\big)$$

is a local energy function for $\mathcal{S}_\mathcal{F} = \text{span}(\mathcal{H}_\mathcal{F})$ under (FTRL-$\mathcal{B}$). Hence, by Theorem G.2, $\mathcal{S}_\mathcal{F}$ is asymptotically stable under (FTRL-$\mathcal{B}$). By Remark E.9, the strategy flow trajectory starting from any $x_0 \in \mathcal{F}$ coincides with the orbit of (FTRL-$\mathcal{B}$) on the minimal face of $\mathcal{F}$ containing $x_0$, therefore $\mathcal{S}_\mathcal{F}$ is asymptotically stable for the restriction of the strategy flow to $\mathcal{F}$. Concretely: for each face $\mathcal{F}$ with $\mathcal{S}_\mathcal{F} \neq \varnothing$, there exists a neighborhood $U_\mathcal{F} \subseteq \mathcal{F}$ of $\mathcal{S}_\mathcal{F}$ (in the relative topology of $\mathcal{F}$) such that every strategy flow trajectory starting in $U_\mathcal{F}$ remains in $U_\mathcal{F}$ and satisfies $\text{dist}(X_t, \mathcal{S}_\mathcal{F}) \rightarrow 0$.

Let now $\mathscr{F}_\mathcal{S}$ be the (finite) collection of faces $\mathcal{F}$ of $\mathcal{X}$ such that $\mathcal{S}_\mathcal{F} \neq \varnothing$, and for each such $\mathcal{F}$ let $U_\mathcal{F} \subseteq \mathcal{F}$ be as above. Fix any point $x \in \mathcal{S}$. Let $\mathscr{F}(x) := \{\mathcal{F} \in \mathscr{F}_\mathcal{S} : x \in \mathcal{F}\}$. Because $\mathcal{X}$ has finitely many faces, the quantity

$$r_x := \tfrac{1}{2} \min\{\text{dist}(x, \mathcal{F}') : \mathcal{F}' \text{ a face of } \mathcal{X}, \ x \notin \mathcal{F}'\}$$

is well-defined and strictly positive. In particular, $B_{r_x}(x) \cap \mathcal{X}$ meets only faces that contain $x$. Moreover, for each $\mathcal{F} \in \mathscr{F}(x)$, since $U_\mathcal{F}$ is a neighborhood of $x$ in $\mathcal{F}$, there exists $\rho_\mathcal{F}^x > 0$ such that $B_{\rho_\mathcal{F}^x}(x) \cap \mathcal{F} \subseteq U_\mathcal{F}$. Set

$$\rho_x := \min\Big(r_x, \min_{\mathcal{F} \in \mathscr{F}(x)} \rho_\mathcal{F}^x\Big) > 0.$$

Then for every $z \in B_{\rho_x}(x) \cap \mathcal{X}$, the (unique) minimal face $\mathcal{F}_z$ containing $z$ must contain $x$, hence belongs to $\mathscr{F}(x)$, and therefore $z \in U_{\mathcal{F}_z}$. Thus the family $\{B_{\rho_x}(x) \cap \mathcal{X}\}_{x \in \mathcal{S}}$ covers $\mathcal{S}$. By compactness of $\mathcal{S}$, there exist $x^1, \ldots, x^m \in \mathcal{S}$ such that

$$\mathcal{S} \subseteq U := \bigcup_{k=1}^m \big(B_{\rho_{x^k}}(x^k) \cap \mathcal{X}\big),$$

and $U$ is an open neighborhood of $\mathcal{S}$ in $\mathcal{X}$. Now take any $X_0 \in U$ and let $\mathcal{F}$ be the minimal face of $\mathcal{X}$ containing $X_0$. By construction of $U$, $\mathcal{F}$ contains some $x^k \in \mathcal{S}$ and $X_0 \in U_\mathcal{F}$. Since faces are invariant (Proposition E.7), $X_t \in \mathcal{F}$ for all $t \geq 0$, and we have $\text{dist}(X_t, \mathcal{S}_\mathcal{F}) \rightarrow 0$. As $\mathcal{S}_\mathcal{F} \subseteq \mathcal{S}$, this implies $\text{dist}(X_t, \mathcal{S}) \rightarrow 0$. This proves that $\mathcal{S}$ is attracting. The stability part is obtained by the same patching argument, using the stability component of asymptotic stability on each $\mathcal{F} \in \mathscr{F}_\mathcal{S}$.

Therefore $\mathcal{S} = \text{span}(\mathcal{H})$ is compact, invariant, and asymptotically stable for the strategy flow, hence an attractor. Since this flow coincides with (RD) in the entropic case, $\text{span}(\mathcal{H})$ is an attractor under (RD). □

