# OpenReview forum: "What Preferences Can—and Cannot—Predict in Multi-Agent Online Learning"
_ICML.cc/2026/Conference — ICML 2026 spotlight_

### Official Review · Reviewer_2wjx · 2026-02-20

**Soundness:** 3
**Presentation:** 2
**Significance:** 2
**Originality:** 3
**Overall Recommendation:** 5
**Confidence:** 2

**Summary:**

This work’s major theme is the interplay between ordinal preference-based solution concepts in games and the outcomes of no-regret learning dynamics, particularly focusing on how preference graphs constrain but do not always determine long-run behavior. It aims to understand whether and when preferences alone can predict stable outcomes under adaptive dynamics like FTRL, SD, which is critical for applications where payoff magnitudes may not align with ordinal incentives.

Key Contributions:
1. Constructed 3-player counterexample demonstrates that preferential stability does not guarantee dynamic stability, refuting a prior conjecture.
2. For subgame, it shows that dynamically stable sets must be closed under profitable deviations (preferentially stable), establishing necessary conditions for stability based on preference graphs.
3. Introduces a cardinal condition, leaklessness, to restore stability, providing a sufficient payoff-based criterion for asymptotic stability in general games.

**Compliance With Llm Reviewing Policy:**

Affirmed.

**Final Justification:**

The authors provide some examples how these theories can be applied into practice. My concerns have been adequately addressed. I change my score to accept.

**Key Questions For Authors:**

What I care about most is how these theories can be applied to solve real-world problems. Could the authors provide some actionable insights or directions?

**Limitations:**

No.

**Strengths And Weaknesses:**

Soundness
The theoretical claims are rigorously supported. The counterexample (Section 5.2) effectively demonstrates the insufficiency of ordinal information.

Presentation
1. Lack of Related Work: One alone section  to show the difference and improvement compared with the previous is necessary, which can better help me understand the value of the work.
2. Figures are distant from their textual references (e.g., Fig. 1 on p.2 is first cited on p.7). Fig. 3 is never explicitly cited.

Significance
The work advances game-theoretic learning by formalizing when ordinal preferences suffice for stability and when cardinal information is needed. However, practical implications are underdeveloped:
1. Applications to multi-agent RL or mechanism design are mentioned only briefly.
2. The leaklessness condition, while novel, lacks empirical validation or algorithmic insights.

Originality
1. The counterexample resolving Biggar and Papadimitriou’s conjecture is a notable contribution.
2. Leaklessness is a novel concept, but its connection to existing stability notions (e.g., evolutionary stability) is unclear.

Overall, technically, this might be a solid piece of work, but I fail to see the potential application value of the research problem addressed. Additionally, there are some minor issues in the presentation,

---

> ### Author Rebuttal · Authors · 2026-03-31
>
> Dear reviewer,
>
> Thank you for your time. We respond to your remarks and questions below.
>
> > One alone section to show the difference and improvement compared with the previous is necessary, which can better help me understand the value of the work.
>
> Our paper bridges several topics, so providing a standalone related-work section while keeping a leisurely exposition within 8 pages is very difficult. Our guiding principle throughout was to be comprehensible rather than comprehensive: we therefore included only references that were either fundamental to the questions at hand or closely related to our analysis.
>
> That said, we appreciate your point, and we will be happy to use the extra page available at the camera-ready stage to expand the comparison with a survey of a broader body of prior work.
>
> > Figures are distant from their textual references (e.g., Fig. 1 on p.2 is first cited on p.7). Fig. 3 is never explicitly cited.
>
> This is easy to fix. Consider it already done :-)
>
> > Applications to multi-agent RL or mechanism design are mentioned only briefly.
>
> Again, space constraints forced us to make difficult choices in order to streamline the presentation of our theoretical contributions, which are the main focus of the paper. Still, these results were motivated by practical concerns, namely: when can one ensure that certain outcomes are robust under learning?
>
> As an example, consider a swarm of robots transporting objects through a narrow region, where the designer wants to avoid congestion and deadlock (for a survey, see Stern et al., 2019, "Multi-agent pathfinding: Definitions, variants, and benchmarks"). These are naturally ordinal objectives, yet they must be encoded through numerical rewards. Leaklessness suggests how to choose these payoffs so that coordinated outcomes—those in which exactly one robot passes while the others wait—remain stable under learning.
>
> A similar issue arises in modern preference-learning and elicitation settings, where one starts from ordinal feedback and fits a scalar objective. Particularly relevant for us is recent work on preference alignment for language models, which formulates the problem explicitly in game-theoretic terms and studies it through regularized no-regret learning dynamics of the same type as ours (for example, Zhang et al., 2024, "Iterative Nash policy optimization: Aligning LLMs with general preferences via no-regret learning"). Our results suggest that leaklessness may provide a useful design principle when such ideas are extended to richer multi-agent settings.
>
> Thus, while our paper does not provide an operational pipeline for this design problem, it does identify the conceptual and computational principles needed to formulate it in these settings, and it opens the door to exciting applications in multi-agent RL.
>
> > Leaklessness is a novel concept, but its connection to existing stability notions (e.g., evolutionary stability) is unclear [and] lacks empirical validation or algorithmic insights.
>
> Two points are important to note here.
>
> First, regarding evolutionary stability, in finite asymmetric games—the setting of our paper—any notion of evolutionary stability reduces to strict Nash equilibrium (Weibull, 1995, Chap. 4). [By contrast, in symmetric single-population games, evolutionarily stable strategies may be fully mixed.] In a certain, precise sense, strictly leakless sets are a setwise generalization of strict Nash equilibria, not only strategically (cf. Proposition 3) but also dynamically (Theorem 5). In this sense, strict leaklessness may be viewed as a setwise generalization of evolutionary stability in asymmetric finite games, while leaklessness is the counterpart of neutral evolutionary stability in the same setting.
>
> Second, from an algorithmic perspective, (strictly) leakless sets are, as we show, attractors of FTRL in continuous time. By the ODE method of stochastic approximation (Benaïm, 1999, "The dynamics of stochastic approximation algorithms"), it is therefore plausible that they are also attractors of algorithmic implementations of FTRL in discrete time, possibly with imperfect payoff observations or even bandit, realized-payoff feedback only. Of course, proving this rigorously would require a paper of its own; however, our paper provides the necessary first step for such an algorithmic analysis, which otherwise seems entirely out of reach.
>
> Finally, leaklessness has important advantages over the classical notion of mixed Nash equilibrium, both from a computational and learning viewpoint [For more details, see [our reply to Reviewer gSpL](https://openreview.net/forum?id=5W30WwL8wt&noteId=8tzQVyuBG6).]
>
> ---
>
> Thank you again for your time—we will be happy to incorporate the above points at the first revision opportunity. In the meantime, we trust and hope that our replies have addressed your concerns, and we look forward to any further questions you may have.
>
> Kind regards,
> The authors

---

> > ### Author Rebuttal · Reviewer_2wjx · 2026-04-02
> >
> > The authors provide some examples how these theories can be applied into practice. My concerns have been adequately addressed. I change my score to accept.

---

> > > ### Author Response · Authors · 2026-04-02
> > >
> > > Dear reviewer 2wjx,
> > >
> > > Thank you for your reply, your time, and your increased support, we are very happy that your concerns were adequately addressed.
> > >
> > > Kind regards,
> > >
> > > The authors

---

### Official Review · Reviewer_Jgzi · 2026-03-07

**Soundness:** 3
**Presentation:** 3
**Significance:** 4
**Originality:** 3
**Overall Recommendation:** 5
**Confidence:** 3

**Summary:**

This paper studies when ordinal preference information, captured by a game’s preference graph over pure profiles, can predict the long-run outcomes of payoff-driven no-regret learning dynamics, focusing on FTRL / regularized dynamics (with replicator as a special case). The authors first establish necessary links from dynamical stability to ordinal closure properties at the level of pure profiles, showing that stable or attracting sets must satisfy appropriate preference-graph closure conditions. They then identify a regime where preferences are sufficient, showing that for subgames, the span is asymptotically stable if and only if it is closed under better replies (Theorem 3). Beyond subgames, they provide a 3-player counterexample in which a preferentially stable pure set has an unstable span, demonstrating that preferences alone are insufficient in general. To recover stability guarantees, they introduce leaklessness, a cardinal condition depending on payoff magnitudes, and prove attractor results for (strictly) leakless sets/spans, including a particularly simple energy-based argument in the entropic / replicator case (Theorems 5–6).

**Compliance With Llm Reviewing Policy:**

Affirmed.

**Final Justification:**

The rebuttal has addressed my main concerns.

**Key Questions For Authors:**

1. Could the authors provide a clearer high-level roadmap of how the main results fit together?
In particular, I would appreciate a more explicit summary of which results should be viewed as necessary conditions, which ones give positive characterizations (e.g., in the subgame setting), which ones are separation/counterexample results, and where leaklessness fits into this overall picture.

2. What is the key structural reason that product structure makes ordinal information sufficient in Theorem 3?
I found Theorem 3 very interesting, but it would help to have a more intuitive explanation of why subgames are exactly the regime in which closure in the preference graph becomes equivalent to asymptotic stability of the span.

**Limitations:**

yes

**Strengths And Weaknesses:**

Strengths
* Strong theoretical scope and coherence: The paper addresses a clean and fundamental question: to what extent ordinal preference information can predict long-run behavior under regularized no-regret learning dynamics. I found the overall arc of the paper conceptually strong, moving from necessary constraints, to a positive characterization in the subgame setting, to a counterexample beyond that regime, and finally to a payoff-based condition that restores stability guarantees.

* A novel and meaningful conceptual contribution: The introduction of leaklessness is, in my view, one of the paper’s most interesting ideas. It provides a principled way to incorporate cardinal payoff information exactly where the ordinal perspective becomes insufficient, and it leads to attractive stability guarantees, including refined results in the entropic / replicator setting.

* Interesting and nontrivial technical contributions: The paper appears technically substantial, with mathematically precise claims and a coherent progression of results. In particular, the positive characterization for subgames, the separation result beyond subgames, and the introduction of leaklessness together make for a meaningful and well-rounded theoretical contribution.

* High Relevance to Multi-Agent ML: The results offer significant insights for the online learning and reward design communities. By clarifying when purely ordinal specifications are predictive and when cardinal magnitudes qualitatively shift learning outcomes, the work provides a valuable compass for researchers designing preference-based specifications in multi-agent environments.


Weakness:
* The paper appears technically strong, but I found the exposition difficult to follow. The presentation moves across several layers of abstraction (graph structure, spans/faces, and continuous-time dynamics), which made it hard to build intuition for the main results and to confidently assess which ingredients are conceptually central versus mainly technical.
* The paper is clearly relevant to game dynamics and no-regret learning theory, but its broader ML significance remains somewhat indirect. The connection to multi-agent learning and reward design is plausible, yet largely motivational rather than operational.
* While I do not think large-scale experiments are necessary for a theory paper of this type, the paper would benefit substantially from a few simple trajectory visualizations or toy simulations illustrating the key positive/negative examples.

---

> ### Author Rebuttal · Authors · 2026-03-31
>
> Dear reviewer,
>
> Thank you for your time and positive evaluation! We reply to your remarks and questions below:
>
> > The paper appears technically strong, but the presentation moves across several layers of abstraction (graph structure, spans/faces, and continuous-time dynamics), which made it hard to build intuition for the main results. [...] Could the authors provide a clearer high-level roadmap of how the main results fit together? What is the key structural reason that product structure makes ordinal information sufficient in Theorem 3?
>
> We understand your concern but, at the same time, our paper bridges many topics, which are difficult to present at a leisurely pace in the span of 8 pages. That being said, your suggestion of providing a roadmap is right on point, and we will be happy to include one at the first revision opportunity.
>
> In a nutshell, the main elements are as follows:
> 1. First, we show that dynamic stability implies preferential stability.
> 2. We then show that the converse is much more subtle:
>     - For subgames, we prove that the converse *does* hold, so there is an equivalence between preferential and dynamic stability. [In this case, "*preferences are enough*"]
>     - For general subsets of pure strategy profiles, *this is no longer the case:* we provide a specific counterexample of a set of pure strategy profiles which is closed under better replies (club), but whose span is not stable under FTRL. [In this case, "*preferences are *not* enough*"]
> 3. To recover a partial converse, we introduce the (cardinal) notion of leaklessness, and we show that strictly leakless sets are asymptotically stable under FTRL.
>
> The key in the above lies in the geometry of the sets under consideration and how it ties to the game's underlying strategic structure.
> 1. For subgames, spans correspond to faces, i.e., *uncoupled* restrictions of supports: each agent mixes over a restricted action set independently of the others. If such a face is club, each agent has a local incentive to remain within said face regardless of what the others do. This uncoupledness is precisely what makes preferential stability sufficient for dynamic stability in this case.
> 2. By contrast, a general span is a union of faces, so it may contain _coupled_ regions. In this setting, an agent’s locally preferred deviation may depend on the precise (cardinal) behavior of the others, even close to the span in question. This creates an inherent coordination issue: agents may have individually rational deviations that are not aligned with each other.
> 3. Leaklessness addresses this issue by imposing a form of aggregate stability that rules out such miscoordinations: even if some agents have a local incentive to deviate, the combined effect of unilateral deviations keeps the trajectory near the span, and dynamic stability is recovered.
>
> > The paper is clearly relevant to game dynamics and no-regret learning theory, but its broader ML significance remains somewhat indirect.
>
> While our focus is indeed theoretical, our motivation is largely operational.
>
> To wit, consider the following example from autonomous driving: when designing rewards and penalties for an RL model, one must clearly assign a much larger penalty to a state where changing lanes could lead to an accident, relative to the reward of utilizing a less congested lane. In practice, for the most part, penalties and rewards are assigned arbitrarily—or, at best, empirically. But this leads to a natural question: how robust is the resulting RL model to the chosen penalties and rewards? Do the rewards matter, or are ordinal preferences enough in this case?
>
> Other examples could easily be found in the context of auction and market design, recommender systems, multi-robot coordination, preference-learning, etc. Our paper remains application-agnostic, but it lays down the groundwork and opens the door for such studies; in particular, examining these questions in a bona fide multi-agent RL setting—a stochastic/Markov game—is a very fruitful direction for future research that our paper points to. [See also the relevant part of [our reply to Reviewer 2wjx](https://openreview.net/forum?id=5W30WwL8wt&noteId=BDqDog4MKJ)]
>
> > While I do not think large-scale experiments are necessary for a theory paper of this type, the paper would benefit substantially from a few simple trajectory visualizations or toy simulations illustrating the key positive/negative examples.
>
> Thanks for this remark. In addition to the ones provided in the main text, we provide additional numerics in Appendix F (p33), and we will be happy to complement our slate of numerical results with further games, illustrating in particular the different graphical configurations that can occur in 3x2 and 2x2x2 games (corresponding to the examples of Appendix A).
>
> ---
> Thank you again for your time and positive evaluation—and please let us know if you have any further questions in the meantime.
>
> Kind regards,
>
> The authors

---

> > ### Author Rebuttal · Reviewer_Jgzi · 2026-04-02
> >
> > thanks for your reply, i think you have solved my questions.

---

> > > ### Author Response · Authors · 2026-04-02
> > >
> > > Dear reviewer Jgzi,
> > >
> > > Thank you for your reply, your detailed input, and your continued support, we are very happy that your concerns were adequately addressed.
> > >
> > > Kind regards,
> > >
> > > The authors

---

### Official Review · Reviewer_FpTu · 2026-03-12

**Soundness:** 3
**Presentation:** 3
**Significance:** 3
**Originality:** 3
**Overall Recommendation:** 5
**Confidence:** 4

**Summary:**

This paper deals with finite normal form games and studies the fundamental connections between the preference graph and online no-regret regularized learning dynamics of such games. The paper aims to study whether the performance of learning dynamics can reveal information about preferential stability, and vice versa. In the former direction, the paper establishes that regions stable/attracting of the learning dynamics indeed correspond to preferentially stable graph components. The paper shows that the latter direction contains more subtleties - for subgames, preferential stability does span stable/attracting regions for the learning dynamics, however, the paper shows by counter example that this need not be true for any (non-subgame) arbitrary collection of preference graph vertices. Following this shortfall of preferences to predicting stability by themselves, the paper introduces a new condition for games: leaklessness, which as an additional information is sufficient to link preferential stability with learning stability/attraction.

**Compliance With Llm Reviewing Policy:**

Affirmed.

**Final Justification:**

The authors have addressed my questions and I maintain a positive opinion of the paper.

**Key Questions For Authors:**

- I have some trouble understanding the use of the Fenchel gap in Theorem 3. Can the authors clarify on what is different about it from prior work which enables its use with non-steep regularizers?
- Complete nitpick, but for the sake of mathematical clarity, I ask the authors to formally define the notion of a "proper" attractor in line of theorem 2 (I assume by proper they mean the attractor is a strict subset of $\mathcal{X}$)

**Limitations:**

yes

**Strengths And Weaknesses:**

- The results of the paper are very fundamental and will help the design and analysis of algorithms for learning in games.
- The notion of leaklessness is interesting, and I agree with the authors in that it opens a new avenue for studying long term behavior of learning dynamics in the future.

I am hard pressed to point out "weaknesses" as the paper presents fundamental theory. Instead, I ask that the authors clarify some points which is listed in the questions below

---

> ### Author Rebuttal · Authors · 2026-03-31
>
> Dear reviewer,
>
> Thank you for your time and positive evaluation! We reply to your remarks and questions below:
>
> > I have some trouble understanding the use of the Fenchel gap in Theorem 3. Can the authors clarify on what is different about it from prior work which enables its use with non-steep regularizers?
>
> The first thing to note is that the Fenchel gap can be seen as a setwise generalization of the Fenchel coupling $$F(p,y) = h(p) + h^\ast(y) - \langle y,p\rangle$$ itself a "primal-dual" version of the Bregman divergence $$D(p,x) = h(p) - h(x) - \langle\nabla h(x),p-x\rangle$$
> To provide some context, if the score profile $y$ is mapped to the mixed strategy profile $x=Q(y)$ under the regularized choice map $Q$ that defines FTRL, we have $F(p,y) \geq D(p,x)$, with equality whenever $x$ is fully mixed. This is always the case if $h$ is steep, but not otherwise; if it isn't, then the Bregman divergence can no longer be used as a Lyapunov function for FTRL, because $x$ can get "stuck" in the space of strategies, even though $y$ continues to evolve in the space of payoffs. By contrast, the Fenchel coupling carries all payoff-relevant information, so it is remains a valid Lyapunov function for FTRL.
>
> The Fenchel gap provides a setwise generalization of the Fenchel coupling by setting $$F_S(y) = \inf_{p\in S} F(p,y)$$i.e., by looking at the minimal "primal-dual" distance between $S$ and $y$. In this regard, the Fenchel gap can be seen as a primal-dual measure of distance between a *set* of strategy profiles $S$ and the score variable $y$, so it is a natural Lyapunov candidate for studying the evolution of FTRL relative to $S$. As above, if one where to consider the setwise Bregman divergence $$D_S(x) = \inf_{p\in S} D(p,x)$$the same problem would arise, namely, the score variable $y$ might evolve under FTRL, but it could be mapped to the same strategy profile $x\in\mathcal{X}$, thus rendering the "primal-primal" measure $D_S(x)$ useless as a Lyapunov function. It is precisely this duality that enables the use of the Fenchel gap for setwise solution concepts and non-steep regularizers—and what sets it apart from previous work on the subject.
>
> As for our use of the Fenchel gap in the proof of Theorem 3, it is fairly technical but the main idea is that the time derivative of the Fenchel gap along the FTRL trajectory can be written as the sum, over all players, of the current payoff of each player minus the payoff that player would obtain after projecting their strategy onto the face spanned by the subgame—under the notion of projection induced by the regularizer (i.e. the choice map restricted to the face). We then show, via a novel trick exploiting the simplex-constrained first-order conditions and the decomposability of the regularizer, that this difference is locally negative near the face, meaning that all players locally strictly prefer to project their strategy onto the face. This is precisely what drives convergence to it.
>
> We hope this clarifies the point, and we would be happy to elaborate further if useful.
>
> > Complete nitpick, but for the sake of mathematical clarity, I ask the authors to formally define the notion of a "proper" attractor in line of theorem 2 (I assume by proper they mean the attractor is a strict subset of $\mathcal{X}$)
>
> Indeed. Will do—and thanks for pointing it out, we greatly appreciate your attention to detail.
>
> ---
> Thank you again for your time and positive evaluation—and please let us know if you have any further questions!
>
> Kind regards,
>
> The authors

---

> > ### Author Rebuttal · Reviewer_FpTu · 2026-04-03
> >
> > Thank you for answering my questions. I maintain a positive opinion of the paper.

---

> > > ### Author Response · Authors · 2026-04-04
> > >
> > > Dear Reviewer FpTu,
> > >
> > > Thank you for your reply, your thoughtful feedback, and continued support. We are very pleased that our response addressed your concerns.
> > >
> > > Kind regards,
> > >
> > > The authors

---

### Official Review · Reviewer_gSpL · 2026-03-13

**Soundness:** 4
**Presentation:** 3
**Significance:** 3
**Originality:** 3
**Overall Recommendation:** 5
**Confidence:** 2

**Summary:**

The paper asks the question when preference orderings are sufficient to determine the outcome of multi-agent learning dynamics. They find that ordinal stability to deviations forces cardinal stability to deviations, provide a counterexample for which in general preference orderings are insufficient, and provide an axiom that forces consistency.

**Compliance With Llm Reviewing Policy:**

Affirmed.

**Final Justification:**

I am in favor of acceptance.

**Key Questions For Authors:**

How strong is the leak condition?

**Limitations:**

yes

**Strengths And Weaknesses:**

Strenghts:
 - The paper is clear and sound
 - The question of ordinal vs. cardinal preferences goes back to early questions aroudn the utility foundations of modern economics (jstor.org/stable/2548836)

Weaknesses:
 - It is unclear how strong the leak condition is.

---

> ### Author Rebuttal · Authors · 2026-03-31
>
> Dear reviewer,
>
> Thank you for your time and strongly positive evaluation! We reply to your remarks and questions below:
>
> > How strong is the leak condition?
>
> There are several points to keep in mind here.
>
> First, every game admits at least one leakless set and at least one *strictly* leakless set. In this regard, leakless sets are similar to mixed Nash equilibria—that is, every game admits one—and can be seen as a *setwise* relaxation of pure Nash equilibria—in the same sense that mixed Nash equilibria can be seen as a *convex* relaxation thereof. On the other hand, unlike mixed Nash equilibria—which are *never* asymptotically stable under FTRL—strictly leakless sets are always asymptotically stable, so they can be seen as the "proper" generalization of strict Nash equilibria, from both a strategic and a dynamic viewpoint.
>
> In terms of computational complexity, leaklessness is substantially easier to handle than Nash equilibrium in the explicit normal-form model, where the game's payoff table is known. Indeed, one can find a leakless set in polynomial time—in fact, linear in the number of players and quadratic in the total number of pure strategy profiles. By contrast, computing a mixed Nash equilibrium in this same normal-form setting is famously known to be computationally inaccessible (PPAD-complete to be exact, by the seminal result of Daskalakis et al., 2009, "The complexity of computing a Nash equilibrium").
>
> With regard to other setwise solution concepts, strict leaklessness is stronger than closedness under better replies, so one cannot in general expect it to yield non-trivial outcomes when the preference graph is strongly connected (in which case it may span the entire graph). Nonetheless, except for this fringe case (where myopic players would not converge to any proper subset of pure strategy profiles), leaklessness provides a natural way to identify subsets of pure profiles toward which players, in aggregate, tend to deviate. In this sense, leaklessness is a condition that ensures local coordination of agents toward a given outcome. While we do not claim that all forms of stable and coordinated outcomes arise from leakless sets, this condition provides, to the best of our knowledge, the first simple, efficient and natural criterion for identifying such setwise outcomes in general normal-form games.
>
> ---
> Thank you again for your time and positive evaluation—and please let us know if you have any further questions!
>
> Kind regards,
>
> The authors

---

> > ### Author Rebuttal · Reviewer_gSpL · 2026-04-03
> >
> > Thanks, keeping my score. Well done!

---

> > > ### Author Response · Authors · 2026-04-04
> > >
> > > Dear Reviewer gSpL,
> > >
> > > Thank you for your reply, your positive feedback, and your continued support. We are very happy that our response adequately addressed your concerns.
> > >
> > > Kind regards,
> > >
> > > The authors

---

### Decision · Program_Chairs · 2026-04-30

**Decision:**

Accept (spotlight)

**Comment:**

The paper studies how ordinal, preference-based solution concepts relate to the long-run behavior of payoff-driven learning dynamics such as no-regret algorithms. It shows that dynamically stable sets must be preferentially stable, and that preferences fully characterize asymptotic stability for subgames, but not in general, as demonstrated by a three-player counterexample. To address this gap, the paper introduces the notion of leaklessness, providing a payoff-based condition that ensures dynamic stability and attraction of sets of pure strategy profiles.

All the Reviewers were very positive on this paper, thus I strongly recommend acceptance.